# Activated protein C protects from GvHD via PAR2/PAR3 signalling in regulatory T-cells

Satish Ranjan[1], Alexander Goihl[2], Shrey Kohli[1], Ihsan Gadi[1], Mandy Pierau[3], Khurrum Shahzad[1,4], Dheerendra Gupta[1], Fabian Bock[1], Hongjie Wang[1], Haroon Shaikh[1], Thilo Kähne[5], Dirk Reinhold[2], Ute Bank[2], Ana C. Zenclussen[6], Jana Niemz[7], Tina M. Schnöder[8,9,10], Monika Brunner-Weinzierl [3], Thomas Fischer[10], Thomas Kalinski[11], Burkhart Schraven[2,7], Thomas Luft[12], Jochen Huehn[7], Michael Naumann[5], Florian H. Heidel[8,9,10] & Berend Isermann[1]

Graft-vs.-host disease (GvHD) is a major complication of allogenic hematopoietic stem-cell (HSC) transplantation. GvHD is associated with loss of endothelial thrombomodulin, but the relevance of this for the adaptive immune response to transplanted HSCs remains unknown. Here we show that the protease-activated protein C (aPC), which is generated by thrombomodulin, ameliorates GvHD aPC restricts allogenic T-cell activation via the protease activated receptor (PAR)2/PAR3 heterodimer on regulatory T-cells ($T_{regs}$, CD4$^+$FOXP3$^+$). Preincubation of pan T-cells with aPC prior to transplantation increases the frequency of $T_{regs}$ and protects from GvHD. Preincubation of human T-cells (HLA-DR4$^-$CD4$^+$) with aPC prior to transplantation into humanized (NSG-AB°DR4) mice ameliorates graft-vs.-host disease. The protective effect of aPC on GvHD does not compromise the graft vs. leukaemia effect in two independent tumor cell models. Ex vivo preincubation of T-cells with aPC, aPC-based therapies, or targeting PAR2/PAR3 on T-cells may provide a safe and effective approach to mitigate GvHD.

[1] Institute of Clinical Chemistry and Pathobiochemistry, Otto-von-Guericke- University Magdeburg, Leipziger Str. 44, 39120 Magdeburg, Germany. [2] Institute of Molecular and Clinical Immunology, Otto-von-Guericke-University Magdeburg, Leipziger Str. 44, Magdeburg 39120, Germany. [3] Department of Experimental Pediatrics, Otto-von-Guericke-University Magdeburg, Leipziger Str. 44, Magdeburg 39120, Germany. [4] Department of Biotechnology, University of Sargodha, Sargodha 40100, Pakistan. [5] Institute of Experimental Internal Medicine, Center of Internal Medicine, Otto-von-Guericke University Magdeburg, Leipziger Str. 44, Magdeburg 39120, Germany. [6] Experimental Obstetrics and Gynecology, Medical Faculty, Otto-von-Guericke University, Magdeburg 39108, Germany. [7] Department of Experimental Immunology, Helmholtz Centre for Infection Research (HZI), Inhoffenstrasse 7, Braunschweig 38124, Germany. [8] Internal Medicine II, Hematology and Oncology, University Hospital Jena, Am Klinikum 1, 07747 Jena, Germany. [9] Leibniz-Institute on Aging, Fritz-Lipmann-Institute, 07745 Jena, Germany. [10] Department of Hematology and Oncology, Center of Internal Medicine, Otto-von-Guericke University Magdeburg, Leipziger Str. 44, Magdeburg 39120, Germany. [11] Institute for Pathology, Otto-von-Guericke University Magdeburg, Leipziger Str. 44, Magdeburg 39120, Germany. [12] Department of Medicine V, University of Heidelberg, Im Neuenheimer Feld 410, Heidelberg 69120, Germany. Correspondence and requests for materials should be addressed to B.I. (email: berend.isermann@med.ovgu.de)

A subset of malignant and non-malignant hematological diseases can exclusively be cured by cellular immunotherapy, namely allogenic hematopoietic stem-cell transplantation (HSCT)[1]. However, the success of HSCT is impacted by graft-vs.-host disease (GvHD), a potentially lethal complication[1]. Acute GvHD can be distinguished from chronic GvHD based on the timeframe and organ involvement[1]. Acute GvHD, which affects up to 60% of patients, primarily affects three organ systems (skin, liver, and gastrointestinal tract)[2]. Current GvHD prophylaxis and treatment are only partially effective, with an increased risk for infections, disease relapse, and long-term adverse effects[3]. High-dose steroids remain the standard therapy for acute GvHD, but carries significant risks[4]. Furthermore, some patients fail to respond to steroid therapy, resulting in steroid-resistant GvHD. Thus, there remains a medical need to identify new therapies mitigating GvHD. Suppression of the transplanted immune system, aiming to restrict its activity against non-malignant host-cells and thus limiting GvHD, has to be balanced with sustained activity of the transplanted immune system against tumour cells, which determines the success of HSCT in the context of malignant haematological diseases[5]. Pre-clinical and clinical studies suggest that regulatory T-cells (T$_{regs}$) hold promise to address this therapeutic need[6, 7]. One of the major challenges remaining is the identification of efficient and safe methods for robust expansion of donor-derived T$_{regs}$[8, 9].

Analyses of steroid-resistant GvHD revealed involvement of endothelial dysfunction, e.g. increased serum levels of soluble thrombomodulin (TM)[10–13], which reflect loss of endothelial TM function[14]. Targeting TM-dependent effects may hence constitute a new therapeutic approach to mitigate GvHD. Indeed, pre-clinical studies in mice suggested that soluble TM ameliorates GvHD, but the underlying mechanism remained unknown[15]. TM is required for efficient activation of the anticoagulant and cytoprotective signaling-competent protease-activated protein C (aPC)[14, 16]. aPC signals predominantly via G-protein coupled protease activated receptors (PARs) in a cell- and context-specific manner[17–19]. The role of aPC in innate immunity is firmly established[17], whereas its role in adaptive immunity and in particular on T-cells remains largely unknown. In a series of elegant reports Hancock et al.[20] studied the effect of aPC in solid organ transplantation, focusing, however, on innate immune mechanisms. In addition, previous work showed that aPC dampens activation of effector T-cells and increases the frequency of T$_{regs}$ in a model of type 1 diabetes mellitus, but the underlying mechanism, e.g. which immune cell type is targeted by aPC and the receptors involved, remained unknown[21]. Considering the loss of TM in GvHD, the known cytoprotective effects of aPC, and the development of new and safer aPC-based drugs we investigated aPC's role in acute GvHD. Using a combination of in vivo and in vitro approaches we show that aPC signaling in

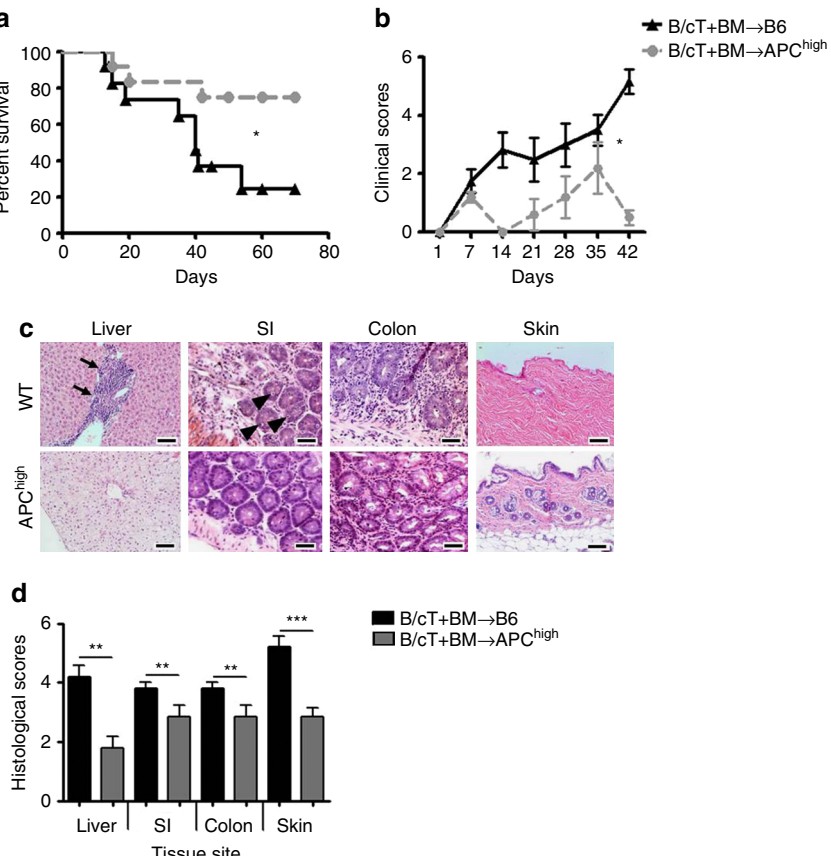

**Fig. 1** aPC ameliorates murine GvHD. **a**, **b** Recipient C57BL/6 wild-type (B6) mice or C57BL/6 mice with endogenous high levels of aPC (APC$^{high}$) were lethally irradiated (13 Gy) and transplanted with $5 \times 10^6$ whole-bone marrow (*BM*) and $2 \times 10^6$ T-cells from donor BALB/c mice (B/cT + BM). Recipient mice were monitored for survival (**a** Kaplan–Meier curve) and physical parameters (including weight loss, mobility, hunched posture, ruffled fur, and skin integrity), yielding a composite clinical score **b**; pooled data from three independent experiments each with four recipient mice per genotype. **c**, **d** Photomicrographs (**c**) depicting typical morphology in liver (note lymphocyte infiltration, *arrow*), small intestine (SI, note apoptotic bodies, *arrow head*), colon, and skin (note for example loss of hair follicles and epidermic atrophy) and bar graph (**d**) summarizing histological disease scores. Mean value ± SEM of six B6 recipient and six APC$^{high}$-recipient mice **d**; haematoxylin and eosin stained sections (**c** size bar: 50 µm); *$P < 0.05$, **$P < 0.01$, ***$P < 0.001$ (**a** log-rank test; **b** ANOVA; **d** *t*-test)

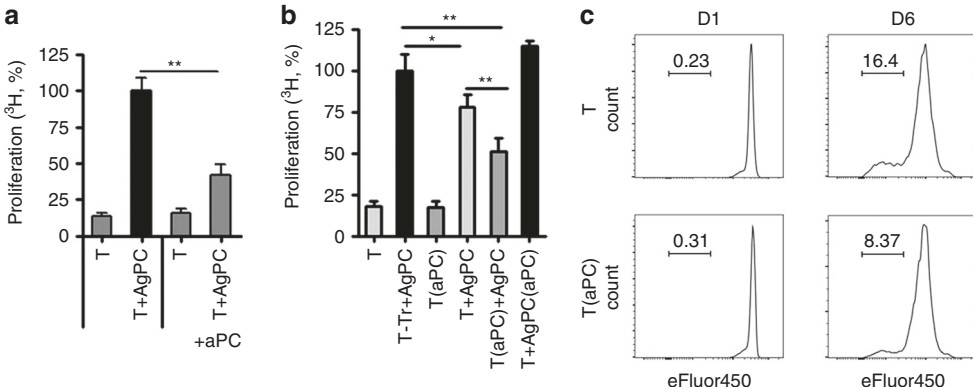

**Fig. 2** T-cell preincubation with aPC reduces allogenic T-cell reactivity and proliferation. **a** Splenic T-cells isolated from C57BL/6 wild-type (*wt*) mice (T) were co-cultured with BALB/c allogenic antigen-presenting cells (*AgPC*) for 96 h and T-cell reactivity was measured by thymidine incorporation during the final 16 h. Concomitant incubation with aPC (**a** +aPC, 20 nM, every 12 h) reduces T-cell reactivity. **b** aPC-preincubation (20 nM, 1 h, 37 °C) of mouse pan T-cells (T(aPC) + AgPC), but not of mouse AgPC (T + AgPC(aPC)), reduces T-cell reactivity in the MLR in comparison to $T_{reg}$-depleted T-cells (T-Tr + AgPC) and T-cells (T + AgPC). **c** Mouse pan T-cells without (T) and with (T(aPC)) aPC-preincubation were labelled with eFluor450 and then stimulated with plate bound αCD3 (1 µg/ml) and αCD28 (0.5 µg/ml) antibodies and proliferation was measured on day 1 (*D1*) and day 6 (*D6*) by FACS. Results of at least three independent experiments, each containing cells from three different mice **a**, **b** are shown; representative histogram plots **c** from triplicate wells of three independent experiments are shown. Mean value ± SEM **a**, **b**; *$P < 0.05$, **$P < 0.01$ (**a** t-test, **b** ANOVA)

T-cells via the PAR2/PAR3 heterodimer increases the frequency of $T_{regs}$, thus ameliorating GvHD without impeding the GvL effect.

## Results

### A hyperactivatable PC-mutant protects mice from GvHD.
To investigate the role of endogenous aPC in acute GvHD, we transplanted lethally irradiated C57BL/6 APC^high (transgenic mice expressing a hyperactivatable PC-mutant, resulting in elevated aPC plasma levels)[22] and C57BL/6 wild-type (wt) mice with $5 \times 10^6$ BM (bone marrow) cells and $2 \times 10^6$ splenic T-cells from BALB/c mice. Survival and physical appearance ("clinical" score composed of weight loss, mobility, hunched posture, ruffled fur, and skin integrity) were markedly improved in APC^high mice (Fig. 1a, b). Likewise, histopathological analysis of small and large bowel, liver, and skin demonstrated amelioration of GvHD in APC^high mice (Fig. 1c, d). Hence, endogenously generated aPC protects from GvHD.

### Preincubation of T-cells with aPC limits T-cell reactivity.
As GvHD is primarily a T-cell driven disease we next explored aPC's effect on T-cell activation using in vitro mixed lymphocyte reactions (MLRs)[23]. Co-culture of C57BL/6 wt pan T-cells with irradiated allogenic antigen-presenting cells (AgPC, BALB/c) for 96 h induced T-cell reactivity, which was markedly blunted in the presence of aPC (20 nM, every 12 h, Fig. 2a), establishing that aPC inhibits MLR-driven allogenic T-cell reactivity. Preincubation of T-cells with aPC (20 nM, once, 1 h, 37 °C, followed by a washing step to remove remaining aPC; T(aPC) + AgPC) was sufficient to inhibit allogenic T-cell reactivity in comparison to non aPC preincubated T-cells (T + AgPC) or the positive control $T_{reg}$-depleted T-cells (T-Tr + AgPC, Fig. 2b). In addition, preincubation of eFluor450 labelled T-cells with aPC reduced T-cell proliferation (Fig. 2c). Of note, preincubation of allogenic AgPC with aPC (T + AgPC(aPC)) following the same protocol had no effect on the MLR (Fig. 2b), suggesting that aPC directly acts on T-cells.

### T-cell preincubation with aPC amends GvHD and expands $T_{regs}$.
To assess the in vivo effect of aPC-preincubation on allogenic T-cells driven GvHD, donor T-cells were exposed to aPC (20 nM, once, 1 h, 37 °C) prior to transplantation. We transplanted $5 \times 10^6$ BM cells and $0.5 \times 10^6$ splenic T-cells (both C57BL/6-derived) without (B6T + BM) or with (B6T(aPC) + BM) aPC-preincubation into lethally irradiated BALB/c recipients. Engraftment of donor-derived cells (H2^b) and specifically of donor-derived CD3^+ (H2^b+CD3^+) or CD4^+ (H2^b+CD4^+) T-cells was not affected by T-cell preincubation with aPC prior to transplantation (Supplementary Fig. 1). Preincubation of allo-genic T-cells with aPC markedly improved survival, physical appearance, and histopathology in mice with GvHD (Fig. 3a–d).

To evaluate the mechanism underlying the protective effect of T-cell preincubation with aPC, we analysed splenocytes from GvHD mice on day 14 post transplantation ex vivo. aPC-preincubation of T-cells prior to transplantation increased the frequency of donor-derived regulatory T-cells (B6-H2^b+CD4^+FOXP3^+, $T_{regs}$) about fourfold (non aPC-preincubated, 1.7% vs. aPC-preincubated, 7.3%), whereas that of donor-derived Th1 (B6-H2^b+CD4^+T-bet^+) and Th17 (B6-H2^b+CD4^+ROR-γt^+) T-cells was markedly reduced (non aPC-preincubated, 16.2% vs. aPC-preincubated, 1.5%, and non aPC-preincubated, 13.4% vs. aPC-preincubated, 3.4%, respectively, Fig. 3e; Supplementary Fig. 2). Concomitantly, donor CD4^+ T-cells expressed less IFN-γ, TNF-α, and IL-17A, but more IL-10 in mice that received aPC-preincubated T-cells (Fig. 3f; Supplementary Fig. 3). Congruently, plasma levels of IFN-γ, TNF-α, IL-17A, and IL-6 were reduced, whereas plasma levels of TGFβ1 and IL-10 were increased in mice receiving aPC-preincubated T-cells (Fig. 3g). Although the plasma cytokine profile in mice is in agreement with the cytokine expression pattern observed in murine donor CD4^+ T-cells, it is likely that cells other than T-cells contributed to the plasma cytokine profile. Of note, IL-10 and TGFβ are produced by various cell types, including macrophages, neutrophils, or platelets, which are subject to regulation by coagulation proteases[24, 25]. These results establish that aPC-preincubation of donor T-cells improves GvHD, which is associated with a sustainable expansion of $T_{regs}$, reduction of Th1 effector cells, and a protective cytokine profile[26].

### aPC preincubation restricts allo-reactivity of human T-cells.
The effect of aPC on allogenic activation of human T-cells was next assessed[23]. Conducting the MLR using human T-cells and AgPC in the presence of aPC (20 nM, every 12 h) reduced T-cell reactivity to 67% as compared to the MLR without aPC (Fig. 4a).

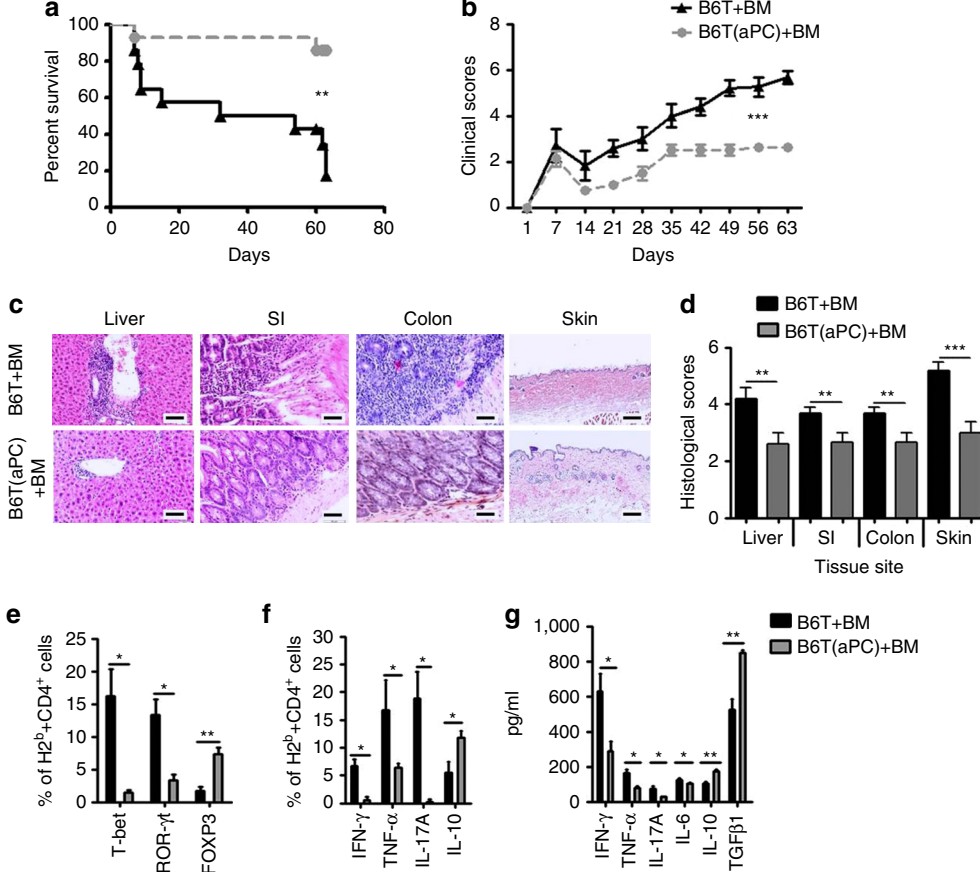

**Fig. 3** T-cell preincubation with aPC ameliorates GvHD in mice. **a**, **b** Recipient BALB/c mice were lethally irradiated (11 Gy) and transplanted with $5 \times 10^6$ BM and $0.5 \times 10^6$ T-cells without (B6T + BM) or with (B6T(aPC) + BM) aPC-preincubation (20 nM, 1 h, 37 °C) from donor C57BL/6 wt mice. Recipient mice were monitored for survival (**a** Kaplan–Meier curve) and physical parameters (**b** clinical score); pooled data from three independent experiments each with four recipients per group. **c**, **d** Photomicrographs (**c**) depicting typical morphology in liver, small intestine (*SI*), colon, and skin and bar graph (**d**) summarizing histological disease score. **e**, **f** In separate experiments recipients were euthanized 2 weeks post transplantation, splenic T-cells were harvested and stained for H2$^b$, CD4, T-bet, ROR-γt, or FOXP3 (**e**) or H2$^b$, CD4, IFN-γ, TNF-α, IL-17, or IL-10 (**f**) and analysed by flow cytometry; pooled data from three independent experiments each with four recipients per group. **g** Cytokines (IFN-γ, TNF-α, IL-17, IL-6, IL-10, and TGFβ1) were measured by ELISA in mouse plasma samples ($N = 12$ per group) collected from recipient mice euthanized 2 weeks post transplantation. Mean value ± SEM (**d–g**); haematoxylin and eosin stained sections (**c** size bar: 50 μm); MFI and exemplary FACS images corresponding to Fig. 3e, f are shown in Supplementary Figs. 2 and 3; *$P < 0.05$, **$P < 0.01$, ***$P < 0.001$ (**a** log-rank test; **b** ANOVA; **d–g**: *t*-test)

Furthermore, preincubation of human pan T-cells with aPC (T(aPC) + AgPC) reduced T-cell reactivity in the MLR, both in comparison to $T_{reg}$-depleted T-cells (53 vs. 100% $^3$H incorporation in (T-Tr) + AgPC, Fig. 4b) and in comparison to pan T-cells without aPC-preincubation (53 vs. 77% $^3$H incorporation in T + AgPC, Fig. 4b). An aPC concentration of 5 nM was sufficient for aPC's inhibitory effect (Supplementary Fig. 4). Preincubation of human AgPC cells with aPC (T + AgPC(aPC)) had no effect (Fig. 4b). Of note, aPC preincubation of human $T_{reg}$-depleted pan T-cells ((T-Tr)(aPC) + AgPC) partially reduced T-cell reactivity (83 vs. 100% $^3$H incorporation in (T-Tr) + AgPC, Fig. 4b). The latter indicates that aPC's inhibitory effect in the MLR is only partially dependent on pre-existing $T_{regs}$. Congruently with aPC's inhibitory effect on $^3$H incorporation, proliferation of aPC preincubated T-cells was reduced (eFluor450-labeling, 54.9% in aPC-preincubated T-cells vs. 67.7% in T-cells without aPC-preincubation, Fig. 4c). Assessment of annexin V and propidium iodide (PI) staining of T-cells demonstrated that preincubation of T-cells with aPC did not change T-cell apoptosis in the MLR (Supplementary Fig. 5).

Preincubation of human pan T-cells with aPC reduced the frequency of Th1 (CD4$^+$T-bet$^+$) and Th17 (CD4$^+$ROR-γt$^+$) cells, while increasing the abundance of $T_{regs}$ (CD4$^+$FOXP3$^+$) following

allogenic stimulation (Fig. 4d; Supplementary Fig. 6). Concomitantly, expression of IFN-γ, TNF-α, and IL-17A in T-cells was reduced, whereas expression of IL-10 in T-cells and TGFβ1 level in the supernatant were increased (Fig. 4e,f; Supplementary Fig. 7). Thus, as in mice with GvHD, preincubation of human T-cells with aPC prior to MLR generates a cytokine profile favouring $T_{reg}$ expansion and increases the frequency of $T_{regs}$.

**Amelioration of GvHD by aPC depends on $T_{regs}$.** To ascertain the functional relevance of aPC-mediated $T_{reg}$ expansion for amelioration of GvHD we transplanted $T_{regs}$ isolated from mice expressing the diphtheria-toxin receptor (DTR) under the control of the *FOXP3* locus (DEREG-mice), allowing selective depletion of $T_{regs}$[27]. BALB/c mice were irradiated and transplanted with BM ($5 \times 10^6$) and T-cells ($0.4 \times 10^6$) obtained from C57BL/6 mice and $T_{regs}$ ($0.1 \times 10^6$) from DEREG-mice or DTR-negative littermates (C57BL/6). Diphtheria toxin was injected on day 1 and 2 post transplantation in all groups (Supplementary Fig. 8). $T_{reg}$-depletion was confirmed by FACS analyses of splenocytes at day 14 (Supplementary Fig. 9). In mice receiving $T_{regs}$ not expressing the DTR aPC ameliorated GvHD as described above (B6T + B6$T_{reg}$(aPC) + BM + DT, Fig. 5a). Following

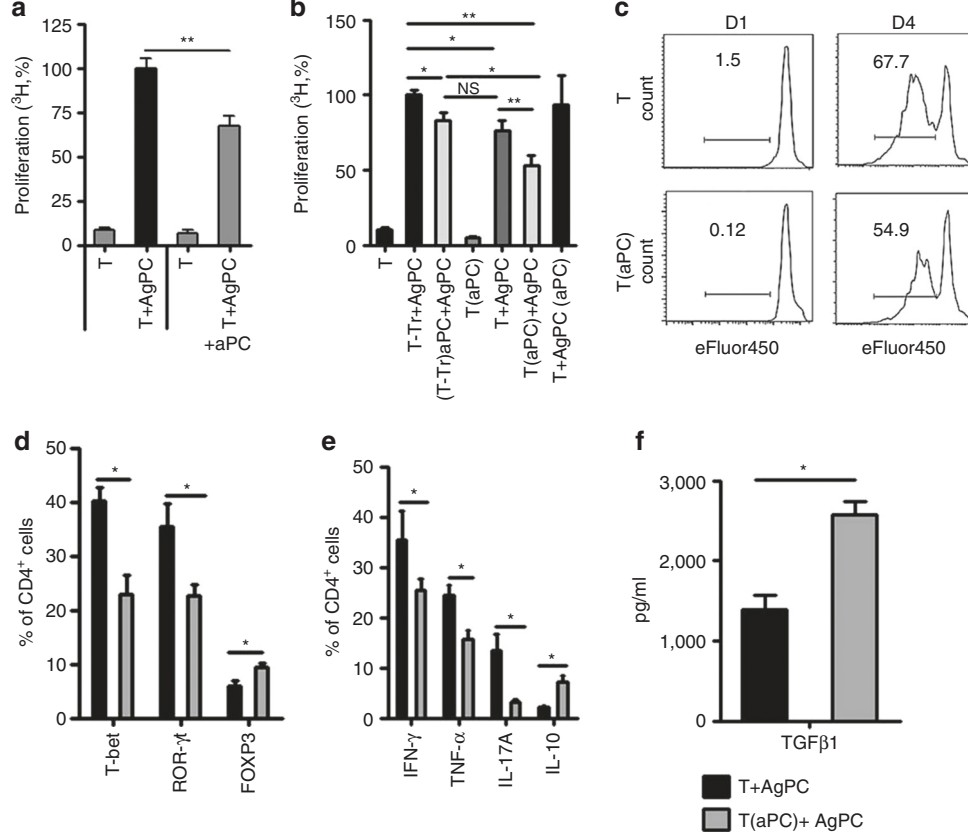

**Fig. 4** aPC promotes $T_{reg}$ expansion and inhibits T-cell reactivity in human T-cells. **a** Mixed lymphocyte reaction (*MLR*) of human peripheral blood T-cells (T) with allogenic antigen-presenting cells (*AgPC*) for 96 h. T-cell reactivity was measured by thymidine incorporation during the final 16 h. Concomitant incubation with aPC (+aPC; 20 nM, every 12 h) reduces T-cell reactivity. **b** aPC-preincubation (20 nM, 1 h, 37 °C) of human pan T-cells (T(aPC) + AgPC), but not of human AgPC (T + AgPC(aPC)) reduces T-cell reactivity in the MLR in comparison to $T_{reg}$-depleted T-cells (T-Tr + AgPC) and T-cells (T + AgPC). **c** Human pan T-cells without (T) or with (T(aPC)) aPC-preincubation were labelled with eFluor450, stimulated with AgPC, and proliferation was measured on day 1 (*D1*) and day 4 (*D4*) by FACS. **d**, **e** aPC-preincubation of human pan T-cells reduces the frequency of CD4$^+$ T-bet$^+$ and CD4$^+$ROR-γt$^+$ T-cells while increasing CD4$^+$FOXP3$^+$ $T_{regs}$ (**d**). Concomitantly, expression of the pro-inflammatory cytokines IFN-γ, TNF-α, and IL-17A, as measured by flow cytometry after 48 h of MLR, is reduced, while that of IL-10 is increased (**e**). **f** TGFβ1 levels in the supernatant of human T-cells MLR, measured after 96 h by ELISA. Results of at least three independent experiments (**a**, **b**, and **d**–**f**), each containing cells from three different donors, are shown; representative histogram plots **c** from triplicate wells of three independent experiments are shown; MFI and exemplary FACS images corresponding to Figs. 4d, e are shown in Supplementary Figs. 6 and 7; *$P < 0.05$, **$P < 0.01$ (**a**, **d**–**f**: *t*-test, **b** ANOVA)

depletion of $T_{regs}$, the protective effect of aPC was lost (B6T + B6-DTR-$T_{reg}$(aPC) + BM + DT), and these mice did not differ from mice, which received wt-$T_{regs}$ (B6T + B6T$_{reg}$ + BM + DT) or DTR-expressing $T_{regs}$ (B6T + B6-DTR-$T_{reg}$ + BM + DT) in the absence of aPC-preincubation (Fig. 5a). Thus, $T_{regs}$ are required for aPC's protective effect in GvHD following aPC-preincubation of T-cells.

**Preincubation of T-cells with aPC induces and expands $T_{regs}$.** To determine whether aPC induces $T_{reg}$ differentiation from $T_{reg}$-depleted T-cells or expands pre-existing $T_{regs}$, we isolated human $T_{regs}$ ($T_{reg}$), $T_{reg}$-depleted T-cells (T) and AgPC cells. Cells were combined without any aPC-preincubation (T + $T_{reg}$ + AgPC), using $T_{reg}$-depleted T-cells with aPC-preincubation together with $T_{regs}$ (T(aPC) + $T_{reg}$ + AgPC), or using $T_{regs}$ with aPC-preincubation together with $T_{reg}$-depleted T-cells (T + $T_{reg}$(aPC) + AgPC). In all three settings, $0.8 \times 10^5$ $T_{reg}$-depleted T-cells, $0.2 \times 10^5$ $T_{regs}$, and $3 \times 10^5$ AgPC were used. Following preincubation of pre-existing $T_{regs}$ with aPC (T + $T_{reg}$(aPC) + AgPC), a strong increase of $T_{reg}$ frequency (17.6% compared to T + $T_{reg}$ + AgPC, 5.5, and T(aPC) + $T_{reg}$ + AgPC, 10.5, respectively) was observed, suggesting that aPC-preincubation expands pre-existing $T_{regs}$. However,

preincubation of $T_{reg}$-depleted T-cells with aPC, likewise, increased the frequency of CD4$^+$FOXP3$^+$$T_{regs}$ (Fig. 5b), albeit to a lesser degree, reflecting induction of $T_{regs}$ from CD4$^+$ T-cells (i$T_{regs}$). The induction of $T_{regs}$ from $T_{reg}$-depleted T-cells is in agreement with the aPC-mediated induction of IL-10 and TGFβ1 and suppression of IL-6 observed in vivo (Figs. 3f,g), induction of IL-10 and TGFβ1 in vitro (Figs. 4e,f), and the partial suppressive effect observed following preincubation of $T_{reg}$-depleted T-cells with aPC in the MLR (Fig. 4b).

The predominant increase of $T_{regs}$ (CD4$^+$FOXP3$^+$) following aPC-preincubation of pre-existing $T_{regs}$ suggests that aPC reduces T-cell reactivity primarily by expanding pre-existing $T_{regs}$. However, given the induction of $T_{regs}$ from $T_{reg}$-depleted T-cells aPC may in addition reduce T-cell reactivity by inducing $T_{regs}$ from CD4$^+$CD25$^-$ effector T-cells. To specifically determine the functional relevance of these possibilities, we separately isolated effector T-cells ($T_{eff}$; CD4$^+$CD25$^-$) and $T_{regs}$ (CD4$^+$CD25$^+$) by MACS. MLR-experiments were conducted using effector T-cells ($T_{eff}$(aPC) + AgPC) or $T_{regs}$ ($T_{eff}$ + $T_{reg}$(aPC) + AgPC) preincubated with aPC, and the effect was compared to that observed in the absence of $T_{regs}$ ($T_{eff}$ + AgPC) or in the absence of aPC-preincubation of $T_{eff}$ cells ($T_{eff}$ + $T_{reg}$ + AgPC). In these experiments, $0.8 \times 10^5$ $T_{eff}$, $0.2 \times 10^5$ $T_{regs}$, and $3 \times 10^5$ AgPC

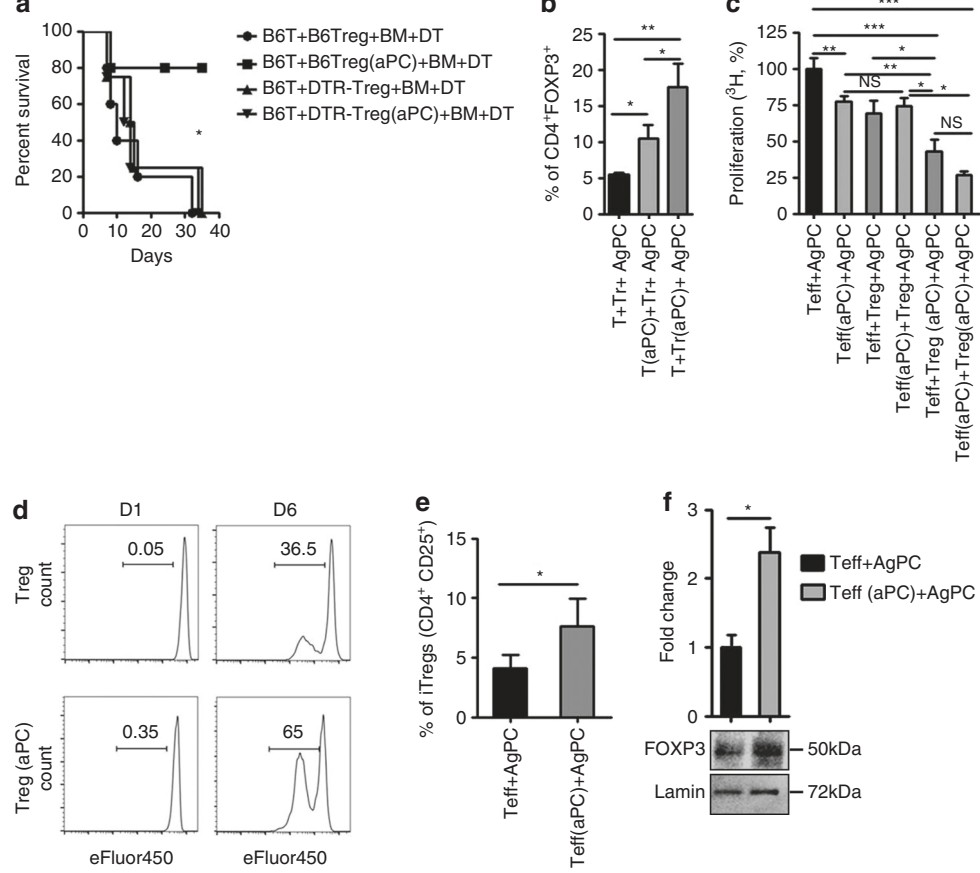

**Fig. 5** aPC restricts allogenic T-cell reactivity via regulatory T-cells. **a** BALB/c mice were irradiated and transplanted with BM ($5 \times 10^6$) and T-cells ($0.4 \times 10^6$) obtained from C57BL/6 mice and $T_{regs}$ ($0.1 \times 10^6$) from C57BL/6 (transgene negative DEREG littermate mice) without (B6T + B6T$_{reg}$ + BM + DT) or with (B6T + B6T$_{reg}$(aPC) + BM + DT) aPC-preincubation or from DEREG mice without (B6T + DTR-T$_{reg}$ + BM + DT) or with aPC-preincubation (B6T + DTR-T$_{reg}$(aPC) + BM + DT). Diphtheria toxin (*DT*, 20 ng/g body weight) was injected on day 1 and 2 post transplantation in all groups. **b** aPC-preincubation of pre-existing human $T_{regs}$ with aPC (T + Tr(aPC) + AgPC) increases $T_{reg}$ frequency to a larger extent than aPC-preincubation of $T_{reg}$-depleted T-cells (T(aPC) + Tr + AgPC), $T_{reg}$ frequency was measured by flow cytometry after 48 h of MLR. **c** Preincubation of human $T_{regs}$ with aPC ($T_{eff}$ + $T_{reg}$(aPC) + AgPC) markedly reduces T-cell reactivity, as determined by thymidine incorporation in the MLR. In the absence of $T_{regs}$ aPC-preincubation of effector T-cells ($T_{eff}$(aPC) + AgPC) is less efficient, but the effect is comparable to that observed when combing effector T-cells with $T_{regs}$ without aPC-preincubation ($T_{eff}$ + $T_{reg}$ + AgPC). Results are compared with T-cell reactivity of effector T-cells in the absence of $T_{regs}$ ($T_{eff}$ + AgPC). **d** aPC induces proliferation of human $T_{regs}$: $T_{regs}$ without ($T_{reg}$) or with aPC-preincubation ($T_{reg}$(aPC)) were labelled with eFluor450 (4 µM) and stimulated with plate bound αCD3 (10 µg/ml) and αCD28 (8 µg/ml) antibodies. Proliferation was measured on day 1 and 6 by FACS. **e** aPC induces human CD4$^+$CD25$^+$ cells: effector T-cells (CD4$^+$CD25$^-$) were left untreated or were preincubated with aPC (20 nM, 1 h, 37 °C). Following allogenic simulation with AgPC the frequency of CD4$^+$CD25$^+$ cells, determined at day 4, was increased following aPC-preincubation; results from three repeat experiments each with three biological disjunct donors. **f** Following preincubation of $T_{eff}$-cells with aPC nuclear levels of FOXP3 are increased. Representative immunoblot (*bottom*) and bar graph (*top*) summarizing results of five independent repeat experiments. *T* $T_{reg}$-depleted T-cells, *T(aPC)* $T_{reg}$-depleted T-cells preincubated with aPC (20 nM, 1 h, 37 °C), *AgPC* allogenic antigen-presenting cells, $T_{eff}$ effector T-cells; $T_{reg}$ regulatory T-cells. Exemplary FACS images corresponding to Fig. 5e are shown in Supplementary Fig. 10. Mean value ± SEM (**b**, **c**, **e**, and **f**); *$P < 0.05$, **$P < 0.01$ (**a** log-rank test, **b**, **c** ANOVA; **e**, **f** t-test)

were combined. Preincubation of $T_{regs}$ with aPC resulted in a strong suppression of T-cell reactivity compared to control ($T_{eff}$ + AgPC, 100% vs. $T_{eff}$ + $T_{reg}$(aPC) + AgPC, 42%, Fig. 5c). This effect was significantly stronger than that observed when using $T_{regs}$ not preincubated with aPC ($T_{eff}$ + $T_{reg}$ + AgPC, 69%, vs. $T_{eff}$ + $T_{reg}$(aPC) + AgPC, 42%). Of note, preincubation of T-effector cells with aPC ($T_{eff}$(aPC) + AgPC) likewise reduced T-cell reactivity to a comparable level as observed when combining effector T-cells with $T_{regs}$ without aPC-preincubation ($T_{eff}$(aPC) + AgPC 77% vs. $T_{eff}$ + $T_{reg}$ + AgPC, 69%, Fig. 5c). The suppressive effect observed following aPC-preincubation of effector T-cells was significantly less than that observed following aPC-preincubation of $T_{regs}$ ($T_{eff}$(aPC) + AgPC 77% vs. $T_{eff}$ + $T_{reg}$(aPC) + AgPC, 42%). Preincubation of both $T_{eff}$ and $T_{reg}$ separately with aPC ($T_{eff}$(aPC) + $T_{reg}$(aPC) + AgPC) suppressed T-cell reactivity

more than preincubation of either $T_{eff}$ or $T_{reg}$ only, but this effect was not significantly different from that observed following aPC-preincubation of $T_{regs}$ only (42 vs. 27%; Fig. 5c).

To ascertain increased proliferation of pre-existing $T_{regs}$ following aPC-preincubation and stimulation, we isolated human $T_{regs}$ by FACS. $T_{regs}$ without ($T_{reg}$) or with ($T_{reg}$(aPC)) aPC-preincubation were stimulated with plate-bound αCD3 (10 µg/ml) and αCD28 (8 µg/ml) antibodies. The proliferation of stimulated aPC-preincubated $T_{regs}$ ($T_{reg}$(aPC)) was almost twice as high as that of control $T_{regs}$ (65.0% in $T_{reg}$(aPC) vs. 36.5% in $T_{regs}$, Fig. 5d).

As aPC-preincubation of effector T-cells reduced T-cell proliferation, we next ascertained whether aPC-preincubation induced $T_{regs}$ from effector T-cells. To this end, we determined the frequency of CD4$^+$CD25$^+$ cells following co-incubation

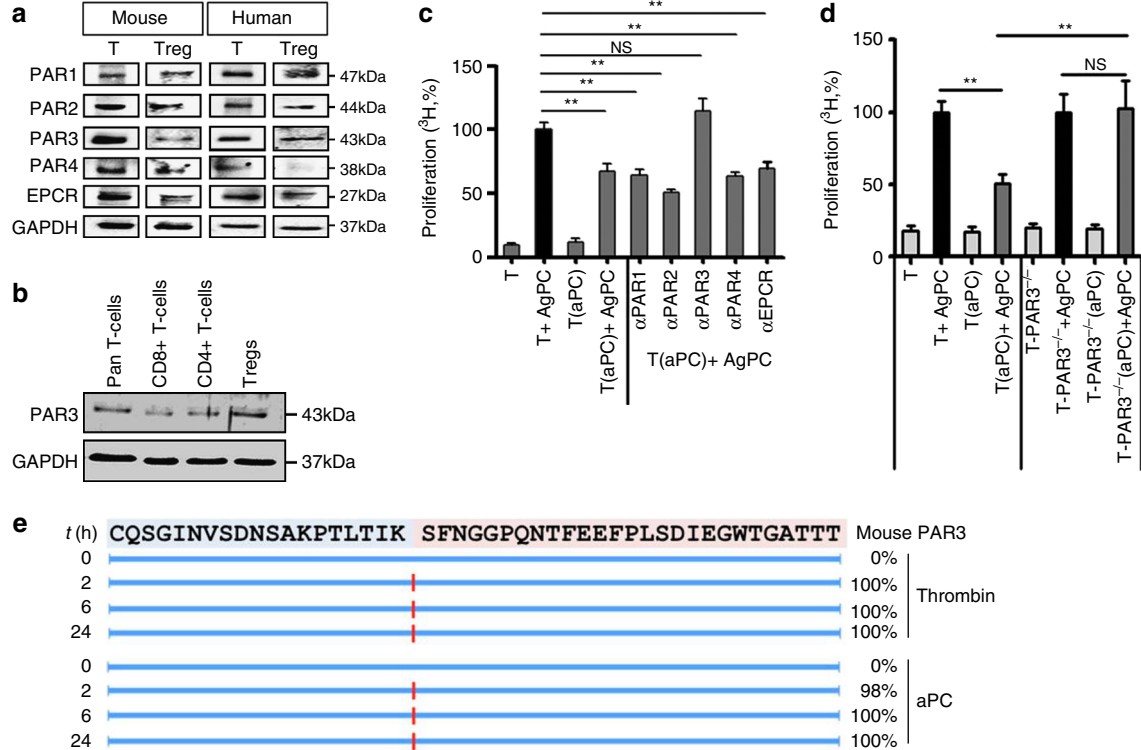

**Fig. 6** aPC inhibits allogenic T-cell activation via PAR3. **a** Expression of PARs and EPCR on human and mouse pan T-cells (T) and regulatory T-cells (T$_{reg}$) was determined by immunoblotting. GAPDH was used as loading control. Exemplary immunoblots of three independent repeat experiments. **b** Expression of PAR3 in human pan T-cells, CD8$^+$ T-cells, CD4$^+$ T-cells, and T$_{regs}$ (CD4$^+$CD25$^+$). Exemplary immunoblots of three independent repeat experiments. **c** Preincubation of human peripheral blood T-cells (T) with N-terminal binding antibodies against PAR3 (αPAR3), but not against PAR1 (αPAR1), PAR2 (αPAR2), or PAR4 (αPAR4), or with an EPCR blocking antibody (αEPCR) prior to aPC-preincubation and MLR abrogates aPC's inhibitory effect in regard to T-cell reactivity. **d** Preincubation of T-cells (T) from C57BL/6 PAR3-deficient (PAR3$^{-/-}$) mice with aPC and subsequent allogenic stimulation (T-PAR3$^{-/-}$(aPC) + AgPC) abrogates aPC's inhibitory effect on T-cell reactivity. **e** Proteolytic fragments (indicated by *red lines*) observed following incubation of the N-terminal end of mouse PAR3 (sequence shown at the *top*) with human thrombin (10 nM) or human aPC (500 nM). Proteolysis was determined at indicated time-points (*left*, t(h)) and estimated cleavage efficiency is shown in percentage (*right*). Mean value ± SEM (**c**, **d**), results of at least three repeat experiments each with three biological disjunct donors. **P < 0.01 (**c**, **d**: ANOVA)

of effector T-cells with AgPC without (T$_{eff}$ + AgPC) or with (T$_{eff}$(aPC) + AgPC) aPC-preincubation. After 4 days, the frequency of CD4$^+$CD25$^+$ cells was increased following preincubation of effector T-cells with aPC (T$_{eff}$ + AgPC, 4.1% vs. T$_{eff}$(aPC) + AgPC, 7.6%, Fig. 5e; Supplementary Fig. 10). In parallel, we observed a strong induction of nuclear FOXP3 in T$_{eff}$s following preincubation with aPC (Fig. 5f). Altogether, these data suggest that aPC expanses T$_{regs}$ by both expanding pre-existing T$_{regs}$ and inducing T$_{reg}$ differentiation.

**T-cell inhibition by aPC depends on PAR2-PAR3 cofactoring.** The above observations demonstrate that aPC promotes T$_{reg}$ expansion and inhibits T-cell reactivity, but the receptors involved remain unknown. Jurkat T-cells express PARs, which are the pivotal receptors for aPC-dependent signaling[17, 28, 29]. Similarly, primary human and mouse pan T-cells and mouse T$_{regs}$ express all four PARs as well as EPCR (endothelial protein C receptor), whereas human T$_{regs}$ express EPCR and all PARs except PAR4 (Fig. 6a)[28]. Intriguingly, in human T$_{regs}$ PAR3 expression appears to be higher as compared to CD8$^+$ or CD4$^+$ T-cells (Fig. 6b). To ascertain which PAR is required for aPC's effect in the MLR, we first used inhibitory antibodies. These antibodies inhibit proteolytic cleavage of the N-terminal receptor sequence and thus the generation of the corresponding tethered ligand. Inhibition of PAR3 cleavage abolished aPC's effect, whereas N-terminal blocking antibodies to PAR1, PAR2, or PAR4

had no effect (Fig. 6c). The requirement of PAR3 for aPC's inhibitory effect on T-cell activation was confirmed using T-cells isolated from PAR3-deficient mice (Fig. 6d).

Burnier and Mosnier[30] previously established generation of a human PAR3-dependent tethered ligand by aPC. We previously demonstrated that aPC cleaves the N-terminal end of mouse PAR3, but the cleavage site remained unknown[31]. To analyse aPC cleavage of mouse PAR3, we followed established protocols[30]. Using the human PAR3-derived N-terminal peptide, we first validated the approach by replicating previous results[30] (Supplementary Fig. 11). Cleavage analyses of the mouse PAR3-derived N-terminal peptide revealed that both human thrombin and human aPC efficiently cleave at Lys37 within 2 h (Fig. 6e). This result is congruent with our previous mutagenesis studies[31] and establishes the aPC cleavage site within the mouse PAR3-derived N terminus.

As PAR3 is not signaling competent itself this finding indicates the involvement of a co-receptor[18]. EPCR is typically, but not in all cell-types or under all conditions, required for aPC signaling[17, 31, 32]. An antibody blocking aPC binding to EPCR did not abolish aPC's inhibitory effect on T-cell activation (Fig. 6c), suggesting a signaling mechanism of aPC on T-cells independent of EPCR. Of note, the tethered ligand of PAR3 is capable to activate PAR1 or PAR2 ("cofactoring"), presumably by binding to the second extracellular loop of these receptors[18, 19]. Inhibitory peptides against PAR1 and PAR4 (blocking binding of a ligand to the second extracellular loop) failed to abolish aPC's effect in the

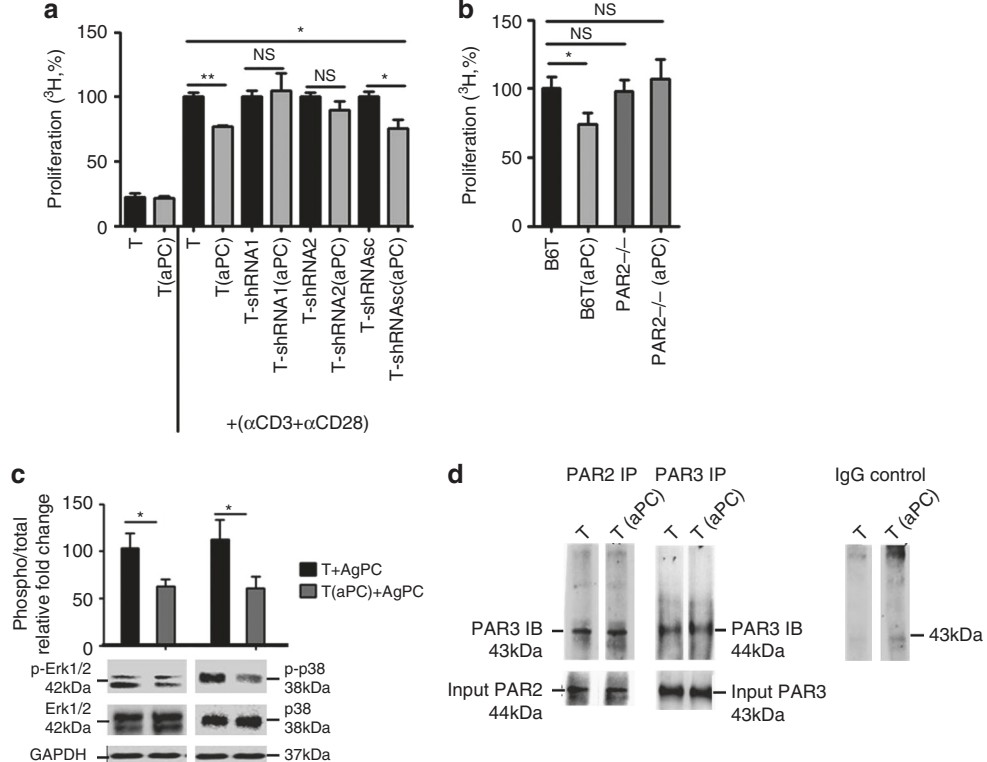

**Fig. 7** aPC signaling requires PAR2 in addition to PAR3 on T-cells. **a** Following knockdown of PAR2 in human primary pan T-cells (T) using two different shRNAs (shRNA1 and shRNA2) the inhibitory effect of aPC on T-cells after stimulation with αCD3 and αCD28 antibodies is lost. Scrambled shRNA (shRNAsc) had no effect. Bar graph summarizing results from five repeat experiments using five biological disjunct donors, each in triplicates. **b** Preincubation of T-cells from C57BL/6 wild type (B6T(aPC)) mice, but not of C57BL/6 PAR2-deficient (PAR2$^{-/-}$(aPC)) mice, with aPC and subsequent stimulation with plate-bound αCD3 and αCD28 antibodies, abrogates aPC's inhibitory effect on T-cell reactivity. **c** Preincubation of human pan T-cells with aPC reduces ERK1/2 and p38 activity after 96 h MLR as reflected by reduced phosphorylation of these proteins (normalized to the corresponding total protein levels) as determined by immunoblot; bar graph reflecting mean results ± SEM (*top*) and representative immunoblots (*bottom*) are shown. **d** Exemplary immunoblot showing interaction of PAR2 and PAR3 in human T$_{regs}$ analysed using immunoprecipitation for PAR2 followed by immunoblotting of PAR3 and vice versa. Immunoprecipitation using IgG was used as negative control. Immunoblot of PAR2 and PAR3 was done as a input control. Mean value ± SEM (**a**, **b**), results of at least three repeat experiments each with three biological disjunct donors (**a**, **c**) or from five different mice (**b**). *$P < 0.05$, **$P < 0.01$, NS non significant (**a**, **b** ANOVA; **c** *t*-test)

MLR (Supplementary Fig. 12). Considering the lack of specific PAR2-inhibitory peptides[33], we knocked down PAR2 expression in primary human T-cells. Efficient shRNA-mediated knockdown of PAR2 was achieved in pan T-cells using transient transfection with lentiviral particles, whereas a scrambled control shRNA had no effect (Supplementary Fig. 13). PAR2 knockdown in human primary T-cells abolished the inhibitory effect of aPC following stimulation with αCD3 and αCD28 (Fig. 7a) antibodies. Furthermore, the requirement of PAR2 for aPC's inhibitory effect on T-cell activation was confirmed using T-cells isolated from PAR2-deficient mice (Fig. 7b).

Signaling of aPC via PARs involves the MAPK pathway, which is also known to modulate T-cell function and T$_{reg}$ differentiation[17, 19, 34]. Analyses of T-cells preincubated with aPC and isolated by MACS after 96 h of MLR revealed reduced Erk1/2 and p38 phosphorylation (Fig. 7c), which is consistent with aPC signaling via PAR2 in T-cells. Co-immunoprecipitation of PAR2 or PAR3 confirmed the interaction of PAR2/PAR3 heterodimers on human T$_{regs}$ (Fig. 7d). Taken together, these data indicate the requirement of a PAR2/ PAR3 heterodimer for aPC's inhibitory effect in allogenic activation of human and mouse T-cells.

**Amelioration of GvHD by aPC requires both PAR2 and PAR3.**
To assess whether PAR2 and PAR3 are required for

aPC's ameliorating effect on GvHD, we transplanted lethally irradiated BALB/c mice with allogenic (C57BL/6) $5 \times 10^6$ BM and $0.5 \times 10^6$ C57BL/6 wild-type (B6T), C57BL/6 PAR2$^{-/-}$ (PAR2$^{-/-}$T), and C57BL/6 PAR3$^{-/-}$ (PAR3$^{-/-}$T) T-cells. Wild-type (B6T + BM) or receptor-deficient T-cells without (PAR2$^{-/-}$T + BM; PAR3$^{-/-}$T + BM) or with (B6T(aPC) + BM; PAR2$^{-/-}$T(aPC) + BM; PAR3$^{-/-}$T(aPC) + BM) aPC preincubation were used. When using PAR2$^{-/-}$ or PAR3$^{-/-}$ pan T-cells the protective effect of aPC was lost (Figs. 8a,b), corroborating the above in vitro results (Figs. 6 and 7).

To ascertain the function of PAR2 and PAR3 specifically on T$_{regs}$, we separately isolated T$_{regs}$ and T-cells (B6T) from C57BL/6 wt, PAR2$^{-/-}$, or PAR3$^{-/-}$ mice, and transplanted these together with C57BL/6-derived BM into irradiated-recipient BALB/c mice. Although preincubation of wt T$_{regs}$ with aPC (B6T + T$_{reg}$(aPC) + BM) ameliorated GvHD as compared to T$_{regs}$ without aPC-preincubation (B6T + Treg + BM), this protective effect was lost when by using PAR2-deficient or PAR3-deficient T$_{regs}$ preincubated with aPC (PAR2$^{-/-}$T + PAR2$^{-/-}$T$_{reg}$(aPC) + BM; PAR3$^{-/-}$T + PAR3$^{-/-}$T$_{reg}$(aPC) + BM, Figs. 8c,d). Thus, loss of PAR2 or PAR3 specifically on T$_{regs}$ is sufficient to abolish the protective effect of aPC in GvHD.

**aPC limits GvHD in a humanized model of GvHD in mice.** The above data suggest that ex vivo preincubation of pan T-cells

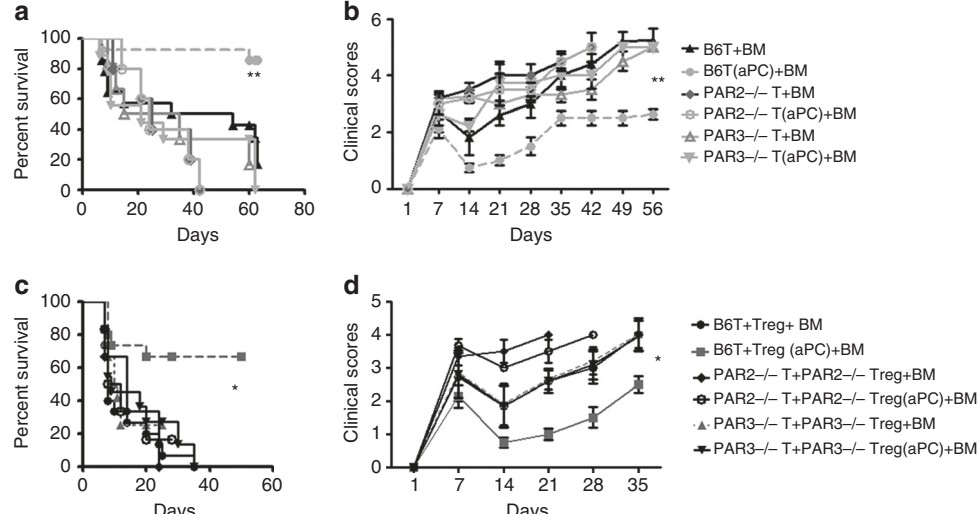

**Fig. 8** PAR2-PAR3 cofactoring on T-cells is required for inhibition of GvHD by aPC. **a**, **b** Lethally irradiated (11 Gy)-recipient BALB/c mice were transplanted with C57BL/6 wt-derived $5 \times 10^6$ BM cells and $0.5 \times 10^6$ T-cells without (B6T + BM) or with (B6T(aPC) + BM) aPC-preincubation (20 nM, 1 h, 37 °C). Alternatively, lethally irradiated BALB/c mice received $5 \times 10^6$ C57BL/6 wt-derived BM cells and $0.5 \times 10^6$ C57BL/6 PAR3$^{-/-}$ T-cells (PAR3$^{-/-}$T + BM) or $0.5 \times 10^6$ C57BL/6 PAR2$^{-/-}$ T-cells (PAR2$^{-/-}$T + BM) or $5 \times 10^6$ C57BL/6 wt-derived BM cells and $0.5 \times 10^6$ C57BL/6 PAR3$^{-/-}$ T-cells (PAR3$^{-/-}$T(aPC) + BM) or $0.5 \times 10^6$ C57BL/6 PAR2$^{-/-}$ T-cells (PAR2$^{-/-}$T(aPC) + BM; aPC preincubation in both cases: 20 nM, 1 h, 37 °C). aPC's protective effect is lost in mice transplanted with PAR2-deficient or PAR3-deficient T-cells; survival (**a** Kaplan–Meier curve) and clinical score (**b**) from two (PAR2$^{-/-}$T + BM and PAR2$^{-/-}$T(aPC) + BM) or three (all other groups) independent experiments each with four mice per group are shown. **c**, **d** Lethally irradiated (11 Gy)-recipient BALB/c mice were transplanted with C57BL/6-derived $5 \times 10^6$ BM cells and the following combination of C57BL/6-derived T-cells: (i) $0.4 \times 10^6$ T-cells and $0.1 \times 10^6$ T$_{regs}$ (B6T + T$_{reg}$ + BM), (ii) $0.4 \times 10^6$ T-cells and $0.1 \times 10^6$ T$_{regs}$ with aPC-preincubation (B6T + T$_{reg}$(aPC) + BM), (iii) $0.4 \times 10^6$ PAR3$^{-/-}$T-cells and $0.1 \times 10^6$ PAR3$^{-/-}$ T$_{regs}$ (PAR3$^{-/-}$T + PAR3$^{-/-}$T$_{reg}$ + BM), (iv) $0.4 \times 10^6$ PAR3$^{-/-}$T-cells and $0.1 \times 10^6$ PAR3$^{-/-}$ T$_{regs}$ with aPC-preincubation (PAR3$^{-/-}$T + PAR3$^{-/-}$T$_{reg}$(aPC) + BM), (v) $0.4 \times 10^6$ PAR2$^{-/-}$T-cells and $0.1 \times 10^6$ PAR2$^{-/-}$ T$_{regs}$ (PAR2$^{-/-}$T + PAR2$^{-/-}$T$_{reg}$ + BM), or (vi) with $0.4 \times 10^6$ PAR2$^{-/-}$T-cells and $0.1 \times 10^6$ PAR2$^{-/-}$ T$_{regs}$ with aPC-preincubation (PAR2$^{-/-}$T + PAR2$^{-/-}$T$_{reg}$(aPC) + BM). aPC-preincubation was conducted as before (20 nM, 1 h, 37 °C). Loss of PAR2 or PAR3 on T$_{regs}$ abrogates the GvHD protective effect of aPC. Survival (**c** Kaplan–Meier curve) clinical score (**d**) from two (PAR2$^{-/-}$T + BM and PAR2$^{-/-}$T(aPC) + BM) or three (all other groups) independent experiments, each with four mice per group, are shown. *$P < 0.05$, **$P < 0.01$, (**a**, **c** log-rank test; **b**, **d** ANOVA)

or T$_{regs}$ prior to transplantation may be an easy, efficient, and safe new therapeutic strategy to mitigate GvHD. To corroborate the translational relevance, we transplanted NSG-Ab°DR4 mice, which lack expression of the murine *Prkdc* gene, the X-linked *Il2rg* gene, and MHC class II while expressing the human leukocyte antigen DR4 gene[35], with $4 \times 10^6$ human CD4$^+$ T-cells (HLA-DR4$^-$) without (hCD4$^+$) or with (hCD4$^+$(aPC)) aPC-preincubation. Survival, physical appearance, and histological damage of NSG-Ab° DR4 mice transplanted with aPC-preincubated human CD4$^+$ T-cells (hCD4$^+$(aPC)) was markedly improved compared to control (hCD4$^+$) mice (Fig. 9).

**aPC ameliorates GvHD without impeding the GvL effect**. Finally, we evaluated aPC's impact on the graft-vs.-leukaemia (GvL) effect using MLL-AF9 (MA9) as one of the best-characterized oncogenes in mouse models of acute myeloid leukaemia (AML). We transplanted lethally irradiated BALB/c mice with syngenic $5 \times 10^3$ GFP positive MA9 transformed leukaemic cells (BALB/c background)[36], $5 \times 10^6$ allogenic BM cells, and $0.5 \times 10^6$ allogenic T-cells without (B6T + BM + MA9) or with (B6T(aPC) + BM + MA9) aPC-preincubation. These groups were compared to BALB/c mice receiving only allogenic BM and T-cells (B6T + BM) or receiving only allogenic BM and MA9 (BM + MA9) leukaemic cells.

Mice receiving BM and T-cells only (BM + B6T) displayed increased lethality (Fig. 10a) and developed typical hallmarks of GvHD; and GvHD was confirmed by histology (e.g., increased frequency of cryptic apoptosis in the gastrointestinal tract). Contrary, mice receiving only BM and MA9 cells (BM+MA9) lacked morphological and histological signs of GvHD, but these

mice nevertheless died as early as BM + B6T mice (Fig. 10a), presumably due to leukaemia progression as indicated by a slightly increased leukaemic burden after 1 week (Fig. 10b). The tumour load, determined 1 week post transplantation, did not differ between mice receiving BM, MA9 cells, and T-cells without (B6T + BM + MA9) or with (B6T(aPC) + BM + MA9) aPC-preincubation (Fig. 10b), indicating comparable tumour engraftment in these mice. Yet, peripheral leukaemic load determined 4 weeks post transplantation was markedly reduced in B6T(aPC) + BM + MA9 mice, but not in B6T + BM + MA9 mice, (Fig. 10c; Supplementary Fig. 14), indicating a sustained GvL effect in B6T(aPC) + BM + MA9 mice. Importantly, mice receiving T-cells preincubated with aPC (B6T(aPC) + BM + MA9) did not display signs of GvHD (e.g., increased frequency of cryptic apoptosis in the gastrointestinal tract). Accordingly, survival of mice receiving T-cells preincubated with aPC was significantly improved compared to all other groups (Fig. 10a).

To corroborate that the GvL effect is sustained following preincubation of T-cells with aPC, we generated an independent leukaemia model using the clinically relevant cooperating oncogenes AML1-ETO and KRAS (AE9aKRAS). BM of primary recipient mice was harvested upon onset of clinical symptoms and a total number of $2 \times 10^4$ GFP$^+$ cells were injected into lethally irradiated C57BL/6 mice. Leukaemic cells were injected along with $5 \times 10^6$ supporter BM cells and $1 \times 10^6$ T-cells (with and without aPC preincubation) derived from C3H/HeNRj mice. Again, aPC preincubation of T-cells ameliorated GvHD while maintaining a GvL effect (Supplementary Fig. 15). Hence, ex vivo preincubation of T-cells with aPC ameliorates GvHD without compromising the GvL effect.

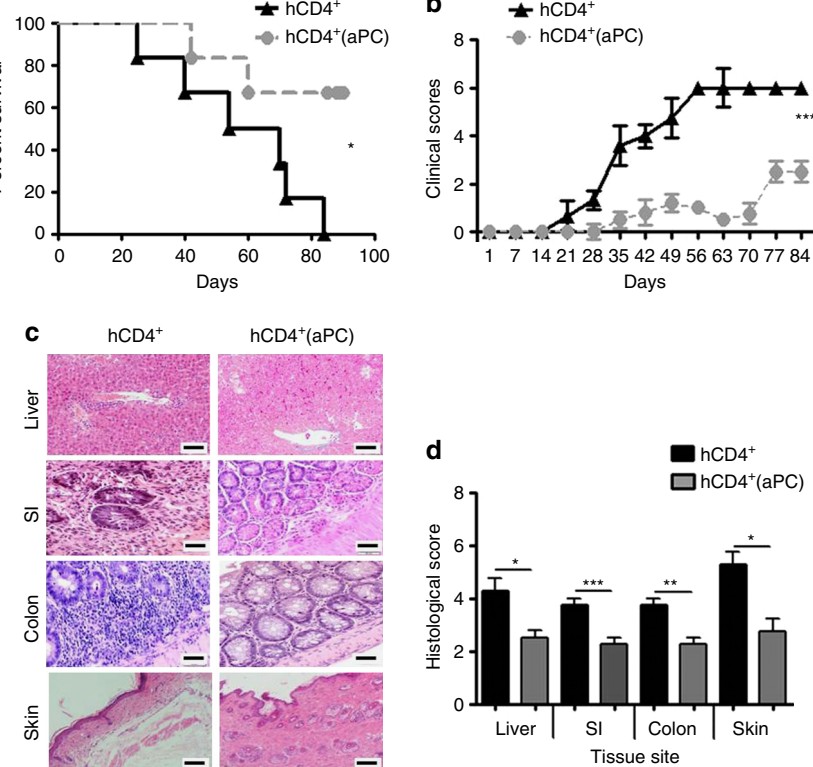

**Fig. 9** aPC mitigates human CD4[+] T-cell mediated GvHD. **a, b** Recipient NSG-Ab DR4 mice were irradiated with 2 Gy and transplanted 4 h later with 4 × 10[6] of human CD4[+] T-cells (HLA-DR4[−]) without (hCD4[+]) or with (hCD4[+](aPC)) aPC-preincubation. Survival (**a** Kaplan–Meier curve) and physical parameters (**b** clinical score) are shown; pooled data from three independent experiments each with two recipients. **c, d** Photomicrographs depicting typical morphology in liver, small intestine (*SI*), colon, and skin; haematoxylin and eosin stained section (**c**), size bar 50 μm and bar graph summarizing histological disease scores; mean value ± SEM (five mice per group, **d**). *$P < 0.05$, **$P < 0.01$; ***$P < 0.001$ (**a** log-rank test; **b** ANOVA; **d** t-test)

## Discussion

Here we uncover a new pathway targeting $T_{regs}$ and ameliorating GvHD. Although a crucial role of $T_{regs}$ to protect against GvHD is established, methods to easily and efficiently enrich $T_{regs}$ have been lacking so far[8, 9]. The current data demonstrate that aPC-preincubation of T-cells increases $T_{reg}$ frequency through expansion of pre-existing $T_{regs}$ and induction of $T_{regs}$. The latter observation is supported by a cytokine profile promoting $T_{reg}$ induction (increase TGFβ1 and IL-10, but reduced IL-6 levels). Although expansion of $T_{regs}$ following ex vivo preincubation of pan T-cells with aPC protects from GvHD, it does not compromise the GvL effect when using two independent tumor cell models. A sustained GvL effect despite suppression of GvHD has been previously reported following the adoptive transfer of $T_{regs}$ in animal and clinical studies[6, 7]. Hence, our observation is congruent with these earlier reports, but by proposing an easy, safe, and efficient way to expand donor-derived $T_{regs}$, the current study identifies an approach of potential translational relevance.

The sustained GvL effect following preincubation of T-cells with aPC, and subsequent expansion of $T_{regs}$ is congruent with previous studies demonstrating that $T_{regs}$ do not only prevent GvHD, but simultaneously convey an efficient GvL effect[6, 7]. Several mechanisms have been proposed for the dual effect of $T_{regs}$ in regard to GvHD and the GvL effect, e.g., inhibition of JAK1/JAK2 signaling, expression of NKG2D by CD8[+] cells, IL-21 signaling, or the differential expression of granzyme B by CD4[+]CD25[+] and CD8[+] cells[37–40]. Whether these mechanisms are regulated by aPC remain unknown. Likewise, the relevance of other T-cell populations, such as Th1, Th17, or CD8[+] T-cells for aPC's effects as observed in the current study remains to be evaluated.

The expansion of $T_{regs}$ by aPC-PAR2/PAR3 signaling and the efficacy of aPC to improve GvHD following ex vivo preincubation of T-cells identify a new function of aPC-signaling and PAR-signaling in T-cells. This finding adds to the previous reports suggesting a role of PARs in AgPC[41]. An effect of aPC on $T_{reg}$ expansion was previously reported by Xue et al.[21] in NOD mice in the context of pancreatic islet inflammation. However, the underlying mechanisms, e.g., the immune cells targeted by aPC and the receptors required for aPC's effect, remained unknown. Likewise, expression of PARs on T-cells has been reported before, but these studies evaluated Jurkat (immortalized human T lymphocyte cells) cells and hence the physiological relevance of PARs in adaptive immunity remained unknown[17, 28]. To decipher the interaction of PAR2 and PAR3 on T-cells, we first used inhibitory antibodies, which prevent proteolytic cleavage of the N-terminal end of PARs. Only the antibody targeting PAR3 abolished aPC's effect on allogenic T-cell stimulation. Proteolytic cleavage of human and mouse PAR3 by aPC has been previously reported by us and others[30, 31], and we now identify the aPC cleavage site within the mouse PAR3 N-terminal peptide.

PAR3 is generally not considered to be signaling competent itself and requires a co-receptor[19]. Analyses of murine PAR2[−/−] T-cells and human PAR2 knockdown T-cells revealed that—in addition to PAR3—PAR2 is required for aPC's inhibitory effect on allogeneic T-cell activation. Furthermore, the functional relevance of PAR2 and PAR3 on $T_{regs}$ was confirmed in vivo, as aPC's protective effect was lost following preincubation of $T_{regs}$ lacking either PAR3 or PAR2. Signaling through heterodimeric PAR-complexes has not been reported in the context of adaptive immunity hitherto[19].

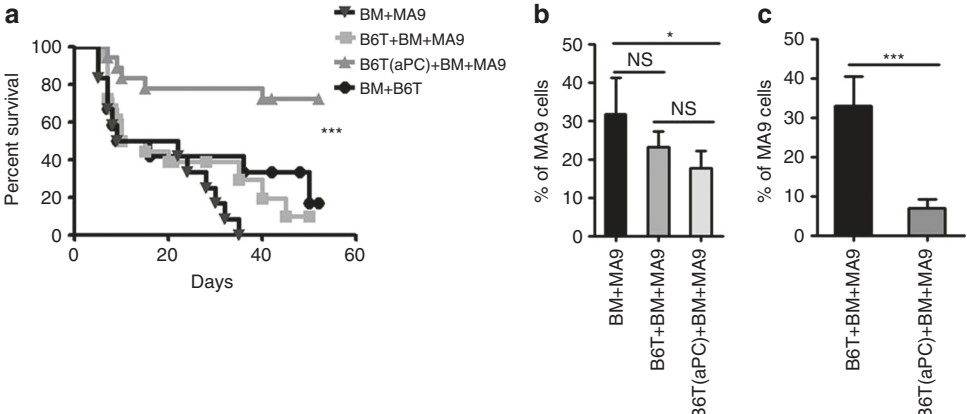

**Fig. 10** aPC-preincubation of mouse pan T-cells does not impair GvL effect. **a, b** Lethally irradiated (11 Gy)-recipient BALB/c mice were transplanted with $5 \times 10^6$ C57BL/6-derived BM cells, $5 \times 10^3$ GFP$^+$MLL-AF9 (MA9) leukaemic cells, and $0.5 \times 10^6$ C57BL/6 T-cells without (B6T + BM + MA9) or with (B6T(aPC) + BM + MA9) aPC-preincubation. These groups were compared with the BALB/c mice receiving only allogenic BM and T-cells (B6T + BM) or receiving only allogenic BM and MA9 leukaemic cells (BM + MA9). In mice transplanted with aPC-preincubated T-cells and MA9 leukaemic cells (B6T(aPC) + BM + MA9) survival is improved (**a** Kaplan–Meier curve) and leukaemic load (determined by flow cytometry 1 week (**b**) and 4 weeks (**c**) after transplantation in peripheral blood samples) is markedly decreased; pooled data from three independent experiments each with four recipients. Mean value ± SEM; corresponding exemplary FACS images for Fig. 10b, c are shown in Supplementary Fig. 14; *P < 0.05; ***P < 0.001, NS non significant (**a** log-rank test; **b, c** t-test)

We acknowledge that the exact mechanism of action through which PAR2 and PAR3 interact remains currently unresolved. However, cofactoring of PARs is established and activation of other PARs, including PAR2, by the PAR3-derived tethered ligand has already been proposed[19, 29]. Intriguingly, although aPC signals via species-specific PAR-heterodimers in human and mouse podocytes (human: PAR3/PAR2; mouse: PAR3/PAR1), the same PAR-heterodimer (PAR3/PAR2) is required for aPC's effect on mouse and human T-cells and mouse T$_{regs}$. This observation emphasizes the plasticity of PAR-signaling, both in regard to different cell-types and species.

The current results together with our previous studies[31] establish that aPC cleaves the N-terminal end of PAR3, thus generating a tethered ligand, which may interact with other PARs. As we used human aPC throughout our study, the finding that human aPC cleaves mouse PAR3 is relevant in the context of the current study. Future studies need to address whether likewise mouse aPC cleaves mouse PAR3. Intriguingly, thrombin and aPC cleaved mouse PAR3 at the same residue site (Lys37), contrasting the observations made with the human PAR3-derived peptide[30] (and current study). Signaling specificity of thrombin and aPC via mouse PAR3 can hence not be explained by different tethered ligands generated by either protease. Biased signaling independent of the cleavage site, but rather secondary to involvement of specific receptor and signaling complexes, has been recently proposed for aPC and thrombin-dependent β-arrestin-2 signaling via PAR1[42]. Accordingly, we propose that specificity of PAR3 signaling is not only dependent on specific PAR3 cleavage sites, but also on a specific PAR3-containing protein complex.

In addition to PARs, other receptors for aPC have been reported[17], and we can currently not exclude the involvement of other components within the aPC-receptorsome on T-cells. Of note, the current data suggest that EPCR is not required for aPC-mediated inhibition of allogenic T-cell activation. Signaling of aPC independent of EPCR has been reported in endothelial cells and in particular in non-endothelial cells[31, 32]. Intriguingly, aPC increases Akt phosphorylation in human leukaemic monoblast U937 cells independent of EPCR and modulates geneexpression in dendritic cells partially independent of EPCR[43], supporting the notion that aPC signaling in immune cells does not strictly depend on EPCR.

Previous studies demonstrated that the efficacy of aPC in MLR is enhanced in the presence of protein S[20]. Protein S functions as a non-proteolytic cofactor of protein C in the context of coagulation inhibition. In addition, protein S is expressed by T-cells and engages TAM receptors (Axl and Mertk) on dendritic cells to restrict the immune response[44]. Whether protein C and protein S co-ordinately modulate T-cell activation remains to be shown.

The protective role of T$_{regs}$ in GvHD is well documented in animal models and has led to several clinical studies[9, 45]. For example, ex vivo umbilical cord blood-derived T$_{regs}$ reduced the risk of acute and chronic GvHD compared to identically treated historical controls[45]. Both adoptive transfer of T$_{regs}$ and induction of T$_{regs}$ provides protection from GvHD[9, 46]. Efficient and safe methods to expand T$_{regs}$, as proposed in the current study, may facilitate the use of T$_{regs}$ to combat GvHD. Although the current study provides strong evidence for a protective role of aPC in acute GvHD through expansion of T$_{regs}$, we cannot exclude that aPC may convey additional beneficial effects during other phases of the GvHD[1]. Thus, aPC's cytoprotective and anti-apoptotic effects are well established[17], and aPC has been shown to protect from radiation injury[47], suggesting that aPC may protect from the initial tissue damage during the transplant conditioning regimen and/or from damage ensuing during the allo-response[1]. Protection using aPC during these stages would, however, require systemic therapies with the potential caveat of side effects, such as haemorrhage. Using aPC mutants conveying cytoprotective signaling but lacking anticoagulant function[48], may avoid the latter. Further studies are needed to evaluate the potential role of aPC in other GvHD phases and the effect of aPC mutants on T$_{regs}$ and T-cell reactivity.

Endothelial dysfunction has been repeatedly demonstrated following whole body irradiation or chemotherapy, therapies typically used in transplant conditioning regimen[14] and in the setting of GvHD[10–13, 47]. On the basis of these findings, endothelial protective therapies have been proposed to convey beneficial effects in GvHD, but their translation was only partial successful[49]. Endothelial dysfunction is intimately associated with a loss of endothelial TM and lower plasma levels of aPC[22, 50, 51]. Reconstitution of aPC's effect ex vivo may be a safe yet efficient approach to compensate for the inevitable impairment of

endothelial-function and TM-function during pre-conditioning of patients.

The efficacy of aPC to protect from GvHD despite ex vivo preincubation and washing of T-cells prior to their transfer into mice is an intriguing observation of potential translational relevance. The mechanism underlying the sustained effect of aPC on T-cells and $T_{regs}$ remains to be fully explored. We demonstrate that aPC-preincubation increases the frequency of $T_{regs}$ by expanding pre-existing $T_{regs}$ and inducing $T_{regs}$ from $CD4^+$ T-cells. $T_{regs}$ mainly exert their suppressive function within the first 2 days after transplantation[8, 52]. The presence of $T_{regs}$ specifically during the initial days is sufficient for a long lasting protective effect in GvHD. Accordingly, we speculate that even a transient increase of $T_{regs}$ following aPC ex vivo preincubation might be sufficient to increase the frequency of $T_{regs}$ during this initial phase. Alternatively, we recently demonstrated that aPC epigenetically controls gene-expression in the context of diabetic nephropathy[51]. The importance of epigenetically controlled gene-expression for $T_{reg}$ differentiation is well established[53]. Hence, it appears possible, but remains to be shown, that aPC induces $T_{regs}$ through epigenetic mechanisms.

The findings of the current work suggest a novel mechanistic link between the adaptive immunoreaction, $T_{reg}$ function, and aPC, and thus between endothelial function and the adaptive immune response. Recently, Gur-Cohen et al. established a novel function of TM-dependent PC-activation for retention of hematopoietic stem cell recruitment by limiting NO-production via aPC-EPCR-PAR signaling[54]. These and the current findings provide novel insights into the regulation of leucocyte homeostasis and their function through mechanisms depending on the coagulation protease aPC. Intriguingly, the receptors targeted by aPC on hematopoietic stem cells and T-cells are partially disjunct, which may allow targeting the underlying mechanisms through distinct pharmacological approaches. By identifying a mechanism allowing $T_{reg}$ expansion, the current study proposes a new therapeutic approach to mitigate GvHD while maintaining the GvL-effect.

## Methods
**Reagents.** The following antibodies were obtained from eBioscience (Germany): FITC or eFluor450-conjugated anti-mouse MHC Class I (H-2D[b]), FITC-conjugated anti-mouse MHC Class I (H-2K[b]), FITC or APC-conjugated anti-mouse CD4, PE-conjugated anti-mouse FOXP3, PE-Cyanine7-conjugated anti-human FOXP3, PE-conjugated anti-human/mouse Ror-γt, FITC-conjugated anti-human CD4. The following antibodies were obtained from BioLegend (Germany): PE-conjugated anti-mouse IFN-γ, IL-17A, TNF-α, IL-10, PerCp-Cy5.5 or PE-conjugated anti-human/mouse T-bet, AlexaFluor 647-conjugated anti-human/mouse FOXP3, APC (Allophycocyanin)-conjugated anti-human CD4, PE-conjugated anti-human IFN-γ, IL-17A, IL-10, TNF-α. Mouse monoclonal antibodies to human PAR1 (ATAP2) and PAR2 (SAM11), rabbit polyclonal antibody to PAR3 (H103) were used to detect and to block the corresponding PAR (SantaCruz, Heidelberg, Germany). Goat polyclonal antibody to PAR4 (S-20) was used to block PAR 4 and rabbit polyclonal antibody to PAR4 (H-120) was used to detect PAR4 (SantaCruz, Heidelberg, Germany). Rat monoclonal antibody to human endothelial cell protein C receptor (EPCR) antibody was used to detect and to block EPCR (Sigma-Aldrich, Germany). Following antibodies were obtained from Cell Signalling Technology (Germany): p44/42 MAPK (Erk1/2, 1:1000), phospho-p44/42 MAPK (Erk1/2, Thr202/Tyr204, 1:1000), p38 MAPK (1:1000), phospho-p38 MAPK (Thr180/Tyr182, 1:1000), HRP-conjugated α-rabbit or α-mouse IgG secondary antibodies. Anti-human FOXP3 antibody (BioLegend) was used to detect FOXP3 by immunoblot.

Cell Proliferation Dye eFluor450 (eBioscience) was used to label the cells and analyse cell proliferation. FITC-conjugated AnnexinV (eBioscience) together with propidium iodide (PI) was used for apoptosis detection. Streptavidin Particles Plus-DM (Becton Dickinson, Franklin Lakes, NJ, USA) and a Dynal magnet (Life Technologies, USA) were used for lineage depletion.

Further reagents obtained from Sigma-Aldrich, Germany: RPMI 1640, and Brefeldin A (BFA).

Other reagents used in the current study were protease inhibitor cocktail (Roche diagnostics GmbH, Germany); AIM V Medium (ThermoScientific, Germany); BCA reagent (ThermoScientific, Germany); PVDF membrane and

immobilion enhanced chemiluminescence reagent (Millipore GmbH, Germany); FOXP3/Transcription Factor Staining Buffer Set (eBioscience).

LEGENDplex™ Mouse Th Cytokine Panel (13-plex, BioLegend) was used for cytokine measurement in mouse plasma. Human/mouse TGFβ1 ELISA Ready-SET-Go! (2nd generation, eBioscience) was used to measure TGFβ1 in mouse plasma and in supernatant of primary human T-cell MLR. The details of antibodies are provided in Supplementary Table 1.

**Mice.** C57BL/6 (B6;H2[b]), BALB/c (H2[d]), and C3H/HeNRj (H2[k]) were purchased from Janvier S.A.S., St. Berthevin Cedex, France. Humanized NOD.Cg-Prkdc[scid] Il2rg[tm1Wjl] H2-Ab1[tm1Gru]Tg (HLA-DRB1)31Dmz/SzJ (NSG-Ab°DR4, female, 8–10 weeks old) mice[35] were purchased from Jackson Laboratory. These NSG-Ab° DR4 mice lack expression of the murine *Prkdc* gene, the X-linked *Il2rg* gene, and MHC class II, but express the human leukocyte antigen DR4 gene. The expression of HLA-DR4 in these mice leads to the development of allo-GvHD after engraftment of human DR4[–]negative CD4[+] T-cells.

APC[high] mice (female, 8–10 weeks old), which express a transgene resulting in expression of a human protein C variant (D167F/D172K) in the liver, which can be efficiently activated in the absence of thrombomodulin, resulting in high plasma concentrations of aPC, have been previously described and have been backcrossed onto the C57BL/6 (B6;H2[b]) background for more than 10 generation[22]. PAR3[−/−] (male, 8–10 weeks old) and PAR2[−/−] (male, 8–10 weeks old) mice were obtained from Jackson Laboratory and have been backcrossed onto the C57BL/6 (B6;H2[b]) background for at least 10 generation[31]. DEREG mice have been backcrossed onto the C57BL/6 (B6;H2[b]) background for at least 10 generation. In this transgenic mouse model, the diphtheria toxin receptor (DTR) and the enhanced green fluorescent protein (eGFP) are expressed under the control of an additional FOXP3 promoter. This enables specific depletion of FOXP3[+] $T_{regs}$ by injection of diphtheria toxin (DT). Mice transplanted with DEREG mice (male, 8–10 weeks old)-derived $T_{regs}$ were injected intraperitoneally with diphtheria toxin (20 ng/g body weight) on day 1 and 2 post transplantation to deplete $T_{regs}$[27].

Mice were housed in pathogen-free conditions in individually ventilated cages in the central animal facility of the Medical Faculty of the Otto-von-Guericke University, Magdeburg. Permission for all animal experiments was granted by the local Animal Care and Use Committee (Landesverwaltungsamt Halle, Germany) and all experiments were conducted following standards and procedures approved.

**Preparation of aPC.** aPC was generated as previously described with slight modifications[55]. Briefly, prothrombin complex (Prothromplex NF600), containing PC-dependent vitamin K-dependent coagulation factors, was reconstituted with sterile water and supplemented with $CaCl_2$ (final concentration: 10 mM). The calcium dependent monoclonal antibody to PC (HPC4) covalently linked to a column filled with Affigel-10 resin was used for PC purification. This column was equilibrated at 4 °C with 1 column volume (CV) of washing buffer (0.1 M NaCl, 20 mM Tris, pH7.5, 5 mM benzamidine HCl, 2 mM $Ca^{2+}$, 0.02% sodium azide) at a flow rate of 0.5 ml/min, which was used during the whole purification step. After PC-binding the column was washed first with 1 CV of washing buffer, followed by 1 CV with a buffer containing high salt concentration (0.5 M NaCl, 20 mM Tris, pH 7.5, 5 mM benzamidine HCl, 2 mM $Ca^{2+}$, 0.02% sodium azide). Benzamidine was washed off the column using a buffer of 0.1 M NaCl, 20 mM Tris, pH 7.5, 2 mM $Ca^{2+}$, and 0.02% sodium azide using again 1 CV. To elute PC 1 CV of elution buffer (0.1 M NaCl, 20 mM Tris, pH 7.5, 5 mM EDTA, 0.02% sodium azide, pH 7.5) was used. The eluate was fractionated by collecting 5 ml portions. Peak fractions, identified by measuring absorbance at 280 nm, were pooled. Human plasma thrombin (5% w/w, incubated for 3 h at 37 °C) was used to activated the recovered PC. Residual thrombin was inhibited by hirudin (1 h at 37 °C). To obtain purified aPC thrombin was separated using an ion exchange chromatography with FPLC (ÄKTAFPLC®, GE Healthcare Life Sciences). To this end, a Mono Q anion exchange column (GE Healthcare Life Sciences) was equilibrated with 5 CV of 20 mM Tris pH 7.5, 100 nM NaCl. After applying the solution containing aPC and a washing step with 5 CV of the solution for equilibration a 10–100% gradient using 5 CV of a 20 Mm Tris, pH 7.5, 1 M NaCl buffer was applied to the column to elute aPC. aPC eluted at 0.46 M NaCl (~ 36 mS/cm conductivity). Fractions of 0.5 ml were collected during the peak and pooled. Proteolytic activity, integrity, and purity of aPC was ascertained with the chromogenic substrate SPECTROZYME® PCa and by Coomasie-staining of the purified protein on 10% SDS-PAGE gel.

**GvHD models.** Pan T-cells were isolated from whole spleen by magnetic bead depletion of non-T-cells using mouse Pan T-cell isolation Kit II (Miltenyi Biotec) following the manufacturer's recommendations. Purity of T-cells was ascertained by FACS and ranged from 95 to 98%.

$T_{regs}$ were isolated from whole spleen using the $CD4^+CD25^+$ Regulatory T-cell Isolation Kit (Miltenyi Biotec) and purity was ascertained by FACS analysis and ranged from 95 to 98%. Bone marrow (BM) was prepared by flushing BM from isolated tibia and femur bones using RPMI complete medium. RBCs in BM were lysed using RBC lysis buffer (Buffer EL; Qiagen).

To induce MHC-mismatched GvHD we transplanted either C57BL/6 or BALB/c BM along with C57BL/6 or BALB/c splenic T-cells into BALB/c or C57BL/6, respectively. Recipient mice, 8–10 weeks of age, were conditioned with total body

**Table 1 Assessment of clinical GvHD in transplanted animals**

| Criteria | Grade 0 | Grade 1 | Grade 2 |
|---|---|---|---|
| Weight loss | <10% | >10% to <25% | >25% |
| Posture | Normal | Hunching noted only at rest | Severe hunching impairs movement |
| Activity | Normal | Mild to moderately decreased | Stationary unless stimulated |
| Fur texture | Normal | Mild to moderate ruffling | Severe ruffling/poor grooming |
| Skin integrity | Normal | Scaling of paws/tail | Obvious areas of denuded skin |

irradiation (TBI) of 11 Gy (single dose) for BALB/c and 13 Gy for C57BL/6 mice on day 0. For TBI the BioBeam 8000 (Gamma Service Medical GmbH, Germany) providing gamma irradiation ([137]Cs) was used. Mice were immobilized by anaesthesia with intra-peritoneal injection of 100 μl Ketavet and Rompun solution (Ketavet 20 mg/ml and Rompun 1 mg/ml) and kept in a radiation chamber. Four hours after irradiation, recipient mice received intravenously $5 \times 10^6$ mismatched (C57BL/6 (male, 8–10 weeks old) → BALB/c (female, 8–10 weeks old) or BALB/c (male, 8–10 weeks old) → C57BL/6 (female 8–10 weeks old)) BM cells with purified $0.5 \times 10^6$ mismatched pan T-cells, or $0.5 \times 10^6$ PAR3$^{-/-}$ pan T-cells, or $0.5 \times 10^6$ PAR2$^{-/-}$ pan T-cells, or $0.4 \times 10^6$ PAR3$^{-/-}$ pan T-cells with $0.1 \times 10^6$ PAR3$^{-/-}$ T$_{regs}$, or $0.4 \times 10^6$ PAR2$^{-/-}$ pan T-cells with $0.1 \times 10^6$ PAR2$^{-/-}$ T$_{regs}$. In a subset of experiments, T-cells or T$_{regs}$ were preincubated with aPC (20 nM, 1 h, 37 °C in AIM V serum-free medium). Following 1 h incubation with aPC cells were washed with PBS. Control T-cells were exposed to the same medium without aPC. In further experiments, T$_{regs}$ were isolated from DEREG or transgene-negative littermate (wt-control) mice using the CD4$^+$CD25$^+$ Regulatory T-cell Isolation Kit (Miltenyi Biotec) and purity was ascertained by FACS analysis and ranged from 95 to 98%. T$_{regs}$ or T-cells were separately preincubated with aPC (20 nM, 1 h, 37 °C) prior to BM transplantation. In the experiments in which mice were transplanted with T$_{regs}$ from DEREG or C57BL/6 mice, recipient mice in both groups were injected intraperitoneally with diphtheria toxin (20 ng/g body weight) on day 1 and 2. To induce allogenic GvHD with human T-cells in mice we used the "humanized" NSG-Ab° DR4 mice (N = 12) (see above). For preconditioning these mice received TBI of 2 Gy. After 4 h these mice were intravenously injected with $4 \times 10^6$ of human CD4$^+$ T-cells (HLA-DR4$^-$) without or with aPC-preincubation (20 nM, 1 h, 37 °C). In all experiments, cells were washed with PBS following aPC-preincubation and before transplantation.

**HLA-DRB1-04 genotyping**. To isolate HLA-DRB1-04 (HLA-DR4)-negative T-cells blood samples were obtained from 9 volunteers after getting their written informed consent. Expression of DR4 was assessed by real time polymerase chain reaction (PCR) using a Light Cycler (CFX Connect, Real Time System, BioRad). Briefly, genomic DNA was extracted from donor's PBMC ($1 \times 10^6$) using the phenol chloroform extraction method. The HLA-DRB1-04 gene was amplified using the following forward primer: 5′ GTTTCTTGGAGCAGGTTAAACA-3′ and two reverse primers in the same reaction: 5′-CTGCACTGTGAAGCTCTCAC-3′, 5′-CTGCACTGTGAAGCTCTCCA-3′[56]. The following cycling parameters were used: initially 2 min at 95 °C, followed by 40 cycles of 95 °C, 10 s; 68 °C, 10 s; 72 °C, 23 s with a single acquisition per cycle at 72 °C. All temperature transitions were 20 °C/s. Samples were then subjected to a melting curve analysis with the following conditions: 95 °C, 0 s (slope 20°/s); 65 °C, 10 s (slope 20°/s) and then heated to 95 °C with a slope of 0.3°/s using step acquisition. Positive and negative samples were distinguished by the presence or absence of fluorescence signal during the PCR reaction and the presence of a melting peak (~91 °C). Six individuals were identified as being DR4$^-$.

**Assessment of GvHD**. Individual weights of transplanted mice were obtained and recorded on day l and weekly thereafter until the time of analysis. Survival was checked once daily. The clinical score of GvHD was assessed by a scoring system described in Table 1 that incorporates five physical parameters: weight loss, posture (hunching), activity, fur texture, and skin integrity. Every week mice were evaluated and graded from 0 to 2 for each criterion. A clinical index was subsequently generated by summation of the five criteria scores (maximum index = 10, Table 1)[57].

Representative samples of GvHD target organs (gut, liver, and skin) were excised from recipients 14 days post-BM transplantation. Formalin-fixed tissues were paraffin embedded and sectioned (5-μm-thick sections). Sections were stained with hematoxylin and eosin for histologic examinations. Pathological scoring was conducted by an experienced pathologist (TK) blinded to the groups. Intestinal GvHD was scored based on the frequency of crypt apoptosis (0, rare to none; 1, occasional apoptotic bodies per 10 crypts; 2, few apoptotic bodies per 10 crypts; 3, the majority of crypts contain an apoptotic body; 4, the majority of crypts contain >1 apoptotic body)[58]. The severity of skin GvHD was assessed by a scoring system that incorporates following parameters: epidermic atrophy, hair follicle loss, increased collagen density in dermis, and inflammation. The slides were graded from 0 to 2 for each parameter[58]. The severity of liver GvHD was assessed by a scoring system that incorporates following parameters: bile ducts infiltrated by

lymphocytes (0: not present, 1: one or more lymphocytes in one bile duct, 2: lymphocytes in more than 1 bile duct, 3: lymphocytes in all bile ducts) and portal inflammation (0: not present, 1: inflammatory cells in some portal tracts, 2: inflammatory cells in most portal tracts, 3: packing of inflammatory cells with or without spill-over into adjacent parenchyma in some or most portal tracts)[59].

**Cell isolation and mixed lymphocyte culture**. For ex vivo assessment of allogenic T-cell activation, mixed lymphocyte reaction (MLR) were conducted by incubating pan T-cells or effector T-cells (T$_{eff}$) with non-T-cells, containing AgPC. For isolation of human T-cells or T$_{eff}$ first PBMCs were obtained from peripheral blood using Ficoll-Paque (GE Healthcare) gradient and then pan T-cells were isolated by magnetic bead depletion of non-T-cells using the human pan T-cell Isolation Kit (Miltenyi Biotec) following the manufacturer's recommendations. In a subset of experiments, regulatory T-cells were isolated using the human CD4$^+$CD25$^+$ regulatory T-cell Isolation Kit II (Miltenyi Biotec) and untouched CD4$^+$CD25$^-$ cells obtained during regulatory T-cells isolation were used as effector T-cells. Purity of cells was ascertained by FACS and ranged from 95 to 98%. Non-T-cells were irradiated (30 Gy) and used as AgPC. Pan T-cells and non-T-cells from two genetically distinct (non-related) individuals were co-cultured to trigger allogenic T-cell reactivity. For ex vivo assessment of murine allogenic responses, BALB/c and C57BL/6 splenocytes were isolated by disrupting the spleen with a 100 μm cell strainer. Pan T-cells were isolated from splenocytes by magnetic bead depletion of non-T-cells using mouse pan T-cell Isolation Kit II (Miltenyi Biotec) following the manufacturer's recommendations. Mismatched non-T-cells, used as AgPC, were irradiated with 30 Gy and co-cultured with pan T-cells to trigger the MLR response.

Both human and mouse T-cells were cultured in AIM V serum-free medium (Life Technologies) for 2 h (37 °C, 5% CO$_2$) before performing MLR. MLRs with human or mouse cells were either conducted in the presence of aPC or following preincubation of T-cells with aPC. When conducting MLR in the presence of aPC human or mouse $1 \times 10^5$ pan T-cells were incubated with $3 \times 10^5$ irradiated allogenic non-T-cells (ratio 1:3) for 96 h and aPC was added every 12 h (20 nM final concentration). For experiments with aPC-preincubation human or mouse, $1 \times 10^5$ pan T-cells were preincubated with aPC (20 nM) or an equal volume of PBS (control) in AIM V serum free medium (1 h, 37 °C), washed with PBS to remove aPC, and then co-cultured with $3 \times 10^5$ irradiated allogenic non-T-cells (ratio 1:3) for 96 h. Reactivity of human and mouse T-cells was assessed by measuring [³H] thymidine incorporation during the last 16 h of the incubation time and [³H] thymidine was added at 0.2 Ci per well. At the end of the incubation period, cells were harvested and radioisotope incorporation was measured as an index of lymphocyte proliferation by betaplate liquid scintillation counter (MicroBeta, Wallac, Finland)[60].

**Cell Proliferation assay**. Human and mouse pan T-cells were isolated as mentioned above. Preincubation of pan T-cells with aPC (20 nM) or PBS (control) was conducted as described above and cells were washed with PBS. After washing control (without aPC-preincubation) and aPC preincubated T-cells were labelled with eFluor450 (5 μM) for 5 min at 37 °C. Cells were then washed with RPMI complete medium three times followed by a final washing step with PBS and again re-suspended in AIM V medium. Human and mouse pan T-cells without (PBS control) or with aPC-preincubation were co-cultured with $3 \times 10^5$ irradiated allogenic non-T-cells and proliferation of T-cells were measured on day 1 and 4 by FACS.

For isolation of human T$_{regs}$ pan T-cells were isolated as described above, were labelled with FITC-conjugated anti-CD4 and APC-conjugated anti-CD25 antibodies, and were FACS sorted using BD FACS Aria III (BD Bioscience, Germany). Subsets of T$_{regs}$ were preincubated with aPC (20 nM, 1 h, 37 °C) or PBS (control) in AIM V medium, followed by 1 washing step with PBS. After washing both control and aPC treated T$_{regs}$ were labelled with eFluor450 as described above. T$_{regs}$ without (control) or with aPC-preincubation were seeded into 96 well plates (25,000 cells per well), coated with αCD3 (10 μg/ml) and αCD28 (8 μg/ml) antibodies and proliferation of T$_{reg}$ cells was measured on day 1 and 6 by FACS Canto II (BD Bioscience, Germany). Data was analysed using FlowJo software.

**Apoptosis detection**. MLR was set up with T-cells and allogenic AgPC and T-cell apoptosis was detected on day 3 and 6 using anti-CD3, annexin V surface staining,

and propidium iodide (PI) DNA staining by flow cytometry. Data was analysed using FlowJo software.

**Immunoblotting.** To determine receptor-expression on T-cells cell and T-cell subsets (CD4$^+$, CD8$^+$, T$_{regs}$) lysates of purified human T-cells were prepared using RIPA buffer containing 50 mM Tris (pH7.4), 1% NP-40, 0.25% sodium-deoxycholate, 150 mM NaCl, 1 mM EDTA, 1 mM Na$_3$VO$_4$, 1 mM NaF supplemented with protease inhibitor cocktail. Lysates were centrifuged (13,000×$g$ for 10 min at 4 °C) and pelleted debris was discarded. BCA assay was used to quantify protein concentration in supernatants. Equal amounts of protein were electrophoretically separated on 10% SDS polyacrylamide gel, and transferred to PVDF membranes. Membranes were incubated with primary antibodies against PAR1 (1:200), PAR2 (1:200), PAR3 (1:200), PAR4 (1:200), EPCR (1:500), or GAPDH (1:20,000). After overnight incubation at 4 °C membranes were washed with TBST and incubated with anti-mouse IgG (1:2000) or anti-rabbit IgG (1:2000) horseradish peroxidase-conjugated antibodies for 1 h at room temperature. The enhanced chemiluminescence system was used to develop blots. To compare and quantify levels of proteins, the density of each band was measured using Image J software. GAPDH was used to control for equal loading (full western blot images are shown in Supplementary Figs. 24 and 25).

To analyse the effect of aPC on FOXP3 expression in T$_{eff}$ cells, protein lysates were obtained from T$_{eff}$ cells following MLR without or with aPC-preincubation. Following MLR, T$_{eff}$ cells were re-sorted by MACS and total lysates were prepared using RIPA lysis buffer as above. Immunoblotting was conducted as described in the previous paragraph using a primary antibody against FOXP3 (1:1000) and a secondary anti-mouse IgG (1:2000) horseradish peroxidase-conjugated antibody Lamin was used to control for equal loading (full western blot images are shown in Supplementary Fig. 23).

For analyses of aPC signaling in T-cells, protein lysates were obtained from T-cells following MLR with or without aPC-preincubation. Following MLR T-cells were re-sorted by MACS and total lysates were prepared using RIPA lysis buffer as above. After protein estimation equal amounts of protein were electrophoretically separated on 10% SDS polyacrylamide gel, and transferred to PVDF membranes. Membranes were incubated with primary antibodies against p38 (1:1000), phospho-p38 (1:1000), ERK (1:1000), or phospho-ERK (1:1000). After overnight incubation at 4 °C membranes were washed with TBST and incubated with anti-mouse IgG (1:2000) or anti-rabbit IgG (1:2000) horseradish peroxidise-conjugated antibodies for 1 h at room temperature. The enhanced chemiluminescence system was used to develop blots. To quantify and compare protein levels, the density of each band was measured using Image J software. GAPDH was used to control for equal loading (full western blot images are shown in Supplementary Fig. 26)

**Immunoprecipitation.** For immunoprecipitation total cellular proteins were isolated from human primary T$_{regs}$ with RIPA lysis buffer containing Protease/Phosphatase Inhibitor Cocktail 1X (Cell Signalling Technology, Germany). Lysates (200 μg) were combined with 2 μg of PAR2 (SAM11) or PAR3 (H103) antibody and incubated overnight at 4 °C. Immunoprecipitates were purified with protein A/G agarose beads and washed with PBS. Immunoprecipitates were fractionated by SDS-PAGE (10%), transferred to membranes, and subjected to immunoblotting with PAR3 (H103) or PAR2 (SAM11) antibody, respectively, and secondary antibodies as described above (full western blot images are shown in Supplementary Fig. 27).

**Functional PAR-signaling in vitro assays.** To evaluate the functional relevance of PARs on human pan T-cells, complimentary approaches were used. Human pan T-cells were incubated with N-terminal blocking anti-PAR1 (ATAP-2, 10 μg/ml), anti-PAR2 (SAM-11, 10 μg/ml), anti-PAR3 (H-103, 20 μg/ml), or anti-PAR4 (S-20, 20 μg/ml) antibodies for 1 h, or with inhibitors blocking receptor activation by the tethered ligand (FR1113 for PAR1, ML354 for PAR4; all from Tocris)[31, 61–66]. Following incubation of pan T-cells with N-terminal blocking antibodies or signaling inhibitors, pan T-cells with aPC-incubation (20 nM, 1 h, 37 °C) or without aPC-incubation were combined with allogenic AgPC and co-cultured for 96 h for MLR.

**EPCR blocking assay.** To determine the role of EPCR in aPC signaling in T-cells, primary human T-cells were incubated with a rat monoclonal antibody to human endothelial cell protein C receptor (EPCR, 2 μg/ml, 1 h, 37 °C). Following incubation, cells were washed with PBS, incubated with aPC (20 nM, 1 h, 37 °C), and again washed with PBS. For control the EPCR blocking antibody was omitted, but cells were otherwise treated the same. T-cells with EPCR inhibition and aPC-preincubation, with only aPC-preincubation, or without EPCR inhibition and aPC-preincubation were co-cultured with allogenic AgPC for 96 h for MLR.

**Virus production and transduction.** To ascertain the role of human PAR2, we generated primary PAR2 knockdown pan T-cells. To this end, HEK-293T cells (ATCC) were routinely maintained in DMEM high glucose medium (Sigma-Aldrich) supplemented with 10% heat-inactivated FBS. Cells were transfected using the calcium-phosphate method and a third-generation lentiviral

vector system 24 h after seeding 11 × 10$^6$ cells per 15 cm dish. In total, 39.5 μg pMD2.G (encoding the VSV G envelope protein), 73 μg psPAX2 (encoding the packaging proteins), and 112.5 μg transfer lentiviral plasmids (each at 1 mg/ml in TE buffer, pH 8) were used for five 15 cm dishes per shRNA construct[67]. Viral supernatants were harvested 24, 48, and 72 h post-transfection and filtered using 0.45 μm filters. Lentiviral particles were concentrated using the PEG 6000-based method[68]. Concentrated viral particles were snap frozen on crushed dry ice and stored at −80 °C until usage. For RNA interference we used pLKO.1 lentiviral vectors expressing shRNA for hPAR2 (TRCN0000006769, TRCN0000006770).

Pan T-cells were isolated from PBMCs of healthy donors as described before. In brief, 2 × 10$^6$ cells were collected in 15 ml polypropylene Falcon tubes, re-suspended in a given volume of concentrated lentiviral vector supernatant (50–100 μl) in the presence of Polybrene (5 μg/ml). Tubes were centrifuged at 1500×$g$ for 1.5 h at 32 °C. Cells (1 × 10$^6$) were then seeded in 6-well plates and after 48 h cells were washed and incubated with aPC (20 nM, for 1 h at 37 °C in AIM V serum-free medium) and seeded on 96-well plates coated with anti-human CD3 (1 μg/ml) and anti-human CD28 (0.5 μg/ml) at a density of 5 × 10$^4$ per well for proliferation assay.

**Semi-quantitative reverse transcription (RT)-PCR.** RNA was extracted using the Invitrap® spin cell RNA mini kit (Stratec) following the manufacturer's instructions. Reverse-transcription was done by RevertAid First Strand cDNA Synthesis Kit (Thermo Scientific, Germany). cDNA was amplified by polymerase chain reaction (PCR) using Taq Polymerase (GoTaq, Promega) and the following cycling conditions: denaturation at 95 °C (30 s), annealing at 60 °C (30 s), and extension at 72 °C (20 s); 40 cycles. Amplimers were visualized on an agarose gel containing ethidium bromide. The β-actin was used as a positive control and for normalization of the PAR2 signal to estimate hPAR2 expression. The primer sequences were hPAR2_F: TTGGCTGACCTCCTCTCTGT, hPAR2_R: CGAT-GACCCAATACCTCTGC and β-actin_F: GCCTCGCCTTTGCCGAT, β-actin_R: CCACGATGGAGGGGAAGAC.

**PAR3 cleavage assay.** Analyses of N-terminal PAR3-derived peptides were conducted as previously described[30]. Human ([NH$_2$] SGMENDTNNLAKPTL-PIKTFRGAPPNSFEEFPFSALEGWTGATIT [COOH]) and mouse ([NH$_2$] CQSGINVSDNSAKPTLTIKSFNGGPQNTFEEFPLSDIEGWTGATTT [COOH]) PAR3-derived peptides were obtained from bioSYNTHESIS (Lewisville, TX, USA). Peptides, dissolved in PBS (pH 8.4), were incubated at a final concentration of 50 μM in Hepes buffered saline (HBS: 25 mM Hepes, pH = 7.4, 147 mM NaCl and 4 mM KCl) with 2 mM CaCl$_2$ and 0.6 mM MgCl$_2$ at 37 °C with the addition of aPC (500 nM) and hirudin (25 μg/ml; to inhibit potential residual thrombin) or thrombin (10 nM). The final reaction volume was 50 μl. After 2 h, 6 h, or 24 h, 1 μl of the respective reaction mixture was diluted in 9 μl solvent (2% acetonitrile, 0.1% trifluoric acid in water) and subsequently subjected to mass spectrometry.

LC-MS/MS was performed on a hybrid dual-pressure linear ion trap/orbitrap mass spectrometer (LTQ Orbitrap Velos Pro, Thermo Scientific, San Jose, CA, USA) equipped with an EASY-nLC Ultra HPLC (Thermo Scientific). For analysis, 10 μl of peptide sample preparations were fractionated on a 75 μm (inner diameter), 50 cm PepMap C18-column, packed with 2 μm resin (Dionex, Thermo Scientific). Separation was achieved through applying a gradient from 2 to 35% ACN in 0.1% formic acid over 120 min at a flow rate of 300 nl/min. An Orbitrap full MS scan was followed by up to 15 LTQ MS/MS runs using collision-induced dissociation (CID) fragmentation of the most abundantly detected peptide ions. Essential MS settings were as follows: full MS (FTMS; resolution 60 000; $m/z$ range 400–2000); MS/MS (Linear Trap; minimum signal threshold 500; isolation width 2 Da; no dynamic exclusion; singly charged ions were excluded from selection). Normalized collision energy was set to 35%, and activation time to 10 ms. Raw data processing and cleavage site identification were performed using PEAKS Studio V.8.0 (Bioinformatics Solutions, Canada).

**Flow cytometry analyses.** Murine splenocytes were isolated and stained with FITC or eFluor450-conjugated H2$^b$, FITC or APC-conjugated CD4, PE-conjugated IFN-γ, IL-17A, TNFα, IL-10, PE-conjugated T-bet, ROR-γt, FOXP3 using the FOXP3/Transcription Factor Staining Buffer Set (eBioscience) according to the manufacturer's instructions (gating strategies are represented in Supplementary Figs. 16–19). Intracellular staining for cytokines in human MLR was done 48 h after MLR following addition of Brefeldin A (10 μg/ml) during the last 4 h. Cells were stained with FITC-conjugated CD4 or APC (Allophycocyanin)-conjugated CD4, PE-conjugated IFN-γ, IL-17A, IL-10, TNF-α, PerCp-Cy5.5 or PE-conjugated T-bet, Alexa Fluor 647-conjugated FOXP3 or PE-Cyanine7-conjugated anti-human FOXP3 antibodies, using the FOXP3/Transcription Factor Staining Buffer Set (eBioscience) according to the manufacturer's instructions (gating strategies are represented in Supplementary Figs. 20 and 21). Cells were analyzed using FACS Canto II (BD Biosciences) and FlowJo software (TreeStar).

Human T$_{eff}$ cells were stained with FITC-conjugated CD4, APC-conjugated CD25, using the FOXP3/Transcription Factor Staining Buffer Set (eBioscience) according to the manufacturer's instructions (gating strategy is represented in Supplementary Figs. 22). Cells were analyzed using FACS Canto II (BD Biosciences) and FlowJo software (TreeStar).

**Cytokine measurement by ELISA.** IFN-γ, TNF-α, IL-17A, and IL-10 were measured in mouse plasma collected at day 14 post transplantation using Legendplex TM Mouse TH Cytokine Panel (13-plex). Measurements were conducted with the BD LSR Fortessa analyser (BD Bioscience) and the data were analysed using the software Legendplex TM V7.0 for Windows (VigenTech Inc.). TGFβ1 was measured in same mouse plasma samples or the supernatant from the primary T-cell MLR experiments using the human/mouse TGFβ1 Ready-SET-Go! ELISA according to the manufacturer's instructions.

**Leukaemia models.** Retroviral infection of hematopoietic progenitor cells was performed as previously described with minor modifications[69]. To generate myeloid leukaemia on a BALB/c background lineage-negative, Esam-1-positive, Kit-positive (L-E+K+) cells were sorted from BALB/c BM and infected with an MSCV-MLL-AF9-IRES-GFP construct. Primary, sublethally irradiated (7 Gy) 6–8-week-old female BALB/c recipients were injected with up to $5 \times 10^4$ pre-leukaemic progenitors. Secondary, lethally irradiated BALB/c-recipient mice received $5 \times 10^3$ GFP/Kit-co-expressing (GFP+Kit+) MLL-AF9 leukaemic cells (BALB/c background) along with $5 \times 10^6$ C57BL/6 BM cells and purified $0.5 \times 10^6$ C57BL/6 pan T-cells without or with aPC-preincubation (20 nM, 1 h, 37 °C, in AIM V serum-free medium).

To generate AML1-ETO-driven myeloid leukaemia, C57BL/6 BM was isolated following standard procedures from 8 to 12-week-old mice. Lineage depletion was performed as described before using Streptavidin Particles Plus-DM and a Dynal magnet[69]. Lineage-depleted cells were stained with streptavidin BV421 (405226), cKIT AF647 (105818) and Sca-1 FITC (108106) antibodies (all antibodies were received from BioLegend, San Diego, CA, USA) and then used for cell sorting. Sorted LSK cells (Lin- cKIT+ Sca-1+) were pre-stimulated on RetroNectin-coated plates (Takara Bio USA, Inc., Mountain View, CA, USA) in serum-free medium (StemSpan^TM SFEM, Stemcell Technologies, Vancouver, Canada) supplemented with SCF, TPO, IL-3, and IL-6 (all cytokines from PeproTech, Hamburg, Germany) overnight. Overall, 20 h after stimulation, LSK cells were infected twice (8 h gap) by spinfection with retroviral particles containing the oncogenic vectors MSCV-AML1-ETO9a-IRES-GFP, and MSCV-KRAS-IRES-GFP (AE9a/KRAS). Cells were analyzed 48 h after first infection for GFP-expression using flow cytometry and $7.5 \times 10^4$ GFP+ cells were injected into sublethally irradiated (7 Gy) primary-recipient mice (C57BL/6). BM of diseased mice was isolated and $2 \times 10^4$ GFP+ cells were injected into C57BL/6 wt mice along with $5 \times 10^6$ BM cells and $1 \times 10^6$ T-cells (with and without aPC pretreatment) derived from C3H/HeNRj mice.

Tumor load was determined in peripheral blood samples obtained 2 (C57BL/6 model) or 4 (BALB/c model) weeks following BM-transplantation and injection with leukaemic cells. Tumor load was determined as frequency (%) of GFP+ leukaemic cells using FACS Canto II (BD Biosciences) and FlowJo software (TreeStar).

**Statistics.** Survival was ascertained by Kaplan–Meier log-rank analyses. The Kolmogorov–Smirnov test was used to determine whether the data are consistent with a Gaussian distribution. Statistical analyses were performed with the Student $t$-test, ANOVA, or log-rank test as appropriate. Prism 5 software (GraphPad Software, San Diego, CA, USA) was used for statistical analysis. Values of $P < 0.05$ were considered statistically significant.

**Data availability.** The data that support the findings of this study are available from the corresponding authors upon reasonable request.

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

## Acknowledgements

This work was supported by grants of the "Deutsche Forschungsgemeinschaft" (IS-67/4-3, IS-67/5-3, and SFB 854/B26N to B.I., SFB 854/A20 to T.F. and F.H., SFB 854/B19 to B.S., SFB 854/B14 to M.B.-W., SFB854/B16 to J.H., SFB854/A04 to M.N., SH 849/1-2 to K.S.), Sanderstiftung (2008.067.1 to T.L. and B.I.), and of the "Stiftung Pathobiochemie und Molekulare Diagnostik" to B.I. We thank Kathrin Deneser, Julia Judin, Juliane Friedrich, René Rudat, Stephanie Frey, and Rumiya Makarova for excellent technical support.

## Author contributions

S.R. designed, performed, and interpreted in vivo, in vitro, and ex vivo experiments and contributed to manuscript preparation; A.G. performed the proliferation assays. M.P. performed the FACS analysis with support from J.N., I.G., S.K., K.S., F.B., H.W., H.S., and U.B. supported mouse experiments and ex vivo analysis; D.G. conducted PAR2 kd experiments; D.R., A.C.Z., T.M.S., M.B.-W., T.F., B.S., T.L., J.H., and F.H.H. provided instrumental support and analysed data; Th.K. and M.N. conducted and interpreted mass spectrometry analyses; T.K. conducted histopathological analyses; B.I. designed and interpreted the experimental work and prepared the manuscript.

## Additional information

**Competing interests:** The authors declare no competing financial interests.

