## [Peer Review file · Nature Communications]

Reviewers' comments:

Reviewer #1 (Remarks to the Author):

In the study "Activated protein C protects from GVHD via PAR2/PAR3 signaling in regulatory T cells" the authors Ranjan et al. investigate aPC - a protein mostly recognized for its role as an anti-coagulant with strong cyto-protective and anti-inflammatory properties - in context with amelioration of GVHD.

The authors declare two new findings. First, pretreatment of pan T cells with aPC ameliorates GVHD via an increase in the number of Treg cells that control activation/proliferation of alloreacting T cells, while not reducing GVH activity. And second, Treg suppressive function is mediated through aPC receptor signaling via heterodimer PAR2/PAR3 on Treg cells.

A cardinal failure of the study is that the authors neither mention nor show that aPC in a dose dependent manner almost completely abrogates proliferation of CD4+CD25negative anti-CD3/CD28 activated T cells, in the absence of Treg cells (doi: 10.1074/jbc.M111.325951 Meilang Xue et al. 2012). Accordingly the design of the experiments - except for one - is not discriminating suppressive activity of Treg cells from direct effects of aPC on helper cells thus making conclusions regarding Treg activity invalid. Even the one experiment (Fig.3f) that does discriminate aPC treatment of Treg cells with regard to their suppressive capacity does not show how T cells proliferate in the absence of Treg cells, and is thus lacking a mandatory control.

No data are shown that aPC does cleave PAR3, nor is there evidence for signaling (intermediates) via PAR2. Also see below.

Another drawback of the study is that the cohorts are tiny: 2 mice per treatment cohort and 2 per control. Although the experiment was repeated 3 times the number of mice is too small. Since the mice are not humanized but are mouse/mouse models or transgenes engrafted with human-T cells respectively, thus there is no reason to work with such low numbers.

SEM/SD is missing in many experimental data and some statistics are not state of the art (iMFI). The authors should explain why they evaluated all frequencies by iMFI. Results should be shown in absolute numbers. SEM or SD is lacking in Fig1b, 2d, 4b, and Sup.Fig8.

Format requirements:

The article is not consistent with format requirements for submission.

The abstract should contain no references.

The main text of an Article should begin with an introduction (without heading), followed by sections headed Results, Discussion and Methods.

The Results sections may be divided by topical subheadings.

References inappropriate:

References are inappropriate; The abstract contains references.

The order in which citations are given is wrong, for example cited is: 6,1 or 7,5,8,9, . The font is different compared to others, valid for citation 15. One reference is not in formatted (Isermann, Weiler, 2003, first paragraph of the article). References 11 and 12 are redundant.

Besides that several major and minor issues leave the study incomplete and not convincing.

New findings described in this paper are

- 1) That aPC a natural compound is able to ameliorate aggressive aGVHD which the authors show by using an APChigh strain as recipient for allo T cells. The APChigh strain experiences beneficial effects in terms of improved survival and physical appearance compared to wt strain.
- 2) Reduced alloreactivity in the presence of aPC is then reproduced in vitro in MLR:
 - Pan T cell`s robust proliferation when responding to allo Antigen-presenting-cells was diminished when aPC was administered every 12hs.
 - single dose aPC pretreatment of pan T cells had exactly the same efficacy in inhibiting allogeneic

T cell proliferation.

Comment:

What is the author's definition of suppression? MLC with T responder cells depleted of Treg cells is the proper positive control for alloreactivity in this setting. Suppression should then be given as % proliferation of positive control.

Pretreatment of T cells with aPC was done in serum free medium, so the true control would be T cells pretreated with serum free medium without aPC.

Figure 2a) and b): neither the text nor the legend give a precise quantification of suppression (seems to be around 57% of indicated control).

Moreover the effect of aPC treatment on just allo-Antigen-presenting-cells is not done.

Instead of showing H3 incorporation in MLC CFSE labeling seems more adequate to measure proliferation.

3) The authors then show that single dose aPC pretreatment of pan T cells prior to transplantation improves survival, physical appearance and histopathologic marker expression in an allogeneic GVHD model.

Comment: There are no data about the stability/half-life of the applied aPC and or the pathway that may be (permanently) induced in the (transplanted)T cells. The authors should at least explain why they assume that a single dose of aPC on T cells can prevent GVHD in vivo while in their in vitro assays aPC is provided every 12 hs?

4) To analyze the underlying mechanism the authors characterized splenocytes from these GVHD mice d14 post transplantation and found

a) a 2-fold increase in Treg cells in the cohort receiving aPC pretreated grafts

b) a decrease in Th1 and TH17 cells. Th1 cytokines were less expressed in CD4+ cells while IL-10 producing cells were increased.

Comment:

That aPC increases Treg numbers is not a new finding. Xue et al. showed that aPC in splenic cells within 3 days increased the proportion of CD4+Foxp3+ cells and Foxp3 protein (nicely shown in Western blot) for about 30%.

As for the cytokine FACS analysis: Extremely divergent cell numbers are acquired for Th1 versus Th2 cell cytokine analysis (Sup. Fig2, and Sup. Fig1). It looks as if the cells for IL-17 and IL-10 analysis were pooled from different animals yet not for the Th1 cytokine FACS analysis? Moreover, the few cells acquired for IFN γ and TNF α show clear positive fractions contrasting the IL-17 and IL-10 plots that rather indicate a shift towards positivity. The authors should provide a histogram with the mean MFI and/or an ELISA for further verification of the TH2 cytokine production.

The population that clearly stains positive for Foxp3 is CD127negative while the CD127+/Foxp3 subset seems rather dim/negative for CD127. Expression of CD127 is Treg subtype specific depending on their status of activation and dependency of IL-7. Increased expression of CD127 however is described for activated Foxp3+, CD4+ Treg cells. doi:10.1002/eji.201040531. So the cells that express CD127 in this experiment are rather activated helper cells that transiently become Foxp3low positive when activated.

Why do the authors exclude CD8 T cells from analysis in general? What do the CD8 T cells secrete?

Also: Why do the authors use Biolegend Alexa Fluor 647 anti-mouse mab (Cat# 320013) in context with the FOXP3 / Transcription Factor Staining Buffer Set from eBioscience?

5) To clarify whether aPC induced Treg expansion or the differentiation of non-Treg cells the authors analyzed aPC-treated Teff- and Treg cells for their ability to expand Foxp3 cells in context with AgPC in vitro in MLR. They found

- only aPC-treated Treg were able to expand Treg cells.

Comment: H3 incorporation is insufficient to convincingly discriminate proliferation and expansion. Verification of proliferation recommends: isolation of (n)Tregs (CD4+CD25+ Foxp3+ CD127+RA-) from peripheral blood, CFSE-labeling and exposure to aPC, with doublings readily detectable in FACS. Furthermore: The inhibitory effect of aPC-pretreated Treg cells in MLC with aPC-untreated helper cells is dramatically reduced (Fig.3f, 21% reduction of proliferation) compared to MLC using aPC treated panT cells (suppression about 57% of control) (Fig.2a+b and Sup Fig. 6). These latter assays show a more than 2.5-fold higher suppressive activity when aPC treatment included Teff cells (57%). This either indicates that aPC directly abrogates Teff proliferation (as shown by Xue et al.) and/or that Teff are induced to differentiate into iTreg cells. The latter is suggested by Xue et al. that provide data for aPC-induced production of cytokines (TGF β and IL-2) that favour Treg differentiation, maintenance and function. The assumption is indirectly also supported by Ranjan et al. themselves as they show an increase in IL-10, an important mediator of Treg suppression. The presence of IL-10, in addition to TGF- β , leads to increased expansion of Foxp3(+) iTregs with enhanced CTLA-4 expression and suppressive capability, comparable to that of natural Tregs as shown recently in human (J Immunol. 2015 Oct 15;195(8):3665-74. doi: 10.4049/jimmunol.1402898.)

Also important: The exact number and ratio of Teff/Treg/AgPC in MLR is missing.

6) The authors then turn towards the mode of action ("PAR signaling"). They confirm on protein level that

- all four PAR variants are expressed on mouse and human pan T cells
- PAR3 - only shown for human - is predominantly expressed by Treg cells.
- blocking PAR1,2 and 4 with antibodies had no effect but blocking PAR3 abolished inhibitory effect of aPC on allogeneic human T cell activation. PAR3 as a requirement for aPC function was confirmed with T cells in PAR3-/- mice.
- blocking PAR2 with a peptide but not with an antibody abolished effects of aPC on allogeneic human T cell activation

The authors conclude/suggest a heterodimerformation of PAR2/3 prerequisite for inhibitory function and assume that PAR3 is cleaved by aPC and signaling occurs via PAR2.

Comment: Why is PAR1-4, and in particular PAR2 expression not shown for Treg cells in Western blot? Why is neither PAR2 nor PAR3 expression shown for mouse Treg although most experiments are done in mouse?

The authors should explain the difference between PAR blocking with peptides and inhibitory antibodies respectively and explain why blocking PAR2 with a peptide but not with an antibody abolished effects of aPC on allogeneic human T cell activation. Fig3g. Fig3h;

Fig3h: Why was PAR3 not blocked by a peptide antagonist?

The authors suggest heterodimer formation of PAR2/3. What makes the authors assume that other PAR constellations such as formation of homodimers, co-factoring such as transactivation of PAR2 by PAR1 or the EGFR pathway,... doi: 10.1124/pr.111.004747; doi: 10.4049/jimmunol.176.2.1019) can be excluded?

Western blot experiments for PAR1-4 and PAR3 are inconsistent with regard to control (β -actin vs GAPDH).

7) To assess whether both PAR2 and 3 are necessary for aPC ameliorating effect on allo-GVHD the authors transplanted T cells and Bone marrow from allogeneic donors - a T cell subset was pretreated with PAR2-blocking peptide - prior to treatment with aPC. The authors show that - the protective effect of aPC was lost in the PAR2

blocked cohort.

Comment: Why did the authors preincubate only a part of the T cells with a PAR2 blocking antibody?

Which T cell subset was anti-PAR2 treated: CD4?, CD8?, Treg? $\gamma\delta$ T cells or panT cells? What does subset mean, the authors should specify % of CD3+ or respective subtype that was blocked, and explain why only a subset was blocked.

The description of T(aPC) + AgPC is missing in Fig.3h.

8) Next the authors transplanted T cells and Bone marrow from allogeneic donors from wt and PAR3^{-/-} mice on Balb/c and showed

- the protective effect was lost in mice transplanted with PAR3 deficient T cells.

The authors conclude PAR3 specifically on Tregs is sufficient for the GVHD protective function of aPC.

Comment: In the experiment corresponding to Fig.3j: Control groups are missing such as PAR3^{-/-} T + PAR3^{-/-} Treg(without aPC) + BM, together with or without PR2 blockade

The statement "PAR3 specifically on Tregs is sufficient for the GVHD protective function of aPC " is misleading. The authors themselves show that also PAR2 is essential for aPC ameliorating effect in GVHD, they also state that in the next sentence. That PAR3 specifically on Tregs is sufficient for aPC`s GVHD-protective function is not shown unless control cohorts with PAR2 blocked PAR3^{+/+} T cells, co-transplanted with PAR2 blocked or non-blocked PAR3^{+/+} Treg cells are performed.

9) To investigate aPC pretreatment potential for clinical settings the authors transplanted NSG-Ab{degree sign}DR4 mice transgene for human DR4 with allo CD4⁺ DR4⁻ T cells with and w/o prior aPC exposure and found

- improved survival, performance and histology in mice that had received aPC pretreated human CD4⁺ T cells.

To evaluate aPCs effect on GVL activity of alloreacting T cells the authors transplanted syngeneic MLL-AF9 Leukemia on Balb/c together with allo BM and T cells that had/had not received aPC pretreatment and found

- mice transplanted with Leukemia and aPC treated T cells showed improved survival compared to controls and comparable to recipients that had not received Leukemia
- Leukemia was cleared in individuals that had received aPC pretreated T cells and Leukemia

Comment: How many volunteers that were typed for DR4 were identified as being DR4⁻ and how many of these volunteers were used as donors for the NSG-Ab{degree sign}DR4 alloGVHD experiments?

How many cohorts of the different experiments were transplanted from identical donor(s)?

Did controls receive same donor T cells as treatment cohorts?

Did the respective DR4⁻ donors express and/or differ in their HLA alleles with respect to "GVHD-protective" alleles?

SP Fig. 9 Is the FACS plot really representative? How many experiments have been performed, with how many mice per cohort? And are all 100% free of leukemia?

The data on this experiment are not convincing. Importantly, the authors do not show the GVHD score for MLL-AF9 inoculated mice receiving or not receiving aPC treatment. 100% killing of cancer cells - which is elimination and cure of cancer - while at the same time Tregs are expanded is against all experience. Furthermore, the B6T+BM cohort, the control to whom the authors refer the survival of their GVL cohort is missing (Fig4f).

The authors should provide a mechanistical explanation for the extremely enhanced aPC mediated

anti-tumor effect on MLL-AF9 cells in the absence of GVHD.

Discussion section, In general:

The authors state in the discussion section that "methods to enrich Tregs are lacking", but they should pay attention to the fact that Treg infusion therapy is already under clinical trials
doi:10.1182/blood-2014-03-564401; 10.1182/blood-2010-07-293795; 10.1182/blood-2010-10-311894; 10.1002/cyto.a.20659; 10.1016/j.jcyt.2014.11.005

TM cannot be a direct motivation for investigating aPC nor aPC-based therapies. They need a specific reason why they focused on aPC. Indeed, TM-based therapy is already available. On the other hand, aPC-based therapy is already proven to be ineffective for sepsis or DIC.

They should cite and clarify the difference with previous articles.

PMID22447930 describes aPC induced differentiation to Tregs by up-regulating Treg-inducing-cytokines like IL-2, TGF- β , but not IL-6. This article also shows that aPC doesn't have direct activity on suppressive activity on Tregs. Importantly, and as mentioned before: that aPC directly inhibits proliferation of alloreacting T cells.

The authors should also discuss that activation of PARs increase tyrosine phosphorylation of ZAP-70 and SLP-76, two key proteins in T-cell receptor signaling (PMID: 11849313), thus T cell activation. And PMID19845798 that describes PAR2 activates T cells via DC activation.

Minor:

The date of the pathological assessment is not indicated throughout the article

The statistical significance between T(aPC)+Ag and T-PAE3-/(aPC)+Ag in Sup.Fig6. is missing

The authors should obey consistent description, NSG-AB⁺DR4 / DR4 or NSG-AB⁺DR4

Page 3: HLA-DR4-negative cells should be indicated as HLA-DR4-CD4⁺ cells corresponding to the + in CD4⁺ cells.

Page 5: "first" instead of fist (2.paragraph third line).

9.th line: purity of cells "was" instead of "were"....

11.th line: T-cell "reactivity" seems more appropriate than T cell "activation"

Page 6: line 4 and 7 throughout this paragraph: "mice" has to be changed into "mouse".

Page 8: second paragraph line 3 and 4: Balb/c instead of Balb/C;

line 7: 2 times: C57BL/6 instead of C57Bl/6;

line11: FACS "Canto" II instead of "Conto"

3.rd paragraph, line 5: P \leq 0.05 instead of P \leq 0.05

Gramatically necessary kommas are frequently omitted in several sentences.

Assessment of GVHD: I am doubtful about the validity of GVHD scoring system derived from the year 1996. Some individuals (how many?) with severest GVHD (score of >9) in the transgenic DR4 mouse were kept alive for weeks. This is ethically questionable. What were the endpoint criteria in the studies? Mice with a GVHD as shown in Fig. 4c need to be monitored carefully, at least every other day, scoring once per week is inappropriate. See also: Coding of facial expressions of pain in the laboratory mouse. Nature Methods 7, 447-449 (1 June 2010) | doi:10.1038/nmeth.1455

Reviewer #2 (Remarks to the Author):

In this manuscript by Ranjan et al, the authors suggest that activated protein C (APC) interacts with PAR2/3 co-receptors on Treg to enhance regulatory function and inhibit GVHD after BMT. The concept is highly novel, would be of high clinical interest and could be translated rapidly. However, there are major limitations with the data as presented such that the interpretations are not substantiated to an appropriate level at this point in time. Thus the observation is highly interesting but the mechanistic details are not yet robust.

Major issues:

1. There is a large amount of in vitro data regarding the effects of APC on proliferation that are not substantiated in vivo and are generally difficult to interpret. The stats used in Fig 2a where the data represent a mean and SEM of 3 expts cannot be significant unless the authors are including values from multiple wells for each exp (i.e. pseudoreplicates)? The same is true for Fig 3.
2. The authors do not exclude the possibility that their APC treatment kills T cells, either in vitro or thereafter in vivo and this should be demonstrated/excluded with appropriate staining (caspase 3/7AAD/annexin V etc).
3. The use of the iMFI where authors are multiplying freq of a positive cell population by its MFI is not helpful and the actual plots in the Supplementary data are extremely unconvincing, especially in regard to transcription factor assessment. Presentation of % pos and MFI separately would be the norm. The Suppl data are incorrectly labelled as it seems some plots (the left) are negative controls but are labelled with cytokine or transcription factors?
4. The reason for gating on FoxP3+/CD127+ populations is unclear and seems to be a extrapolation from human data where CD25+CD127- is a useful strategy in the absence of FoxP3 staining that renders cells unable to be selected or functionally interrogated. Would suggest just gating on CD4/FoxP3 with or without CD25 in the mouse system. The authors need to measure T cell subset engraftment and expansion in vivo to confirm their in vitro data regarding Treg specificity (for Fig 1 and 2) and then use appropriate FoxP3-DTR based deletion strategies to confirm Treg dependence in vivo (the Treg transfer expts do not demonstrate that this is the major effect in vivo). What does APC treatment do to Treg in vivo??
5. While the cytokine data looks more robust, it is all ex vivo with PMA/Ionomycin stimulation and event counts appear low whilst absolute numbers of cells are not given. The authors should present sera cytokine data to confirm their effects are relevant in vivo.
6. The GVL data is difficult to understand in that there is more leukemia in the non-APC treated animals that is likely a reflection of the time assayed and the fact that most of the controls have died of GVHD? Nevertheless the data are not interpretable without knowing leukemic burdens at a time point when there are equal numbers of surviving animals and presenting death from leukemia versus GVHD curves, including T cell depleted recipient controls. Ideally would undertake these experiments with low T cell doses that do not induce lethal GVHD and allow a fair comparison of GVL in both groups.
7. The picture of a clearly distressed animal with very severe GVHD in Fig 4c is inappropriate and the animal would have a clinical score of 10 (out of 10). If the experiments were approved by the ethics board what was the threshold for euthanasia or were these death as an endpoint experiments?

Minor issues

1. Fig 1a looks like n=4 per group not 12 as suggested? The skin pictures in 1c and 2e are very poor quality and not assessable. Suggest re-present.

Reviewer #3 (Remarks to the Author):

Ranjan and colleagues provide strong evidence that activated protein C alleviates GvDH by modulating the function of Treg. Specifically, they identify a novel signaling mechanism (activation of protease activated receptors Par3-Par2 heterodimers) that expands Treg and biases their function towards a more tolerogenic state.

This effect of aPC-Par signaling on adaptive immune cells, i.e. Treg, is novel, and in this preclinical study can be exploited to substantially (survival) mitigate GvDH. Given the clinical relevance/burden of GvDH in the setting of BM allografts, the current findings may be expected to provide a significant incentive for extending such studies to humans, and hence exert a substantial impact on the field.

The experimental design and approaches take full advantage of in vivo and ex vivo methods. A strength of the study is the successful effort to corroborate ex vivo findings with in vivo outcomes, and vice versa, and I feel that the data support the author's conclusions. I have only minor questions regarding results and for improving the discussion to place their data into a better context with prior work in this area:

1) introductory paragraph: does the term "endothelial-dependent protease" mean that the in vivo effect is predominantly on the vasculature? The current data, together with other studies documenting innate immune effects of aPC, seem to indicate otherwise?

2) It is striking how the 1h ex vivo pretreatment translates into such a long-term effect in vivo. Do the authors have any explanation for this?

3) It remains somewhat unclear to me whether the beneficial mechanisms of aPC action is predominantly to expand Treg, or to bias their function to a more tolerogenic state? Or both? Please clarify.

4) There seems to be some discrepancy between data shown in figures 3g and 3h: First, in 3h, I assume that the iPAR1, iPAR2, and iPAR4 column data also were T(aPC)+AgPC? If so, please add label to figure 3h as done in 3g. Second, in 3g blocking PAR2 had no effect on aPC's ability to suppress proliferation. On the other hand, all the other data clearly imply that PAR2 is necessary for this effect, such as shown in 3h. Maybe I missed something here in the experimental design?

5) In pretreatment experiments (page 4, bottom), aPC is given at 20 nM, equivalent roughly to about 1.2 microgram/mL. This seems a very high concentration compared to in vivo aPC, even in the APC transgenic mice, and is also much higher than needed in other aPC-Par1/2 signaling assays. I would encourage the authors to conduct a limited dose-response experiment in the MLR to address this issue.

6) Ibidem: In prior experiments by others (PMID 21235882), aPC at 1 microgram/mL lacked any inhibitory activity on MLR. Please discuss!

7) In fact, application of aPC as a potential treatment in the setting of allo- and xenotransplantation was indeed the subject of a series of reports published by Hancock and Bach in the last decade of the last century (summarized in the above-mentioned PMID 21235882). These studies almost exclusively focused on innate immunity, and did not pre-empt the current findings. However, it would be very illuminating to place the current work into a context with these early studies in a dedicated section in the discussion.

Reviewer #4 (Remarks to the Author):

The authors present very interesting and very original data showing that activated protein C (aPC) has the ability to reduce morbidity and mortality in a murine model of acute graft vs. host disease (GVHD). aPC suppresses proliferation of T cells that is driven by antigen presenting cells and aPC also promotes expansion of Treg cells. The paper includes some data that point towards roles for certain protease activated receptors (PAR), PAR2 and PAR3 on T cells for this effect. Overall the study is well done with a few notable exceptions that require revisions and clarifications, as noted below.

Major Points

While most of the data are compelling, the paper would benefit from substantial revisions that include updating references, revising text for great accuracy, and clarifying and correcting erroneous data presentations.

In terms of scholarly citations of relevant data, the paper is very poorly written. E.g., the introduction does not point out that this is an acute GVHD study (not chronic GVD) and the literature citations (including references) are rather outdated by papers published in the past 5 yrs. The authors state in the introductory paragraph "recent insights emphasize ..." and then cite two papers from 10 yrs ago and one from 5 yrs ago. More recent GVHD reviews (e.g., by J. Ferrara or R. Zeiser or ??) would provide the readers with more up-to-date reviews on acute or chronic GVHD.

The paper emphasizes in introduction and in final paragraph that they are studying "endothelial protective" therapies. However, the major point of the paper is that in vitro treatment of pan-T cells with a purified protease, aPC, ameliorates GVHD because aPC affects T cell properties. Except for the one study with "APChigh mice", there are no studies potentially relating to endothelium protection.

The authors also should update the paper in terms of relevant literature, including T.Lapidot group's paper reporting that aPC promotes retention of HSCs in the marrow (Gur-Cohen S, et al, Lapidot T. PAR1 signaling regulates the retention and recruitment of EPCR-expressing bone marrow hematopoietic stem cells. Nat Med. 2015 Nov;21(11):1307-17). Moreover, the directly relevant report that soluble thrombomodulin ameliorates GVHD (T. Ikezoe et al) should be cited.

Three key steps/phases of GVHD include (1) tissue damage and stress to the host, (2) donor T cell activation, differentiation and migration and (3) subsequent tissue damage. The authors interpret their results solely in the terms that aPC pretreatment of splenocytes reduces damage due to alterations of donor cells. Their studies and the discussion do not address whether aPC might alternatively or additionally affect other steps, i.e., the host status, etc. To prove that aPC affects only the donor cells due to their preincubation with aPC, studies need to be done in which the aPC not infused into recipient mice. This can be simply done by washing the cells after preincubation with aPC so that only the aPC-treated cells are infused. Otherwise, the authors need to consider that aPC also provides beneficial effects on the host itself or on the host-donor cell interactions. In fact, the authors do not discuss the results from the study with "APChigh mice" wherein constantly elevated aPC within the host is protective. It is entirely possible that endogenous elevated aPC could affect not only the transplanted T cells but also the status of the host animal and/or the ongoing interactions between donor T cells and host cells over time.

The authors should consider clarifying to a greater extent whether aPC is affecting proliferation, differentiation or both in the various in vitro assays.

There are significant problems with some data presentations. First, supplementary Fig 4 and supplementary Fig 2 contain the same identical data on the right side -- this error needs to be corrected with the correct data being presented!!! Second, Fig. 1 a presents survival data for a study using 12 animals (4 mice in 3 replicate experiments); however, the data for one group shows a plateau at 80% survival, which is not possible for a study of 12 animals. Either the data are wrong or the description of numbers of animals is wrong!

Endothelial cell protein C receptor is the major recognized receptor for aPC and generally is thought to mediate aPC's effects on PAR3 and PAR2. So what can be said about the requirement for EPCR for aPC's effects on T cells? It is an easy study to use blocking antibodies vs. EPCR in

vitro for the mixed cell cultures with easy end points. The authors should address whether EPCR is required for the effects of aPC on T cells.

Minor Comments:

1. Page No 4, line 10 in ms - Reference needs to be in correct format.
2. Figure supplementary 7 B is not very clear that IP was done in Treg cell lysate. Author may show IP in pan T cells whether hetero dimerization is in all T cells or specific for Treg.
3. Supplementary Fig 9 is a poor figure with over staining; if author can show histogram in place of FACS plot that might make things more clear about reduction of leukemic cells.
4. (also noted above) The authors should specify that they are studying acute GVHD.
5. (also noted above) The authors should cite the paper by T. Ikezoe showing that recombinant thrombomodulin alleviates GVHD (Bone Marrow Transplant 2015).
6. References 11. and 12. are the same paper, and this needs correction.

We thank the reviewers for the careful review of our manuscript NCOMMS-16-05605-T ("Activated protein C protects from GvHD *via* PAR2/PAR3 signalling in regulatory T-cells") and the constructive comments and suggestions made. All points raised by the reviewers have been taken into account, new experimental data has been generated, and the paper was rewritten in parts to address the points raised by the reviewer. We believe that the paper has improved significantly based on the new experiments conducted following the suggestions made by the reviewers.

In the following text we address the reviewers' concerns point-by-point.

Reviewer #1 (Remarks to the Author):

Comment 1

In the study "Activated protein C protects from GVHD via PAR2/PAR3 signaling in regulatory T cells" the authors Ranjan et al. investigate aPC - a protein mostly recognized for its role as an anti-coagulant with strong cyto-protective and anti-inflammatory properties - in context with amelioration of GVHD.

The authors declare two new findings. First, pretreatment of pan T cells with aPC ameliorates GVHD via an increase in the number of Treg cells that control activation/proliferation of alloreacting T cells, while not reducing GVL activity. And second, Treg suppressive function is mediated through aPC receptor signaling via heterodimer PAR2/PAR3 on Treg cells.

A cardinal failure of the study is that the authors neither mention nor show that aPC in a dose dependent manner almost completely abrogates proliferation of CD4⁺CD25^{negative} anti-CD3/CD28 activated T cells, in the absence of Treg cells (doi: 10.1074/jbc.M111.325951 Meilang Xue et al. 2012). Accordingly the design of the experiments - except for one - is not discriminating suppressive activity of Treg cells from direct effects of aPC on helper cells thus making conclusions regarding Treg activity invalid. Even the one experiment (Fig.3f) that does discriminate aPC treatment of Treg cells with regard to their suppressive capacity does not show how T cells proliferate in the absence of Treg cells, and is thus lacking a mandatory control.

No data are shown that aPC does cleave PAR3, nor is there evidence for signaling (intermediates) via PAR2. Also see below.

Response 1

We appreciate the critical appraisal and the important suggestions made by this reviewer. The manuscript was initially sent as a short report to a different Nature Journal and hence the manuscript was condensed. We now extended the information within the text, formatted the text according to the specific guidelines of Nature Communication, and added further experimental data, including new data generated following the reviewers' suggestions.

Within the manuscript we focus on the effect of aPC on T_{regs} and the receptor mechanism involved in the context of GvHD and allogenic T-cell stimulation. We concur with the reviewer that the impact of aPC on other T-cell population is an interesting aspect. Indeed, following the suggestions made by this reviewer (see comment 13) we now include new data showing that aPC does not only expand pre-existing T_{regs}, but in addition induces Tregs from CD4⁺CD25⁻ T-cells (Fig. 5b,e). We will address effects of aPC on other T-cell subpopulations in more details in future studies.

The control in Fig. 3f – as suggested by this reviewer (reactivity of effector T-cells following stimulation with antigen-presenting cells in the absence of regulatory T-cells – is now included (please see Fig. 2b and Fig. 4b of the revised manuscript). Additionally – and in agreement with the work published by Xue et al.¹ – we demonstrate that pre-incubation of effector T-cells with aPC reduces T-cell reactivity, compared to effector T-cells in the absence of regulatory T-cells (Fig. 5c of the revised manuscript). This most likely reflects an induction of T_{regs} from the effector T-cell pool (see Fig. 5e). Of note, pre-incubation of T_{regs} (CD4⁺CD25⁺) with aPC (followed by washing of T_{regs} to remove excess aPC) prior to the stimulation of effector T-cells with antigen presenting cells has a significantly stronger suppressive effect (Fig. 5c of the revised manuscript). These new insights are now discussed within the revised manuscript (please see page 9, 2nd paragraph to page 10, 2nd paragraph).

We would like to emphasize that in Fig. 3j of the original manuscript (Fig. 7c,d of the revised manuscript) we also provided evidence for a direct effect of aPC on T_{regs}. Here, *ex vivo* pre-incubation of T_{regs} with aPC (which were washed prior to injection into mice to remove excess aPC) was sufficient for the protective effect of aPC in the GvHD model, and this effect was lost if PAR3 deficient T_{regs} were used. Additional experiments have been now included in this figure to increase the number of mice per group and by including a control group in which PAR3^{-/-} T-cells were transplanted together with PAR3^{-/-} T_{regs} pretreated *ex vivo* with PBS (instead of aPC).

Regarding the activation of PAR3 by aPC we² and others³ previously demonstrated that aPC can cleave PAR3 and that PAR3-cleavage by aPC induces – via PAR-heterodimers – intracellular signalling responses. We² and others³ also demonstrated that mutations within the N-terminal end of PAR3 abolish

its cleavage by aPC and aPC-dependent signalling. These important references are now included in the text:

“Proteolytic cleavage of PAR3 by aPC has been previously reported by us and others in endothelial cells and podocytes (specialized renal epithelial cells)^{2,3}” (page 14, line 23-24).

The approach used to decipher the signalling mechanism of the PAR2/PAR3 heterodimer is now explained in more detail and a scheme illustrating the experimental approach is included in the revised version of the manuscript (Supplementary Fig. 10). Please see also the response to Comment 16 (this reviewer). Furthermore, we now include data describing an impact of aPC on PAR2 signalling intermediates (Fig. 6f):

“Signalling of aPC *via* PARs involves the MAPK pathway, which is also known to modulate T-cell function and T_{reg} differentiation⁴⁻⁸. Analyses of T-cells pre-incubated with aPC and re-isolated by MACS after 96h of MLR revealed reduced Erk1/2 and p38 phosphorylation (**Fig. 6f**). These aPC-induced changes of signalling intermediates were lost following inhibition of PAR2 signalling using the inhibitory peptide (**Fig. 6f**), corroborating the conclusion that aPC requires PAR2 for signalling in T-cells” (page 11, line 5-10).

Comment 2:

Another drawback of the study is that the cohorts are tiny: 2 mice per treatment cohort and 2 per control. Although the experiment was repeated 3 times the number of mice is too small. Since the mice are not humanized but are mouse/mouse models or transgenes engrafted with human-T cells respectively, thus there is no reason to work with such low numbers.

Response 2:

The reviewer concludes correctly that a total of 6 mice per group were used. Since the employment of 6 mice per group, analysed in three independent repeat experiments (three different time-points, three different litters, three different donors of T-cells) resulted in a very convincing and significant outcome (survival, disease scores) we feel that further experiment will not provide new insights. This approach is in accordance with the local Animal Care and Use Committee and the 3R principles⁹. In addition, these mice are not simply transgenic mice, but carry 3 ko alleles (resulting – if homozygous – in a lack of the murine *Prkdc*, the X-linked *Il2rg*, and MHC class II genes) and 1 transgene (resulting in expression of the human leukocyte antigen DR4 gene)¹⁰ and the breeding was rather cumbersome. Furthermore, we would like to point out that experiments involving these mice (receiving T-cells preincubated with aPC) and control mice (receiving T-cells preincubated with buffer only) were conducted in parallel, so that for each experiment at least 4 mice with the desired genotype were required at the same time.

Comment 3:

SEM/SD is missing in many experimental data and some statistics are not state of the art (iMFI). The authors should explain why they evaluated all frequencies by iMFI. Results should be shown in absolute numbers. SEM or SD is lacking in Fig1b, 2d, 4b, and Sup.Fig8.

Response 3:

The missing information (SEM) is now included in Fig. 1b, 3b (former 2d), 7b (former Supplementary Fig 8) and 8b (former 4b). Frequencies are no longer shown by iMFI. Instead, we provide data for frequency in percent (%) and in addition MFI (in supplements) within the revised manuscript.

Comment 4:

Format requirements:

The article is not consistent with format requirements for submission.

The abstract should contain no references.

The main text of an Article should begin with an introduction (without heading), followed by sections headed Results, Discussion and Methods.

The Results sections may be divided by topical subheadings.

Response 4:

We appreciate these comments. As stated above, the manuscript was initially sent as a short report to a different Nature Journal and hence the manuscript was written and presented in a condensed way according to the specific requirements. We used the “forward” option provided within the initial response received, in which it was stated: “It is not necessary to reformat your paper at this point”. We apologize if this caused any confusion. We now extended the information within the text, followed the specific guidelines of Nature Communication when preparing the revised manuscript, and added further experimental data, including new data generated following the reviewers’ suggestions.

Comment 5:

References inappropriate:

References are inappropriate; The abstract contains references.

The order in which citations are given is wrong, for example cited is: 6,1 or 7,5,8,9, . The font is different compared to others, valid for citation 15. One reference is not in formatted (Isermann, Weiler, 2003, first paragraph of the article). References 11 and 12 are redundant.

Response 5:

We apologize for these errors, which have been corrected in the revised manuscript version.

Comment 6:

New findings described in this paper are

1) That aPC a natural compound is able to ameliorate aggressive aGVHD which the authors show by using an APChigh strain as recipient for allo T cells. The APChigh strain experiences beneficial effects in terms of improved survival and physical appearance compared to wt strain.

2) Reduced alloreactivity in the presence of aPC is then reproduced in vitro in MLR:

- Pan T cell's robust proliferation when responding to allo Antigen-presenting-cells was diminished when aPC was administered every 12hs.

- single dose aPC pretreatment of pan T cells had exactly the same efficacy in inhibiting allogeneic T cell proliferation.

What is the author's definition of suppression? MLC with T responder cells depleted of Treg cells is the proper positive control for alloreactivity in this setting. Suppression should then be given as % proliferation of positive control.

Response 6:

We thank the reviewer for this comment. We now conducted new experiments in which we used T_{reg} depleted T-cells as positive controls (see 2b and 4b; please see also response 1 this reviewer).

In the original text version we did not use the term "suppression". Following this reviewer's suggestion we now report the inhibitory effect of aPC on T-cell reactivity as suppression in % compared to appropriate controls (e.g. T_{reg} depleted T-cells in Fig. 2b and 4b).

Comment 7:

Pretreatment of T cells with aPC was done in serum free medium, so the true control would be T cells pretreated with serum free medium without aPC.

Response 7:

We entirely concur with the reviewer and this is what we did. This was stated in the supplementary material and method section. Control T-cells were incubated with AIM V serum free medium (Life Technologies) without aPC, while T-cells preincubated with aPC were incubated with AIM V serum free medium containing 20 nM aPC (final concentration). In the revised manuscript this information has been rephrased and moved from the supplements to the main manuscript. We now state within the method section:

"For experiments with aPC-preincubation human or mouse 1×10^5 pan T-cells were preincubated with aPC (20nM) or an equal volume of PBS (control) in AIM V serum free medium (1h, 37°C)," (page 24, lines 8-10).

Comment 8:

Figure 2a) and b): neither the text nor the legend give a precise quantification of suppression (seems to be around 57% of indicated control).

Response 8:

These information are now included within the revised version of the text.

Comment 9:

Moreover the effect of aPC treatment on just allo-Antigen-presenting-cells is not done.

Response 9:

This information was indeed only shown for the human MLR (Fig. 4b; original figure draft: Fig. 3b). In the revised version we now include the effect of aPC preincubation on allo-antigen-presenting cells in

mouse MLR (Fig. 2b). Preincubation of mouse antigen-presenting cells with aPC (in AIM V serum free medium) has no effect on T-cell proliferation in the MLR. We now state within the revised text-version: “Of note, preincubation of allo-antigen presenting cells with aPC (AgPC(aPC)) following the same protocol had no effect on the MLR (**Fig. 2b**), “ (page 6, lines 21-22).

Comment 10:

Instead of showing H3 incorporation in MLC CFSE labeling seems more adequate to measure proliferation.

Response 10:

We thank the reviewer for this constructive comment. Following this suggestion we repeated a number of key experiments. Using eFlour450 labelling we observed a marked reduction of T-cell proliferation following pre-incubation of T-cells with aPC (as described in the main manuscript). Furthermore, we show that the proliferation of T_{regs} is increased following pre-incubation with aPC. These data are now included within the revised version of the manuscript (Fig. 2c, 4c and 5d). In the corresponding text passages we state:

“In addition to reducing T-cell reactivity preincubation of T-cells with aPC reduced T-cell proliferation (**Fig. 2c**)” (page 6, lines 18-19);

and

“In parallel, proliferation of aPC preincubated T-cells was markedly reduced (54.9% in aPC-preincubated T-cells vs. 67.7% in T-cells without aPC-preincubation, **Fig. 4c**)” (page 7, lines 23-24);

and

“To ascertain increased proliferation of pre-existing T_{regs} following aPC-preincubation and stimulation we isolated human T_{regs} by FACS. T_{regs} without (T_{reg}) or with (T_{reg}(aPC)) aPC-preincubation were stimulated with plate-bound anti-CD3 (10 µg/ml) and anti-CD28 (8 µg/ml). The proliferation of stimulated aPC-preincubated Tregs (T_{reg}(aPC)+CD3+CD28) was almost twice as high as that of control Tregs (T_{reg}+CD3+CD28, 36.5% vs. T_{reg}(aPC)+CD3+CD28, 65.0%, Fig. 5d)” (page 9, lines 28 to page 10, line 2).

Comment 11:

The authors then show that single dose aPC pretreatment of pan T cells prior to transplantation improves survival, physical appearance and histopathologic marker expression in an allogeneic GVHD model. There are no data about the stability/half-life of the applied aPC and or the pathway that may be (permanently) induced in the (transplanted) T cells. The authors should at least explain why they assume that a single dose of aPC on T cells can prevent GVHD in vivo while in their in vitro assays aPC is provided every 12 hs?

Response 11:

Regarding the stability of aPC *in vitro* we time-dependently determined aPC activity in AIM V serum free medium (the same medium used when pre-incubating the T-cells with aPC; Fig. R1, below). Within the first hour the activity of aPC declines by about 25%. Hence, aPC activity remains sufficiently high during the 1 hour preincubation period. We would like to emphasize that cells were carefully washed to remove aPC prior to injecting them into mice or before using them for MLR.

We entirely concur with the reviewer that the sustained effect of a single aPC-exposure is an interesting observation. Previously we demonstrated that aPC could epigenetically control gene-expression in the context of diabetic nephropathy¹¹. Accordingly, we investigated whether exposure of pan T-cells to aPC for 1 hour changes epigenetic marks in T-cells. We observed a reduction of histone 3 acetylation in T-cells preincubated with aPC (Fig. R2a,b, below). Importantly, this effect is lost in mouse T-cells lacking PAR3 (Fig. R2b, below). Thus, we hypothesise that exposure of T-cells *ex vivo* to aPC epigenetically modulates gene expression thus inducing a sustained change of T-cell function. Further studies have been initiated and we will follow up on this observation. This possibility as well as other possible mechanisms are now addressed within the discussion:

“The mechanism underlying the sustained effect of aPC on T-cells and T_{regs} remains to be fully explored. We demonstrate that aPC-preincubation increases the frequency of T_{regs} by expanding pre-existing regulatory T-cells and inducing T_{regs} from CD4⁺ T-cells. T_{regs} mainly exert their suppressive function within the first two days after transplantation and experimental depletion of T_{regs} at later time points has no major impact on GvHD in animal models¹⁰. This observation is agreement with the requirement of T_{regs} during the T_{con} priming phase for effective suppression⁶⁶. Collectively, these studies demonstrated that the presence of T_{regs} specifically during the initial days is sufficient for a long lasting protective effect in GvHD. Accordingly, we speculate that even a transient increase of T_{regs} following aPC *ex vivo* preincubation might be sufficient to increase the frequency of T_{regs} during this initial phase. Alternatively, we recently demonstrated that aPC epigenetically controls gene-expression in

the context of diabetic nephropathy⁶⁵. The importance of epigenetically controlled gene-expression for T_{reg} differentiation is well established^{67,68} and changes of histone methylation and acetylation have been observed in T-cells within few hours following stimulation^{68,69}. Hence, it appears possible, but remains to be shown, that aPC induces T_{regs} through epigenetic mechanisms” (page 16, line 20 to page 17, line 2).

Fig. R1: Half-life of aPC in AIM V serum free medium. Activated PC was added at a final concentration of 20 nM and aliquots were removed from the buffer at indicated time points as shown above. Levels of activated protein C were determined using established protocols¹². **P*<0.05, mean value ± SEM of four independent repeat experiments conducted each in triplicates, ANOVA.

Fig. R2: aPC modulates histone acetylation in T-cells. Human (a) or mouse (b) pan T-cells were left untreated or were preincubated with aPC (following the same protocol as described in the main manuscript) and stimulated with antigen presenting cells (AgPC). After 3 days acetylation of histone H3 (Ach-H3) was determined by FACS. In human (A) and mouse (B) T-cells aPC significantly reduced the frequency of Ach-H3 positive cells. The effect of aPC on H3 acetylation was lost in PAR3 deficient mouse T-cells (T-PAR3^{-/-}). Bar-graphs summarizing the results of at least 3 repeat experiments each with 3 biological disjunct replicates; Mean value ± SEM; **P*<0.05 (a: t-test; b: ANOVA).

Comment 12:

To analyze the underlying mechanism the authors characterized splenocytes from these GVHD mice d14 post transplantation and found

- a) a 2-fold increase in Treg cells in the cohort receiving aPC pretreated grafts
- b) a decrease in Th1 and TH17 cells. Th1 cytokines were less expressed in CD4⁺ cells while IL-10 producing cells were increased.

That aPC increases Treg numbers is not a new finding. Xue et al. showed that aPC in splenic cells within 3 days increased the proportion of CD4⁺Foxp3⁺ cells and Foxp3 protein (nicely shown in Western blot) for about 30%. As for the cytokine FACS analysis: Extremely divergent cell numbers are acquired for Th1 versus Th2 cell cytokine analysis (Sup. Fig2, and Sup. Fig1). It looks as if the cells for IL-17 and IL-10 analysis were pooled from different animals yet not for the Th1 cytokine FACS analysis? Moreover, the few cells acquired for IFN γ and TNF α show clear positive fractions contrasting the IL-17 and IL-10 plots that rather indicate a shift towards positivity. The authors should provide a histogram with the mean MFI and/or an ELISA for further verification of the TH2 cytokine production.

The population that clearly stains positive for Foxp3 is CD127⁻ while the CD127⁺/Foxp3 subset seems rather dim/negative for CD127. Expression of CD127 is Treg subtype specific depending on their status of activation and dependency of IL-7. Increased expression of CD127 however is described for activated Foxp3⁺, CD4⁺ Treg cells. doi:10.1002/eJi.201040531. So the cells that express CD127 in this experiment are rather activated helper cells that transiently become Foxp3^{low} positive when activated.

Why do the authors exclude CD8 T cells from analysis in general? What do the CD8 T cells secrete?
Also: Why do the authors use Biolegend Alexa Fluor 647 anti-mouse mab (Cat# 320013) in context with the FOXP3 / Transcription Factor Staining Buffer Set from eBioscience?

Response 12:

We thank the reviewer for these elaborate comments. We are well aware of the important publication by Xue et al., which we now cite within the revised manuscript. Our findings constitute a significant advance beyond the observations made by Xue et al. for various reasons. For example, we demonstrate for the first time a direct effect of aPC on regulatory T-cells, we identify the receptors involved (PAR2/PAR3), we establish that *ex vivo* incubation of T_{regs} with aPC (followed by washing of T_{regs} to remove excess aPC prior to transferring the T_{regs} into recipient mice) is sufficient for the immune-suppressive effect, and we demonstrate the relevance of these findings for allogenic T-cell stimulation and in a corresponding disease model (GvHD), which is of high medical relevance and addresses and unsolved medical need.

Regarding the analyses of cytokines by FACS we would like to clarify that we did not pool sample for FACS analyses in any case. New and more representative images of FACS analyses are now shown within the manuscript. Following this reviewer suggestion we conducted new analysis of cytokines by ELISA using LEGENDplex™ Mouse Th Cytokine Panel. These newly generated cytokine data corroborate our conclusions, which were based on FACS analysis. This new data are now shown in in Fig. 3g and 4f and within the results section we write:

“Congruently, plasma levels of IFN γ , TNF α , IL17A, and IL6 were reduced, while plasma levels of TGF β 1 and IL10 were increased in mice receiving aPC-preincubated T-cells (**Fig 3g**)” (page 7, lines 12-14);

and

“Concomitantly, expression of IFN γ , TNF α , and IL17A in T-cells was reduced, while expression of IL10 in T-cells and TGF β 1 level in the supernatant were increased (Fig. 4e,f, Supplementary Fig. 6)” (page 7, line 32 to page 8, line 2).

We concur with the reviewer that the markers used to identify regulatory T-cells are not those used conventionally. The initial decision to use CD127 was based on the paper mentioned in this comment by the reviewer (doi:10.1002/eJi.201040531), which suggested that CD127 is a marker of activated regulatory T-cells. However, we agree that the conventional markers for regulatory T-cells (CD4⁺FOXP3⁺) are more appropriate and accordingly we repeated *in vitro* and *in vivo* experiments and characterized regulatory cells as CD4⁺FOXP3⁺ cells. The new analyses had no impact on the overall results. The figures and the text have been updated accordingly.

As stated above (comment 1) we focus within the current study on CD4⁺ cells and T_{regs}. We concur with the reviewer that analysis of other T-cell populations, including CD8⁺ cells, is of relevance and we will address this interesting point in the near future.

To the best of our knowledge the FOXP3 / Transcription Factor Staining Buffer Set from eBioscience, which we used in our study, is not specific for a specific antibody and can be used in combination with antibodies from eBioscience as well as other manufacturer. The combination was chosen given the availability of the antibodies within the group, the good experience we have made with the antibodies used, and the quality of the results obtained. During the revision process we used reagents from eBioscience. This had no impact on the results.

Comment 13:

To clarify whether aPC induced Treg expansion or the differentiation of non-Treg cells the authors analyzed aPC-treated Teff- and Treg cells for their ability to expand Foxp3 cells in context with AgPC *in vitro* in MLR. They found

- only aPC-treated Treg were able to expand Treg cells.

H3 incorporation is insufficient to convincingly discriminate proliferation and expansion. Verification of proliferation recommends: isolation of (n)Tregs (CD4+CD25+ Foxp3+ CD127+RA-) from peripheral blood, CFSE-labeling and exposure to aPC, with doublings readily detectable in FACS. Furthermore: The inhibitory effect of aPC-pretreated Treg cells in MLC with aPC-untreated helper cells is dramatically reduced (Fig.3f, 21% reduction of proliferation) compared to MLC using aPC treated panT cells (suppression about 57% of control) (Fig.2a+b and Sup Fig. 6). These latter assays show a more than 2.5-fold higher suppressive activity when aPC treatment included Teff cells (57%). This either indicates that aPC directly abrogates Teff proliferation (as shown by Xue et al.) and/or that Teff are induced to differentiate into iTreg cells. The latter is suggested by Xue et al. that provide data for aPC-induced production of cytokines (TGF β and IL-2) that favour Treg differentiation, maintenance and function. The assumption is indirectly also supported by Ranjan et al. themselves as they show an increase in IL-10, an important mediator of Treg suppression. The presence of IL-10, in addition to TGF- β , leads to increased expansion of Foxp3(+) iTregs with enhanced CTLA-4 expression

and suppressive capability, comparable to that of natural Tregs as shown recently in human (J Immunol. 2015 Oct 15;195(8):3665-74. doi: 10.4049/jimmunol.1402898)

Response 13:

Following the reviewers suggestion we isolated Tregs (CD4⁺CD25⁺) from peripheral blood. FOXP3 was omitted in these experiments, as this requires permeabilization of T-cells for staining. Isolated cells were labelled with eFlour450, exposed to aPC (or control: PBS) as described in material and methods, and their proliferation was determined by FACS. This approach enables us to confirm that aPC induces proliferation of pre-existing Tregs (Fig. 5d). Within the revised text version we state:

“To ascertain increased proliferation of pre-existing T_{regs} following aPC-preincubation and stimulation we isolated human T_{regs} by FACS. T_{regs} without (T_{reg}) or with (T_{reg}(aPC)) aPC-preincubation were stimulated with plate-bound anti-CD3 (10 µg/ml) and anti-CD28 (8 µg/ml). The proliferation of stimulated aPC-preincubated T_{regs} (T_{reg}(aPC)+CD3+CD28) was almost twice as high as that of control T_{regs} (T_{reg}+CD3+CD28, 36.5% vs. T_{reg}(aPC)+CD3+CD28, 65.0%, **Fig. 5d**“ (page 9, lines 28 to page 10, line 2).

We appreciate the insightful comments made by this reviewer regarding the differential effect of aPC when preincubating only regulatory T-cells with aPC (e.g. Fig. 3e of the original manuscript draft) compared to the effect observed following aPC pre-treatment of T_{reg} depleted T cells. Following the reviewer’s suggestion we determined whether aPC-preincubation does not only expand pre-existing Tregs (as discussed in the previous paragraph) but whether aPC-preincubation additionally induces T_{regs} from CD4⁺CD25⁻ cells. Pre-incubation of CD4⁺CD25⁻ cells with aPC increased the frequency of CD4⁺CD25⁺ cells after 3 days (Fig. 5e). This observation is entirely consistent with the increased plasma levels of IL-10, TGFβ1 and the reduced IL6 plasma levels (Fig. 3g). We would like to emphasize that the suppressive effect of T_{regs} preincubated with aPC was significantly higher than that of CD4⁺CD25⁻ T-cells pre-incubated with aPC in the MLR (Fig. 5c). These new data, which were generated following the reviewer’s suggestion, are now included within the revised version of the manuscript (Fig. 5e, Supplementary Fig. 9). Accordingly, we state within the result section:

“In addition, as aPC-preincubation of effector T-cells reduced T-cell proliferation we next ascertained whether aPC-preincubation induced T_{regs} from CD4⁺CD25⁻ T-cells. To this end we determined the frequency of CD4⁺CD25⁺ cells following co-incubation of CD4⁺CD25⁻ effector T-cells with antigen presenting cells without (T_{eff}+AgPC) or with (T_{eff}(aPC)+AgPC) aPC pretreatment. After 4 days the frequency of CD4⁺CD25⁺ cells was significantly increased following preincubation of effector T-cells with aPC (T_{eff}+AgPC, 4.12% vs. T_{eff}(aPC)+AgPC, 7.61%, *P*=0.04, **Fig. 5e, Supplementary Fig. 9**“ (page 10, lines 3-9).

Furthermore, we added the following passage to the discussion:

“Based on the current results aPC expands T_{regs} through both induction of T_{regs} from CD4⁺CD25⁻ cells and expansion of pre-existing (CD4⁺CD25⁺) T_{regs}” (page 15, lines 25-26).

Taken together, these new data suggest that aPC-preincubation of T-cells does not only induce proliferation of pre-existing regulatory T-cells, but in addition induces regulatory T-cells. This observation nicely explains the different effect of aPC following pre-incubation of regulatory T-cells only or pan-T cells with aPC, which was pointed out by this reviewer.

Comment 14:

Also important: The exact number and ratio of T_{eff}/T_{reg}/AgPC in MLR is missing.

Response 14:

The number of T-cells used for the MLR reaction was given in the supplementary material and methods sections (now regular material and methods section). This ratio was kept constant in all experiments. Thus, we state:

“Human and mouse pan T-cells (1×10⁵ cells) without (PBS control) or with aPC-preincubation were incubated with 3×10⁵ irradiated allogenic non-T-cells” (page 24, lines 23-24).

The number of bone marrow cells and T-cells used for *in vivo* experiments is given in the corresponding text passages and / or figure legends. For example, in the figure legend for Fig. 1 we state:

“Recipient C57BL/6 wild-type (B6) mice or C57BL/6 mice with endogenous high levels of aPC (APC^{high}) were lethally irradiated (13Gy) and transplanted with 5×10⁶ whole bone marrow (BM) and 2×10⁶ T-cells from donor BALB/c mice (B/cT+BM)” (page 29, lines 3-5).

In addition, we now provide information about the ratio of specific T-cell subtypes in individual experiments in the corresponding figure legends. For example, in the figure legend for Fig. 5 we state:

“BALB/c mice were irradiated and transplanted with bone marrow (5×10⁶) and T_{reg} depleted T-cells (0.4×10⁶) obtained from C57BL/6 mice and Tregs (0.1×10⁶) from C57BL/6 (transgene negative DEREGL littermate mice) without (B6T+B6T_{reg}+BM+DT) or with (B6T+B6T_{reg}(aPC)+BM+DT) aPC-preincubation or from DEREGL mice without (B6T+DTR-

T_{reg}+BM+DT) or with aPC-preincubation (B6T+DTR-T_{reg}(aPC)+BM+DT)” (page 31, lines 7-11).

Comment 15:

The authors then turn towards the mode of action ("PAR signaling"). They confirm on protein level that

- all four PAR variants are expressed on mouse and human pan T cells
- PAR3 - only shown for human - is predominantly expressed by Treg cells.
- blocking PAR1,2 and 4 with antibodies had no effect but blocking PAR3 abolished inhibitory effect of aPC on allogeneic human T cell activation. PAR3 as a requirement for aPC function was confirmed with T cells in PAR3^{-/-} mice.
- blocking PAR2 with a peptide but not with an antibody abolished effects of aPC on allogeneic human T cell activation

The authors conclude/suggest a heterodimerformation of PAR2/3 prerequisite for inhibitory function and assume that PAR3 is cleaved by aPC and signaling occurs via PAR2.

Comment: Why is PAR1-4, and in particular PAR2 expression not shown for Treg cells in Western blot? Why is neither PAR2 nor PAR3 expression shown for mouse Treg although most experiments are done in mouse?

Response 15:

We appreciate the reviewer’s comment. Following the reviewer’s suggestions we now include immunoblots probing for the expression of all PARs in human and mouse Tregs (see Fig. 6a). These data demonstrate that on human T_{regs} PAR1, PAR2, and PAR3 are expressed, while on mouse T_{regs} all 4 PARs can be detected. These new results are now included in Fig. 6a and within the result section:

“Similarly, primary human and mouse pan T-cells and mouse T_{regs} express all four PARs as well as EPCR (endothelial protein C receptor), whereas human T_{regs} express EPCR and all PARs except PAR4” (page 10, lines 16-18).

Comment 16:

The authors should explain the difference between PAR blocking with peptides and inhibitory antibodies respectively and explain why blocking PAR2 with a peptide but not with an antibody abolished effects of aPC on allogeneic human T cell activation. Fig3g, Fig3h;

Response 16:

We agree with the reviewer that additional details elucidating the properties of the inhibitory antibodies and inhibitory peptides will improve the manuscript. We left this section short due to the original space restriction. The antibodies used are raised against the N-terminal end of the corresponding PARs and inhibit the generation and / or action of the tethered ligand by the corresponding proteases (in our case: aPC)^{2,4,13}. In contrast to these antibodies the inhibitory peptides block the interaction of the tethered ligand with the second extracellular loop of the receptor and hence block the binding site for the tethered ligand – or other agonists peptides (see Fig. R3 below and Supplementary Fig. 10)⁴. Hence, the inhibitory antibodies probe for the generation of the activating tethered ligand, while the blocking peptides probe for the activation of the receptor by the tethered ligand or any other peptide binding to and activating the second extracellular loop of the receptor. Accordingly, the blocking peptides inhibit not only activation by the tethered ligand derived from the same receptor, but in addition activation by other ligands, e.g. from a disjunct heterodimeric PAR-receptor (in our case the PAR3 derived tethered ligand “reaching over” to PAR2, this is referred to as “cofactoring”, as PAR3 acts as a cofactor for PAR2). In our case, the inhibitory effect of the PAR3 inhibitory antibody implies that aPC cleaves PAR3 (as previously shown in other cell-types^{2,3,13}), while the inhibitory effect of the blocking peptide reveals that signalling *via* PAR2 is required. Based on these observations we conclude that aPC cleaves PAR3, but that the tethered, PAR3 derived peptide activates PAR2 in a heterodimeric PAR2/3 complex. This concept is well-established and in agreement with previously published data^{2,4,13,14}. The interaction of PAR2 and PAR3 is demonstrated by co-immunoprecipitation (Fig. 6g).

The scheme shown below is now included in Supplementary Fig. 10. In addition, the following information was added to the corresponding result section:

“To ascertain the functional relevance of PARs and to identify which PAR may be required for aPC’s effect in the MLR we first used inhibitory antibodies. These antibodies inhibit proteolytic cleavage of the corresponding N-terminal receptor sequence and thus the generation of the corresponding tethered ligand (**Supplementary Fig. 10**). Inhibition of PAR3 cleavage abolished aPC’s effect, while N-terminal blocking antibodies to PAR1, PAR2, or PAR4 had no effect (**Fig. 6c**)” (page 10, lines 20-25);

and

“We used inhibitory peptides blocking the binding of a tethered ligand to the 2nd extracellular loop of the corresponding PAR to assess whether aPC-PAR3 signals through another PAR

(Supplementary Fig. 10). Indeed, blocking PAR2 on human T-cells with an inhibitory peptide abolished aPC's effect, while PAR1 or PAR4 inhibition had no effect (Fig. 6e)" (page 10, line 32 to page 11, line 4).

Fig. R3: Scheme presenting the approach used to determine PAR-signalling on T-cells and the proposed model. In the inactive state (1) the N-terminal end of PARs contains a tethered ligand (red) masked by a N-terminal peptide sequence (green). The receptor is inactive. Following partial proteolytic cleavage of the N-terminal end by an activating protease (red “pac-man” symbol) removes the inhibitory N-terminal peptide sequence (green) and allows binding of the tethered ligand to the 2nd extracellular loop of the receptor, resulting in signalling (2). An inhibitory antibody raised against the N-terminal end of the receptor prevents proteolytic cleavage and thus unmasking of the tethered ligand (3). Conversely, an inhibitory peptide (4) will not prevent the proteolytic unmasking of the inhibitory N-terminal peptides sequence (green) and hence allows generation of the tethered ligand (red), but this peptide will block the interaction of the tethered ligand with the receptor. The tethered ligand remains free to interact with other receptors. Proposed model of PAR2 – PAR3 interaction on regulatory T-cells. PAR-3 is cleaved by aPC, generating the PAR3-derived tethered ligand, which then “reaches over” to activate PAR2. The N-terminal end of PAR2 itself does not need to be cleaved by a protease in this model.

Comment 17:

Fig3h: Why was PAR3 not blocked by a peptide antagonist?

Response 17:

A blocking peptide for PAR3 is not available. The availability of a peptide antagonist is only possible if the receptor is signalling competent. Based on the current literature PAR3 is believed not to be signalling competent itself and accordingly no peptide antagonist has been designed or made commercially available^{4,15}.

Comment 18:

The authors suggest heterodimer formation of PAR2/3. What makes the authors assume that other PAR constellations such as formation of homodimers, co-factoring such as transactivation of PAR2 by PAR1 or the EGFR pathway,... doi: 10.1124/pr.111.004747–24064459; doi: 10.4049/jimmunol.176.2.1019- 16393989) can be excluded?

Response 18:

We appreciate this insightful comment. We did not assume or state that other receptors may not contribute. However, as PAR3 is considered to be signalling incompetent by itself but known to interact with other PARs^{2-4,13,14} we evaluated the potential interaction of PAR3 with other PARs. Following suggestions made by this and another reviewer we now also considered a role of EPCR. Of note, EPCR is required for aPC-dependent signalling in some (e.g. endothelial cells⁵), but not all (e.g. podocytes² cells). However, the newly conducted experiments revealed that blocking EPCR has not impact on the inhibitory activity of aPC in the MLR. These data are now included within the current manuscript. Thus we state within the result section:

“An antibody blocking aPC binding to EPCR did not abolish aPC’s inhibitory effect on T-cell activation (Fig. 6c), suggesting a signalling mechanism of aPC on T-cells independent of EPCR” (page 10, lines 29-31).

Furthermore, to address this point we state in the discussion:

“Further studies have been initiated to delineate the exact mechanism of PAR cofactoring on T-cells, the potential involvement of other co-receptors,...” (page 15, lines 16-18).

Comment 19:

Western blot experiments for PAR1-4 and PAR3 are inconsistent with regard to control (β -actin vs GAPDH).

Response 19:

We now used the same loading controls (GAPDH) for all immunoblots.

Comment 20:

To assess whether both PAR2 and 3 are necessary for aPC ameliorating effect on allo-GVHD the authors transplanted T cells and Bone marrow from allogeneic donors - a T cell subset was pretreated with PAR2-blocking peptide - prior to treatment with aPC. The authors show that
- the protective effect of aPC was lost in the PAR2 blocked cohort.

Comment: Why did the authors preincubate only a part of the T cells with a PAR2 blocking antibody?

Which T cell subset was anti-PAR2 treated: CD4?, CD8?, Treg? $\gamma\delta$ T cells or panT cells? What does subset mean, the authors should specify % of CD3+ or respective subtype that was blocked, and explain why only a subset was blocked.

Response 20:

We apologize for this misleading statement in the main text. As stated in the corresponding figure legend and in the methods we did not treat a specific subset of T-cells with the PAR2 blocking peptide prior to incubation with aPC, but treated pan T-cells with the PAR1 blocking peptide. This has now been corrected in the main text:

“First, pan T-cells were treated with a PAR2 blocking peptide (iPAR2) followed by incubation with aPC (iPAR2(aPC)+BM) prior to transplantation” (page 11, lines 20-21).

Comment 21:

The description of T(aPC) + AgPC is missing in Fig.3h.

Response 21:

In the revised manuscript this error has been corrected (please see Fig. 6e of the revised manuscript).

Comment 22:

Next the authors transplanted T cells and Bone marrow from allogeneic donors from wt and PAR3^{-/-}-mice on Balb/c and showed

- the protective effect was lost in mice transplanted with PAR3 deficient T cells.

The authors conclude PAR3 specifically on Tregs is sufficient for the GVHD protective function of aPC.

Comment: In the experiment corresponding to Fig.3j: Control groups are missing such as PAR3^{-/-} T + PAR3^{-/-} Treg(without aPC) + BM, together with or without PR2 blockade

Response 22:

We include now “PAR3^{-/-}-T+PAR3^{-/-}-T_{reg}+BM” (no aPC preincubation) as a control. The outcome of this group does not differ from that of the “PAR3^{-/-}-T+PAR3^{-/-}-Treg(aPC)+BM” group (see Fig. 7c,d of the revised manuscript). This supports our conclusion that aPC signals *via* PAR3 on Tregs. Within the text we state:

“To ascertain the function of PAR3 specifically on T_{regs} we separately isolated T_{regs} and T_{reg}-depleted T-cells (B6T) from C57BL/6 wt or PAR3^{-/-} mice and transplanted these together with C57BL/6 derived bone-marrow (BM) into irradiated recipient BALB/c mice. While preincubation of wt Tregs with aPC (B6T+T_{reg}(aPC)+BM) ameliorated GvHD as compared to T-cells without aPC-preincubation (B6T+Treg+BM), this protective effect was lost when using PAR-3-deficient T_{regs} preincubated with aPC (PAR3^{-/-}-T+PAR3^{-/-}-T_{reg}(aPC)+BM, Fig. 7c,d). Thus, loss of PAR3 specifically on T_{regs} is sufficient to abolish the protective effect of aPC in GvHD, supporting the requirement of PAR3 for aPC's protective effect in GvHD” (page 11, line 28 to page 12, line 3).

Based on the current understanding of PAR3-signalling^{2-4,13,14} and our *in vitro* results (Fig. 6c,e; former Fig. 3g,h) PAR3 can only convey a signal in the presence of a coreceptor, which in our case is PAR2. Accordingly, using either a PAR2 inhibitory peptide or PAR3^{-/-} Tregs *in vivo* completely abolished the protective effect of aPC and hence an additive effect cannot be expected (Fig. 7a,b, former Fig. 3i). As blocking PAR2 in the absence of PAR3 would not be expected to provide additional insights we refrained from conducting these experiments.

Comment 23:

The statement "PAR3 specifically on Tregs is sufficient for the GVHD protective function of aPC " is misleading. The authors themselves show that also PAR2 is essential for aPC ameliorating effect in GVHD, they also state that in the next sentence. That PAR3 specifically on Tregs is sufficient for aPC's GVHD-protective function is not shown unless control cohorts with PAR2 blocked PAR3^{+/+} T cells, co-transplanted with PAR2 blocked or non-blocked PAR3^{+/+} Treg cells are performed.

Response 23:

We concur with the reviewer that the statement was imprecise. We meant to state: “loss of PAR3 is sufficient to abolish the effect of aPC”. The text passage has been accordingly corrected:

“Thus, loss of PAR3 specifically on T_{regs} is sufficient to abolish the protective effect of aPC in GvHD,...” (page 12, line 1-2).

Comment 24:

To investigate aPC pretreatment potential for clinical settings the authors transplanted NSG-Ab^{degree sign}DR4 mice transgene for human DR4 with allo CD4⁺ DR4⁻ T cells with and w/o prior aPC exposure and found

- improved survival, performance and histology in mice that had received aPC pretreated human CD4⁺ T cells.

To evaluate aPCs effect on GVL activity of alloreacting T cells the authors transplanted syngeneic MLL-AF9 Leukemia on Balb/c together with allo BM and T cells that had/had not received aPC pretreatment and found

- mice transplanted with Leukemia and aPC treated T cells showed improved survival compared to controls and comparable to recipients that had not received Leukemia

- Leukemia was cleared in individuals that had received aPC pretreated T cells and Leukemia

Comment: How many volunteers that were typed for DR4 were identified as being DR4⁻ and how many of these volunteers were used as donors for the NSG-Ab^{degree sign}DR4 alloGVHD experiments?

How many cohorts of the different experiments were transplanted from identical donor(s)?

Did controls receive same donor T cells as treatment cohorts?

Did the respective DR4⁻ donors express and/or differ in their HLA alleles with respect to "GVHD-protective" alleles?

Response 24:

We screened 9 volunteers for DR4. Out of these 6 volunteers we identified as DR4⁻. Four different individuals were randomly chosen to donate blood for T-cell isolation. This information is now provided within the material and methods section:

“To isolate HLA-DRB1-04 (HLA-DR4)-negative T-cells blood samples were obtained from 9 volunteers after getting their written informed consent” (page 21, lines 23-24);

and

“Six individuals were identified as being DR4⁻”(page 22, line 4).

As stated above (comment 2) we conducted 3 experiments each with 2 mice per group (4 mice per experiment). Each of the mice in each experiment received DR4⁻ T-cells from different donors (see table below).

Experiment (No.)	Mouse	DR4 ⁻ donor
1	A1	1
	A2	2
2	B1	3
	B2	4
3	C1	2
	C2	4

Table R1: Distribution of recipient mice (two mice, A/B, each experiment) and DR4⁻ donors (denominated 1 through 4) in 3 independent repeat experiments.

The experiments were conducted in such a way that T-cells of one individual were used in parallel for the experimental (aPC preincubation) and control (buffer only) condition. Thus, for each experimental mouse receiving DR4⁻ T-cells preincubated with aPC one control mouse received T-cells from the same individual and the same preparation, but treated as a control (no aPC, AIM V serum free medium only). Hence, the mice were matched for any HLA alleles, including potentially protective alleles. Additionally, we refrained from determining any potential “GVHD-protective” alleles as we used

human cells in a mouse model and are hence reluctant to extrapolate any conclusions made for “GVHD-protective” alleles in the human system for the current experimental set up.

Comment 25:

SP Fig. 9 Is the FACS plot really representative? How many experiments have been performed, with how many mice per cohort? And are all 100% free of leukemia?

Response 25:

The FACS plot shown represented one of the better results. However, the effect observed was overall impressive and as shown in Fig. 9c (formerly Fig. 4g) the frequency of MLL-AF9 leukemic cells is overall markedly reduced. We show a summary of FACS plots below, including results from new experiments conducted during the revision (Fig. R4). Additionally, we now show two representative FACS plots in supplementary Fig. 11. The number of mice per group is now indicated in the Figure legend.

Furthermore, we analysed earlier time-points (day 7) after injection of MLL-AF9 leukemic cells to ascertain the tumour load at an earlier stage. These data demonstrate that the tumour load at an earlier stage is not different among groups, indicating that the initial engraftment of the tumour cells is comparable¹⁶. Please see also the response to the next comment.

Fig. R4: Representative FACS plots obtained when determining the leukemic load in control mice (receiving T-cells treated with AIM V serum free medium only, B6T+BM+MLL-AF9) or experimental mice (receiving T cells pretreated with aPC in AIM V serum free medium, B6T(aPC)+BM+MLL-AF9).

Comment 26:

The data on this experiment are not convincing. Importantly, the authors do not show the GVHD score for MLL-AF9 inoculated mice receiving or not receiving aPC treatment. 100% killing of cancer cells - which is elimination and cure of cancer - while at the same time Tregs are expanded is against all experience. Furthermore, the B6T+BM cohort, the control to whom the authors refer the survival of their GvL cohort is missing (Fig4f).

The authors should provide a mechanical explanation for the extremely enhanced aPC mediated anti-tumor effect on MLL-AF9 cells in the absence of GVHD.

Response 26:

We appreciate this comment. We agree with the reviewer that the protection from GvHD in our model in parallel with a sustained GvL effect is impressive. While we cannot fully explain this observation at the current time we would like to point out that similar effects – protection from GvHD paralleled by a sustained GvL effect – have been reported previously both in murine and human studies¹⁶⁻²⁰. In particular, an almost complete absence of tumour cells following the adoptive transfer of T_{regs} has been reported before¹⁷. Furthermore, using three murine models Martelli et al. ²⁰ demonstrated that the combined adoptive infusion of T_{regs} and T_{cons} prevented GvHD while maintaining the GvL effect, allowing 100% survival of mice. These authors also contemplate that “The mechanisms underlying T_{reg} suppression of GVHD with no loss of GvL activity are still obscure.” Thus, our observation is in agreement with data published by others. Further studies will be required to dissect the underlying mechanism. This aspect and the related references are now included within the discussion:

“While expansion of T_{regs} following *ex vivo* preincubation of pan T-cells with aPC protects from GvHD, it does not compromise the GvL effect. A sustained GvL effect despite suppression of GvHD has been previously reported following the adoptive transfer of T_{regs} in animal and clinical studies⁶⁻⁹. Hence, our observation is congruent with these earlier reports, but by proposing an easy, safe, and efficient way to expand donor derived T_{regs} the current study identifies an approach of potential translational relevance” (page 14, lines 7-12).

In addition, we now include the “B6T+BM” control as well as a “B6T+MLL-AF9” group. The clinical GvHD score does not allow to reliably differentiate between GvHD and tumour-disease. Hence, we interpret the tumour-load and histological changes to differentiate between GvHD and tumour-disease. These analyses strongly suggest that mice receiving T-cells preincubated with aPC

(B6T(aPC)+BM+MLL-AF9) remained free of GvHD (as compared to the “B6T+BM” group) yet maintained the GvL effect (Fig. 9). These data corroborate that preincubation of T-cells prior to transplantation ameliorates GvHD and at the same time reduces the tumour load (sustained GvL-effect) in recipient mice. Within the results section we report:

“Mice receiving BM and T-cells only (BM+B6T) displayed increased lethality (Fig. 9a) and developed typical hallmarks of GvHD, including weight loss, and GvHD was confirmed by histology (e.g. increased frequency of cryptic apoptosis in the gastro-intestinal tract). Contrary, mice receiving only BM and MLL-AF9 cells (BM+MLL-AF9) lacked morphological and histological signs of GvHD, but these mice nevertheless died as early as BM+B6T mice (Fig. 9a), presumably secondary to uncontrolled tumour growth as suggested by a slightly increased tumour load 1 week post transplantation (Fig. 9b). The tumour load, determined one week post transplantation, did not differ between mice receiving bone marrow, MLL-AF9 cells, and T-cells without (B6T+BM+MLL-AF9) or with (B6T(aPC)+BM+MLL-AF9) aPC-preincubation (Fig. 9b), indicating comparable tumour engraftment in these mice. Yet, peripheral leukemic load determined 4 weeks post transplantation was markedly reduced in B6T(aPC)+BM+MLL-AF9, but not in B6T+BM+MLL-AF9 mice (Fig. 9c, Supplementary Fig. 11), indicating a sustained GvL effect in B6T(aPC)+BM+MLL-AF9 mice. Importantly, mice receiving T-cells preincubated with aPC (B6T(aPC)+BM+MLL-AF9) did not display signs of GvHD (e.g. increased frequency of cryptic apoptosis in the gastro-intestinal tract). Accordingly, survival of mice receiving T-cells preincubated with aPC was significantly improved compared to all other groups (Fig. 9a). Hence, *ex-vivo* preincubation of T-cells with aPC ameliorates GvHD without compromising the GvL effect” (page 12, line 22 to page 13, line 7).

Comment 27:

Discussion section, In general:

The authors state in the discussion section that "methods to enrich Tregs are lacking", but they should pay attention to the fact that Treg infusion therapy is already under clinical trials doi:10.1182/blood-2014-03-564401; 10.1182/blood-2010-07-293795; 10.1182/blood-2010-10-311894; 10.1002/cyto.a.20659; 10.1016/j.jcyt.2014.11.005

Response 27:

We are aware of the studies evaluating the efficacy and clinical utility of T_{regs}. Accordingly, we did not state in the discussion: "methods to enrich T_{regs} are lacking". Rather, we stated that “methods to efficiently enrich T_{regs} have been lacking so far”.

As the reviewer states correctly, several clinical studies are underway to evaluate the clinical utility of T_{regs}. Protocols that have been used are for example associated with a prolonged period of *ex vivo* T_{reg} expansion^{21,22}. The references given by this reviewer actually support our statement, e.g by stating that “One of the major limitations to a broader clinical application of T_{reg} adoptive transfer is the difficulty in obtaining enough cells from a donor due to T_{reg} paucity in the periphery”²³ or “A current challenge that must be overcome is the isolation of pure T_{reg} populations by robust Good Manufacturing Practice (GMP)-compatible procedures and the development of highly efficient expansion protocols that lead to cell populations with stable phenotype and stable suppressive capacity”²². Other current publications likewise state that the generation of sufficient T_{regs} remains a challenge for clinical translation²⁴. We believe that our results may identify a potential mechanism to address this unmet medical need and will hence add to the on-going discussion. This issue is now addressed within the discussion:

“The protective role of T_{regs} in GvHD is well documented in animal models and has led to several clinical studies^{9,11,52,53}. For example, *ex vivo* umbilical cord blood derived T_{regs} reduced the risk of acute and chronic GvHD compared to identically treated historical controls^{53,54}. Efficient and safe methods to expand T_{regs}, as proposed in the current study, may facilitate the use of T_{regs} to combat GvHD. Both adoptive transfer of T_{regs} and induction of T_{regs} (e.g. via IL-21 blockade) provides protection from GvHD^{11,52,55}. Based on the current results aPC expands T_{regs} through both induction of T_{regs} from CD4⁺CD25⁻ cells and expansion of pre-existing (CD4⁺FOXP3⁺) T_{regs}” (page 15, lines 20-26).

Comment 28:

TM cannot be a direct motivation for investigating aPC nor aPC-based therapies. They need a specific reason why they focused on aPC. Indeed, TM-based therapy is already available. On the other hand, aPC-based therapy is already proven to be ineffective for sepsis or DIC.

Response 28:

We appreciate this comment. We and others have shown that endothelial dysfunction is intimately associated with a loss of endothelial TM (resulting in higher plasma levels of soluble TM, sTM) and

lower levels of activated protein C^{12,25,26}. Loss of TM-function has been demonstrated following whole body irradiation or chemotherapy, therapies typically used in transplant conditioning regimen²⁷⁻²⁹ and in the setting of GvHD^{30,31}. Importantly, some of the work linking elevated plasma sTM levels with other markers of endothelial dysfunction and GvHD was conducted by the authors³⁰⁻³³. In parallel, the anti-inflammatory effects of aPC are well known and substitution of aPC can compensate for the loss of aPC^{5,34}. Hence, we speculated that loss of endothelial TM-function, an established marker of endothelial dysfunction, would contribute to GvHD due to a loss of the cyto-protective aPC-functions. Collectively, these observations provide a solid rationale to ascertain the effect of aPC in the setting of GvHD. Somewhat unexpectedly we found that aPC directly modulates the adaptive immunoreaction and T_{reg} function in the setting of GvHD, linking endothelial function (reflected by TM-mediated PC-activation) with the adaptive immunoresponse and T_{reg} function in GvHD. Other effects of aPC in the setting of GvHD cannot be excluded at the current time and will be addressed independently in the future. These aspects are now discussed within the revised manuscript:

“Endothelial dysfunction, as reflected by elevated plasma levels of soluble TM and other markers, has been repeatedly demonstrated following whole body irradiation or chemotherapy, therapies typically used in transplant conditioning regimen^{16,59,60 27-29} and in the setting of GvHD¹²⁻¹⁵. Based on these findings endothelial protective therapies have been proposed to convey beneficial effects in GvHD, but their translation was only partial successful^{61,62}, probably reflecting the lack of relevant mechanistic insights. Endothelial dysfunction is intimately associated with a loss of endothelial TM (resulting in higher plasma levels of soluble TM) and lower plasma levels of aPC⁶³⁻⁶⁵. In parallel, the anti-inflammatory effects of aPC are well known and substitution of aPC can compensate for the loss of TM-dependent PC-activation^{21,26,57}. Reconstitution of aPC’s effect *ex vivo* may be a safe yet efficient approach to compensate for the inevitable impairment of endothelial- and TM-function during pre-conditioning of patients, allowing amelioration of GvHD without hampering the efficacy to eradicate residual malignant cells” (page 16, lines 6-17).

We agree with the reviewer that overall studies evaluating aPC in sepsis failed, mostly due to the increased risk of haemorrhage^{35,36}. However, aPC mutants lacking anti-coagulant properties have been generated and are being tested in clinical studies^{5,34,37}. The latter issue was already addressed in the original text version by pointing out that “new and safer aPC-based drugs“ are being developed (introduction). Furthermore, the mode of action proposed within the current manuscript is based on *ex vivo* treatment and washing of T-cells prior to transplantation. Assuming a volume of 5 ml and a final concentration of 20 nM of aPC for *ex vivo* preincubation of T-cells the final concentration of aPC – if cells were not washed to remove aPC – in the circulation would be about 20 pM, which matches the physiological concentration found in healthy individuals and is much lower than the concentrations achieved during the sepsis trials (about 40 nM)³⁸. However, as in our model T-cells are washed to remove aPC prior to infusion into the recipient (Fig. R5) and as any remaining aPC will be rapidly inactivated *in vivo* and the systemic effect of aPC can be excluded in our setting. We will conduct studies evaluating whether non-anticoagulant aPC variants can be used in the near future.

Fig. R5: Levels of aPC in the supernatant after 1 h of T-cell incubation before (left) and after (right) washing.

Comment 29:

They should cite and clarify the difference with previous articles.

PMID22447930 describes aPC induced differentiation to Tregs by up-regulating Treg-inducing-cytokines like IL-2, TGF- β , but not IL-6. This article also shows that aPC doesn't have direct activity on suppressive activity on Tregs. Importantly, and as mentioned before: that aPC directly inhibits proliferation of alloreacting T cells.

The authors should also discuss that activation of PARs increase tyrosine phosphorylation of ZAP-70 and SLP-76, two key proteins in T-cell receptor signaling (PMID: 11849313), thus T cell activation. And PMID19845798 that describes PAR2 activates T cells via DC activation.

- line11: FACS "Canto" II instead of "Conto"
- 3.rd paragraph, line 5: P ⁴²0.05 instead of P {less than or equal to}0.05
Gramatically necessary kommas are frequently omitted in several sentences.

Response 33:

We thank the reviewer for carefully reading the manuscript. These errors have been corrected.

Comment 34:

Assessment of GVHD: I am doubtful about the validity of GVHD scoring system derived from the year 1996. Some individuals (how many?) with severest GVHD (score of >9) in the transgenic DR4 mouse were kept alive for weeks. This is ethically questionable. What were the endpoint criteria in the studies? Mice with a GVHD as shown in Fig. 4c need to be monitored carefully, at least every other day, scoring once per week is inappropriate. See also: Coding of facial expressions of pain in the laboratory mouse. *Nature Methods* 7, 447-449 (1 June 2010) | doi:10.1038/nmeth.1455

Response 34:

We appreciate this comment and thank for the references provided by this reviewer. In the original text version we cited one of the original reports, in which the scoring system, which is still used by many groups, was established. Citations of the original paper are not uncommon^{43,44}. This has now been changed and we now use a recent publication⁴⁵. Even though a weekly score is shown in Fig. 4b severely sick mice were monitored once daily and the mouse shown in Fig. 4c was euthanized immediately after the picture was obtained. We agree that this picture was not appropriate and apologize for including this within the manuscript. This picture has now been removed from the manuscript.

Reviewer #2 (Remarks to the Author):

In this manuscript by Ranjan et al, the authors suggest that activated protein C (APC) interacts with PAR2/3 co-receptors on Treg to enhance regulatory function and inhibit GVHD after BMT. The concept is highly novel, would be of high clinical interest and could be translated rapidly. However, there are major limitations with the data as presented such that the interpretations are not substantiated to an appropriate level at this point in time. Thus the observation is highly interesting but the mechanistic details are not yet robust.

Comment 1:

There is a large amount of *in vitro* data regarding the effects of APC on proliferation that are not substantiated *in vivo* and are generally difficult to interpret. The stats used in Fig 2a where the data represent a mean and SEM of 3 expts cannot be significant unless the authors are including values from multiple wells for each exp (i.e. pseudoreplicates)? The same is true for Fig 3.

Response 1:

We appreciate the constructive comments and suggestions made by this reviewer. In each *in vitro* experiment we analysed three samples from three different individuals in parallel (three biological replicates per experiment). Experiments with three different biological replicates were conducted at least three times for each condition. This is now clarified within the figure legends:

“Results of at least 3 independent experiments, each containing cells from three different mice, are shown” (page 29, line 26 – 27).

In addition, new experiments were conducted during the revision work increasing in most cases the number of experiments.

Comment 2:

The authors do not exclude the possibility that their APC treatment kills T cells, either *in vitro* or thereafter *in vivo* and this should be demonstrated/excluded with appropriate staining (caspase 3/7AAD/annexin V etc).

Response 2:

We thank the reviewer for this suggestion. Since aPC is primarily known for its anti-apoptotic effect⁵ a pro-apoptotic effect seemed unlikely. Furthermore, pre-treatment of Tregs with aPC increases their frequency T_{regs} while inhibiting allogeneic T-cell activation. Accordingly, a pro-apoptotic effect seems unlikely.

Regardless, we concur that a pro-apoptotic effect cannot be ruled out and hence addressed this question. To this end apoptosis was determined by staining for annexin V and propidium iodide following the reviewer’s suggestion. aPC had no effect on T-cell apoptosis. This new data is now included:

“To assess whether aPC modulates T-cell apoptosis in the MLR T-cells without or with aPC-preincubation were stimulated with allogeneic antigen-presenting cells and stained for annexin V and propidium iodide. Preincubation of T-cells with aPC did not change T-cell apoptosis in the MLR (**Supplementary Fig. 4**)” (page 7, lines 26 – 29).

Comment 3:

The use of the iMFI where authors are multiplying freq of a positive cell population by its MFI is not helpful and the actual plots in the Supplementary data are extremely unconvincing, especially in regard to transcription factor assessment. Presentation of % pos and MFI separately would be the norm. The Suppl data are incorrectly labelled as it seems some plots (the left) are negative controls but are labelled with cytokine or transcription factors?

Response 3:

We apologize for the mistake in the supplementary data. Following the reviewers suggestion we now show percentage of positive cells and MFI separately (e.g. Fig. 3e,f and Supplementary Fig. 2,3).

Comment 4:

The reason for gating on FoxP3+/CD127+ populations is unclear and seems to be an extrapolation from human data where CD25+CD127- is a useful strategy in the absence of FoxP3 staining that renders cells unable to be selected or functionally interrogated. Would suggest just gating on CD4/FoxP3 with or without CD25 in the mouse system. The authors need to measure T cell subset engraftment and expansion *in vivo* to confirm their *in vitro* data regarding Treg specificity (for Fig 1 and 2) and then use appropriate FoxP3-DTR based deletion strategies to confirm Treg dependence *in vivo* (the Treg transfer expts do not demonstrate that this is the major effect *in vivo*). What does APC treatment do to Treg *in vivo*??

Response 4:

are now included in the manuscript, collectively support our conclusion that preincubation of T-cells with aPC ameliorates GvHD while sustaining the GvL-effect. Within the result section we report:

“Mice receiving BM and T-cells only (BM+B6T) displayed increased lethality (**Fig. 9a**) and developed typical hallmarks of GvHD, including weight loss, and GvHD was confirmed by histology (e.g. increased frequency of cryptic apoptosis in the gastro-intestinal tract). Contrary, mice receiving only BM and MLL-AF9 cells (BM+MLL-AF9) lacked morphological and histological signs of GvHD, but these mice nevertheless died as early as BM+B6T mice (**Fig. 9a**), presumably secondary to uncontrolled tumour growth as suggested by a slightly increased tumour load 1 week post transplantation (**Fig. 9b**). The tumour load, determined one week post transplantation, did not differ between mice receiving bone marrow, MLL-AF9 cells, and T-cells without (B6T+BM+MLL-AF9) or with (B6T(aPC)+BM+MLL-AF9) aPC-preincubation (**Fig. 9b**), indicating comparable tumour engraftment in these mice. Yet, peripheral leukemic load determined 4 weeks post transplantation was markedly reduced in B6T(aPC)+BM+MLL-AF9, but not in B6T+BM+MLL-AF9 mice (**Fig. 9c, Supplementary Fig. 11**), indicating a sustained GvL effect in B6T(aPC)+BM+MLL-AF9 mice. Importantly, mice receiving T-cells preincubated with aPC (B6T(aPC)+BM+MLL-AF9) did not display signs of GvHD (e.g. increased frequency of cryptic apoptosis in the gastro-intestinal tract). Accordingly, survival of mice receiving T-cells preincubated with aPC was significantly improved compared to all other groups (**Fig. 9a**). Hence, *ex-vivo* preincubation of T-cells with aPC ameliorates GvHD without compromising the GvL effect” (page 12, line 22 to page 13, line 7).

Furthermore, within the discussion we write:

“While expansion of T_{regs} following *ex vivo* preincubation of pan T-cells with aPC protects from GvHD, it does not compromise the GvL effect. A sustained GvL effect despite suppression of GvHD has been previously reported following the adoptive transfer of T_{regs} in animal and clinical studies^{16,17,20,49}. Hence, our observation is congruent with these earlier reports, but by proposing an easy, safe, and efficient way to expand donor derived T_{regs} the current study identifies an approach of potential translational relevance” (page 14, line 7 – 12).

Comment 7:

The picture of a clearly distressed animal with very severe GVHD in Fig 4c is inappropriate and the animal would have a clinical score of 10 (out of 10). If the experiments were approved by the ethics board what was the threshold for euthanasia or were these death as an endpoint experiments?

Response 7:

We concur with the reviewer’s comment and apologize for including this picture. Clearly distressed animals were monitored at least once daily and this mouse was euthanized immediately after the picture was obtained. Such deaths were counted as an endpoint in the experiments.

Comment 8:

Fig 1a looks like n=4 per group not 12 as suggested? The skin pictures in 1c and 2e are very poor quality and not assessable. Suggest re-present.

Response 8:

We thank the reviewer for bringing this to our attention. Indeed, the survival curve shown was not from the pooled data from three independent experiments but – by mistake – showed only the data from one “representative” experiment with four mice in the “B/cT+BM → B6” group. This error has now been corrected (see Fig. 1a of the revised manuscript).

New images of the skin were obtained and are now included in the revised version of the manuscript.

Taken together, these new data suggest that aPC-preincubation of T-cells does not only induce proliferation of pre-existing regulator T-cells, but in addition induces regulatory T-cells. In future studies we will characterize the underlying mechanisms in greater details.

Following the reviewer's suggestion we aimed to determine whether in addition to increasing T_{reg} frequency aPC also induces a more teleregenic state. We initiated several experiments inhibiting proliferation of Tregs with L-Mimosine or by irradiation. However, the results we obtained so far are non-conclusive and we therefore refrained from addressing this issue within the current manuscript. We will follow up on this interesting question in the near future.

Comment 5:

There seems to be some discrepancy between data shown in figures 3g and 3h: First, in 3h, I assume that the iPAR1, iPAR2, and iPAR4 column data also were T(aPC)+AgPC? If so, please add label to figure 3h as done in 3g. Second, in 3g blocking PAR2 had no effect on aPC's ability to suppress proliferation. On the other hand, all the other data clearly imply that PAR2 is necessary for this effect, such as shown in 3h. Maybe I missed something here in the experimental design?

Response 5:

We apologize for the incomplete labelling in Fig. 3h. The labelling in Fig. 3h (Fig. 6e in the revised version) has been corrected.

We agree with the reviewer that additional details elucidating the properties of the inhibitory antibodies and inhibitory peptides will improve the manuscript. We left this section short due to the original space restriction. The antibodies used are raised against the N-terminal end of the corresponding PARs and inhibit the generation and / or action of the tethered ligand by the corresponding proteases (in our case: aPC)^{2,4,13}. In contrast to these antibodies the inhibitory peptides block the interaction of the tethered ligand with the second extracellular loop of the receptor and hence block the binding site for the tethered ligand – or other agonists peptides (see Fig. R7 below and Supplementary Fig. 10)⁴. Hence, the inhibitory antibodies probe for the generation of the activating tethered ligand, while the blocking peptides probe for the activation of the receptor by the tethered ligand or any other peptide binding to and activating the second extracellular loop of the receptor. Accordingly, the blocking peptides inhibit not only activation by the tethered ligand derived from the same receptor, but in addition activation by other ligands, e.g. from a disjunct heterodimeric PAR-receptor (in our case the PAR3 derived tethered ligand “reaching over” to PAR2, this is referred to as “cofactoring”, as PAR3 acts as a cofactor for PAR2). In our case, the inhibitory effect of the PAR3 inhibitory antibody implies that aPC cleaves PAR3 (as previously shown in other cell-types^{2,3,13}), while the inhibitory effect of the blocking peptide reveals that signalling *via* PAR2 is required. Based on these observations we conclude that aPC cleaves PAR3, but that the tethered, PAR3 derived peptide activates PAR2 in a heterodimeric PAR2/3 complex. This concept is well-established and in agreement with previously published data^{2-4,13,14}. The interaction of PAR2 and PAR3 is demonstrated by co-immunoprecipitation (Fig. 6g).

The scheme shown below is now included in Supplementary Fig. 10. In addition, the following information was added to the corresponding result section:

“To ascertain the functional relevance of PARs and to identify which PAR may be required for aPC's effect in the MLR we first used inhibitory antibodies. These antibodies inhibit proteolytic cleavage of the corresponding N-terminal receptor sequence and thus the generation of the corresponding tethered ligand (**Supplementary Fig. 10**). Inhibition of PAR3 cleavage abolished aPC's effect, while N-terminal blocking antibodies to PAR1, PAR2, or PAR4 had no effect (**Fig. 6c**)” (page 10, lines 20-25);

and

“We used inhibitory peptides blocking the binding of a tethered ligand to the 2nd extracellular loop of the corresponding PAR to assess whether aPC-PAR3 signals through another PAR (**Supplementary Fig. 10**). Indeed, blocking PAR2 on human T-cells with an inhibitory peptide abolished aPC's effect, while PAR1 or PAR4 inhibition had no effect (**Fig. 6e**)” (page 10, line 32 to page 11, line 4).

Figure legend Fig. R7: Scheme presenting the approach used to determine PAR-signalling on T-cells and the proposed model. In the inactive state (1) the N-terminal end of PARs contains a tethered ligand (red) masked by a N-terminal peptide sequence (green). The receptor is inactive. Following partial proteolytic cleavage of the N-terminal end by an activating protease (red “pac-man” symbol) removes the inhibitory N-terminal peptide sequence (green) and allows binding of the tethered ligand to the 2nd extracellular loop of the receptor, resulting in signalling (2). An inhibitory antibody raised against the N-terminal end of the receptor prevents proteolytic cleavage and thus unmasking of the tethered ligand (3). Conversely, an inhibitory peptide (4) will not prevent the proteolytic unmasking of the inhibitory N-terminal peptides sequence (green) and hence allows generation of the tethered ligand (red), but this peptide will block the interaction of the tethered ligand with the receptor. The tethered ligand remains free to interact with other receptors. Proposed model of PAR2 – PAR3 interaction on regulatory T-cells. PAR-3 is cleaved by aPC, generating the PAR3-derived tethered ligand, which then “reaches over” to activate PAR2. The N-terminal end of PAR2 itself does not need to be cleaved by a protease in this model.

Comment 6:

In pretreatment experiments (page 4, bottom), aPC is given at 20 nM, equivalent roughly to about 1.2 microgram/mL. This seems a very high concentration compared to in vivo aPC, even in the APC transgenic mice, and is also much higher than needed in other aPC-Par1/2 signaling assays. I would encourage the authors to conduct a limited dose-response experiment in the MLR to address this issue.

Response 6:

We appreciate this comment. The concentration used in the current manuscript (20 nM) is a concentration typically used by several groups, and it is even lower than that used by others⁵¹⁻⁵⁵. Furthermore, we would like to point out that the concentration measured in blood does not necessarily reflect the concentration of aPC in a specific micromilieu, which may be much higher than those measured in the periphery. However, currently no reliable data determining aPC concentration in a specific micromilieu are available and accordingly this remains speculative. We will address this interesting question in more detail in future studies.

Comment 7:

Ibidem: In prior experiments by others (PMID 21235882), aPC at 1 microgram/mL lacked any inhibitory activity on MLR. Please discuss!

Response 7:

We appreciate this interesting point. We would like to point out that at a robust inhibition in the MLR was seen in the cited publication at a 5-fold higher concentration⁵⁶. We cannot exclude some variations in the activity of the aPC-preparation used by us and the other group. We would also like to point out that in the presence of protein S (PS) aPC at 1 μg/ml reduced T-cell proliferation by about 60%. As we incubated the T-cells with aPC in AIM V serum free medium, the presence of PS in the buffer can be excluded. However, we will in future studies evaluate the possibility that aPC preparation used by us may have contained some PS or that the T-cells expressed PS, as previously suggested by others⁵⁷. To address this interesting point we now include the following passage in the discussion:

“Previous studies demonstrated that the efficacy of aPC in MLR is enhanced in the presence of protein S²⁴. Protein S functions as a non-proteolytic co-factor of protein C in the context of coagulation inhibition, enhancing aPC-mediated inhibition of the coagulation factors fVa and fVIIIa. Additionally, protein S is expressed by T-cells and engages TAM receptors (Axl and Mertk) on dendritic cells to restrict the immune response⁵¹. Whether protein C and protein S co-ordinately modulate T-cell activation remains to be shown. Further studies have been

initiated to delineate the exact mechanism of PAR cofactoring on T-cells, the potential involvement of other co-receptors, and the intracellular signalling pathways underlying the mechanism of aPC induced T_{reg} expansion” (page 15, lines 11-19).

Comment 8:

In fact, application of aPC as a potential treatment in the setting of allo- and xenotransplantation was indeed the subject of a series of reports published by Hancock and Bach in the last decade of the last century (summarized in the above-mentioned PMID 21235882). These studies almost exclusively focused on innate immunity, and did not pre-empt the current findings. However, it would be very illuminating to place the current work into a context with these early studies in a dedicated section in the discussion.

Response 8:

We thank the reviewer for this valuable suggestion. As correctly pointed out by the reviewer the interesting and pioneering work of Hancock et al. focused on the effect of aPC on innate immune cells. Our findings constitute a significant advance beyond the observations made by Hancock et al. for various reasons. For example, we demonstrate for the first time a direct effect of aPC on regulatory T-cells, we identify the receptors involved (PAR2/PAR3), we establish that *ex vivo* treatment of regulatory T-cells with aPC is sufficient for the immune-suppressive effect, and we demonstrate the relevance of these findings in a clinically relevant disease model (GvHD). The work by Hancock is now included within the revised version of the manuscript (please see also the previous response):

“In a series of elegant reports Hancock et al. studied the effect of aPC in solid organ transplantation, focusing, however, on innate immune mechanisms²⁴. (page 4, line 31 to page 5, line 1).

Reviewer #4 (Remarks to the Author):

Comment 1:

The authors present very interesting and very original data showing that activated protein C (aPC) has the ability to reduce morbidity and mortality in a murine model of acute graft vs. host disease (GVHD). aPC suppresses proliferation of T cells that is driven by antigen presenting cells and aPC also promotes expansion of Treg cells. The paper includes some data that point towards roles for certain protease activated receptors (PAR), PAR2 and PAR3 on T cells for this effect. Overall the study is well done with a few notable exceptions that require revisions and clarifications, as noted below.

Response 1:

We thank this reviewer for carefully reviewing the manuscript and the supportive statement as well as the constructive comments.

Comment 2:

While most of the data are compelling, the paper would benefit from substantial revisions that include updating references, revising text for great accuracy, and clarifying and correcting erroneous data presentations. In terms of scholarly citations of relevant data, the paper is very poorly written. E.g., the introduction does not point out that this is an acute GVHD study (not chronic GVD) and the literature citations (including references) are rather outdated by papers published in the past 5 yrs. The authors state in the introductory paragraph "recent insights emphasize ..." and then cite two papers from 10 yrs ago and one from 5 yrs ago. More recent GVHD reviews (e.g., by J. Ferrara or R. Zeiser or ??) would provide the readers with more up-to-date reviews on acute or chronic GVHD.

Response 2:

We appreciate this comment. The manuscript was initially sent as a short report to a different Nature Journal and hence the manuscript was written and presented in a condensed way according to the specific requirements. We used the "forward" option provided within the response we received, in which it was stated: "It is not necessary to reformat your paper at this point". Regardless, we concur that the out-dated references need to be replaced and that more background information is required. The fact that we evaluated the role of aPC in the context of acute GvHD was mentioned in the beginning of the results section. In addition, we now clearly state within the introduction that we evaluate aspects of acute GvHD:

"Acute GvHD can be distinguished from chronic GvHD based on the timeframe and organ involvement¹. Acute GvHD, which affects up to 60% of patients, primarily affects three organ systems: the skin, the liver, and the gastrointestinal tract, and constitutes the most important risk factor for chronic GvHD²⁷" (page 4, lines 4 – 7);

and

"Considering the loss of TM in GvHD, the known cytoprotective effects of aPC, and the development of new and safer aPC-based drugs we investigated aPC's role in acute GvHD" (page 5, lines 4 – 6).

Furthermore, we now extended the information within the text, include current references, and followed the specific guidelines of Nature Communication when preparing the revised manuscript.

Comment 3:

The paper emphasizes in introduction and in final paragraph that they are studying "endothelial protective" therapies. However, the major point of the paper is that in vitro treatment of pan-T cells with a purified protease, aPC, ameliorates GVHD because aPC affects T cell properties. Except for the one study with "APChigh mice", there are no studies potentially relating to endothelium protection.

Response 3:

We acknowledge that due to the condensed form of the original text the aspect of endothelial involvement was not well presented. By discussing the role of the endothelium within the manuscript we wished to express that the cytoprotective protease activated protein C (aPC) is generated predominately by thrombomodulin (TM) expressed on endothelial cells, as shown by us in previous work⁵⁰. This endothelial function is compromised in inflammatory conditions or following whole body irradiation or other transplant conditioning regimen^{28,29}. It is well established that loss of endothelial TM function, reflected by increased levels of soluble TM in the blood, is a biomarker of endothelial dysfunction and is associated with reduced levels of aPC in the blood^{11,25,26}. Of particular relevance in the context of the current study is that loss of TM-function has been demonstrated and in the setting of GvHD^{30,31}. Some of the work linking TM, endothelial injury, and GvHD was conducted by the authors

³⁰⁻³³. The proposed role of endothelial dependent PC-activation and its impairment in the setting of GvHD, are now discussed within the revised manuscript:

“Endothelial dysfunction, as reflected by elevated plasma levels of soluble TM and other markers, has been repeatedly demonstrated following whole body irradiation or chemotherapy, therapies typically used in transplant conditioning regimen^{16,59,60 27-29} and in the setting of GvHD¹²⁻¹⁵. Based on these findings endothelial protective therapies have been proposed to convey beneficial effects in GvHD, but their translation was only partial successful^{61,62}, probably reflecting the lack of relevant mechanistic insights. Endothelial dysfunction is intimately associated with a loss of endothelial TM (resulting in higher plasma levels of soluble TM) and lower plasma levels of aPC⁶³⁻⁶⁵. In parallel, the anti-inflammatory effects of aPC are well known and substitution of aPC can compensate for the loss of TM-dependent PC-activation^{21,26,57}. Reconstitution of aPC’s effect *ex vivo* may be a safe yet efficient approach to compensate for the inevitable impairment of endothelial- and TM-function during pre-conditioning of patients, allowing amelioration of GvHD without hampering the efficacy to eradicate residual malignant cells” (page 16, lines 6-17).

Comment 4:

The authors also should update the paper in terms of relevant literature, including T.Lapidot group's paper reporting that aPC promotes retention of HSCs in the marrow (Gur-Cohen S, et al, Lapidot T. PAR1 signaling regulates the retention and recruitment of EPCR-expressing bone marrow hematopoietic stem cells. Nat Med. 2015 Nov;21(11):1307-17). Moreover, the directly relevant report that soluble thrombomodulin ameliorates GVHD (T. Ikezoe et al) should be cited.

Response 4:

We thank the reviewer for this point. These important references are now included within the revised version of the manuscript:

“Recently, Gur-Cohen et al. established a novel function of TM-dependent PC-activation for retention of hematopoietic stem cell recruitment by limiting NO-production *via* aPC-EPCR-PAR signalling⁷⁰. These and the current findings provide novel insights into the regulation of leucocyte homeostasis and function through mechanisms depending on the coagulation protease aPC. Intriguingly, the receptors targeted by aPC on hematopoietic stem cells and T-cells are partially disjunct, which may allow targeting the underlying mechanisms through distinct pharmacological approaches.” (page 17, lines 5 – 11);

and

“Targeting TM-dependent effects may hence constitute a new therapeutic approach to mitigate GvHD. Indeed, pre-clinical studies in mice suggested that soluble TM ameliorates GvHD, but the underlying mechanism remained unknown^{17”} (page 4, lines 23 – 26).

Comment 5:

Three key steps/phases of GVHD include (1) tissue damage and stress to the host, (2) donor T cell activation, differentiation and migration and (3) subsequent tissue damage. The authors interpret their results solely in the terms that aPC pretreatment of splenocytes reduces damage due to alterations of donor cells. Their studies and the discussion do not address whether aPC might alternatively or additionally affect other steps, i.e., the host status, etc. To prove that aPC affects only the donor cells due to their preincubation with aPC, studies need to be done in which the aPC not infused into recipient mice. This can be simply done by washing the cells after preincubation with aPC so that only the aPC-treated cells are infused. Otherwise, the authors need to consider that aPC also provides beneficial effects on the host itself or on the host-donor cell interactions. In fact, the authors do not discuss the results from the study with "APChigh mice" wherein constantly elevated aPC within the host is protective. It is entirely possible that endogenous elevated aPC could affect not only the transplanted T cells but also the status of the host animal and/or the ongoing interactions between donor T cells and host cells over time.

Response 5:

We appreciate this point and entirely concur with the reviewer. Due to the original space restriction we did not discuss the potential involvement of aPC in other phases of GvHD. Furthermore, we refrained from doing so as we did exactly the experiments proposed here by the reviewer (e.g. see Fig. 2-5, 7-9). Thus, we exposed T-cells *ex vivo* to aPC and then washed the cells to remove aPC before infusing the T-cells into recipient mice. Hence – and in agreement with the point made by the reviewer – we specifically evaluate the effect of aPC on donor-derived T-cells. We acknowledge that this important detail was not well presented in the original text version. This important issue is not addressed in several sections of the revised manuscript, for example:

propidium iodide. Preincubation of T-cells with aPC did not change T-cell apoptosis in the MLR (**Supplementary Fig. 4**)“(page 7, lines 26 – 29).

Comment 7:

There are significant problems with some data presentations. First, supplementary Fig 4 and supplementary Fig 2 contain the same identical data on the right side -- this error needs to be corrected with the correct data being presented!!!

Second, Fig. 1 a presents survival data for a study using 12 animals (4 mice in 3 replicate experiments); however, the data for one group shows a plateau at 80% survival, which is not possible for a study of 12 animals. Either the data are wrong or the description of numbers of animals is wrong!

Response 7:

We apologize for these errors, which have been corrected in the revised version. Indeed, the survival curve shown was not from the pooled data from three independent experiments but – by mistake – showed only the data from one “representative” experiment with four mice in the “B/cT+BM → B6” group. This error has now been corrected (see Fig. 1a of the revised manuscript).

Furthermore, we now show the correct corresponding FACS images in the supplementary figures.

Comment 8:

Endothelial cell protein C receptor is the major recognized receptor for aPC and generally is thought to mediate aPC's effects on PAR3 and PAR2. So what can be said about the requirement for EPCR for aPC's effects on T cells? It is an easy study to use blocking antibodies vs. EPCR in vitro for the mixed cell cultures with easy end points. The authors should address whether EPCR is required for the effects of aPC on T cells.

Response 8:

We thank the reviewer for this suggestion. Accordingly we determined now the expression of EPCR on T-cells, in particular on T_{reg} s. Given the expression of EPCR on T_{reg} s we then determined its functional relevance for the observed aPC dependent effect. Blocking of EPCR had not impact on aPC's effect on allogenic stimulated T-cells. This new data are now included within the revised manuscript. Thus, we state in the results section:

“An antibody blocking aPC binding to EPCR did not abolish aPC's inhibitory effect on T-cell activation (**Fig. 6c**), suggesting a signalling mechanism of aPC on T-cells independent of EPCR” (page 10, lines 29 – 31);

and within the discussion we state:

“In addition to PARs other receptors for aPC have been reported⁵⁰ and we can currently not exclude the involvement of other components within the aPC-receptorsome on T-cells. Of note, the current data suggest that EPCR, albeit being expressed on T-cells, is not required for aPC-mediated inhibition of allogenic T-cell activation. Signalling of aPC independent of EPCR has been reported in endothelial cells and in particular in non-endothelial cells^{37-39,48}. Intriguingly, aPC increases Akt phosphorylation in human leukemic monoblast U937 cells independent of EPCR³⁷ and modulates gene-expression in dendritic cells partially independent of EPCR⁴⁸, supporting the notion that aPC signalling in immune cells does not strictly dependent on EPCR” (page 15, lines 3 – 10).

Comment 9:

1. Page No 4, line 10 in ms - Reference needs to be in correct format.

Response 9:

This error has been corrected.

Comment 10:

Figure supplementary 7 B is not very clear that IP was done in Treg cell lysate. Author may show IP in pan T cells whether hetero dimerization is in all T cells or specific for Treg.

Response 10:

Following the reviewer's suggestion we now clearly state within the text that immunoprecipitation experiments were done using T_{reg} cell lysates. To confirm the interaction we conducted new experiments performing immunoprecipitation for PAR3 and detecting PAR2 in the pull down by immunoblotting. This new data are now included within the manuscript (Fig. 6g).

In addition, as suggested by the reviewer, we determined PAR3/PAR2 heterodimerization in effector T-cells ($CD25^+CD25^-$) without or with aPC-preincubation followed by allogenic

stimulation. The result obtained from immunoprecipitation of PAR3 followed by immunoblotting for PAR2 (**Fig. R8**) suggests that PAR2/PAR3 heterodimerization is induced following aPC-preincubation of effector T-cells. We will follow up on this interesting observation in the future.

Fig. R8: Analyses of PAR3/PAR2 heterodimerization on effector T-cells without or with aPC-preincubation. CD4⁺CD25⁻ effector T-cells (T_{eff}) were isolated, preincubated with aPC or control (PBS). After 1h an induction of PAR2/PAR3 heterodimers is apparent in effector T-cells preincubated with aPC.

Comment 11:

Supplementary Fig 9 is a poor figure with over staining; if author can show histogram in place of FACS plot that might make things more clear about reduction of leukemic cells.

Response 11:

An exemplary histogram for the FACS plots is now shown in Supplementary Fig. 11c and a representative selection of FACS plots is shown below (**Fig. R9**). In addition, we show two representative FACS plots in supplementary Fig. 11b within the revised manuscript. Overall, the suppressive effect of aPC not only on GvHD but also on tumour cells was impressive. We would like to point out that similar effects – protection from GvHD paralleled by a sustained GvL effect – have been reported previously both in murine and human studies¹⁶⁻²⁰. In particular, an almost complete absence of tumour cells following the adoptive transfer of T_{regs} has been reported before¹⁷. Furthermore, using three murine models Martelli et al.²⁰ demonstrated that the combined adoptive infusion of T_{regs} and T_{cons} prevented GvHD while maintaining the GvL effect, allowing 100% survival of mice. These authors also contemplate that “The mechanisms underlying T_{reg} suppression of GVHD with no loss of GVL activity are still obscure.” Thus, our observation is in agreement with data published by others. Further studies will be required to dissect the underlying mechanism. This aspect and the related references are now included within the discussion:

“While expansion of T_{regs} following *ex vivo* preincubation of pan T-cells with aPC protects from GvHD, it does not compromise the GvL effect. A sustained GvL effect despite suppression of GvHD has been previously reported following the adoptive transfer of T_{regs} in animal and clinical studies⁶⁻⁹. Hence, our observation is congruent with these earlier reports, but by proposing an easy, safe, and efficient way to expand donor derived T_{regs} the current study identifies an approach of potential translational relevance” (page 14, lines 7-12).

Fig. R9: Analyses of tumour load in mice with GvHD and injected with MLL-AF9 cells. Representative selection of FACS plots showing leukemic load in control mice (receiving T-cells treated with buffer only - BM+T+MLL-AF9) or experimental mice (receiving T cells preincubated with aPC - BM+T(aPC)+MLL-AF9).

Comment 12:

(also noted above) The authors should specify that they are studying acute GVHD.

Response 12:

It is now clarified within the introduction that we are studying acute GvHD (see comment 2 above).

Comment 13:

(also noted above) The authors should cite the paper by T. Ikezoe showing that recombinant thrombomodulin alleviates GVHD (Bone Marrow Transplant 2015).

Response 13:

As stated above (comment 4), this paper is now included within the revised manuscript version.

Comment 14:

References 11. and 12. are the same paper, and this needs correction.

Response 13:

This error has been corrected.

Reference list:

1. Xue, M., *et al.* Activated protein C enhances human keratinocyte barrier integrity via sequential activation of epidermal growth factor receptor and Tie2. *The Journal of biological chemistry* **286**, 6742-6750 (2011).
2. Madhusudhan, T., *et al.* Cytoprotective signaling by activated protein C requires protease-activated receptor-3 in podocytes. *Blood* **119**, 874-883 (2012).
3. Burnier, L. & Mosnier, L.O. Novel mechanisms for activated protein C cytoprotective activities involving noncanonical activation of protease-activated receptor 3. *Blood* **122**, 807-816 (2013).
4. Lin, H., Liu, A.P., Smith, T.H. & Trejo, J. Cofactoring and dimerization of proteinase-activated receptors. *Pharmacological reviews* **65**, 1198-1213 (2013).
5. Griffin, J.H., Zlokovic, B.V. & Mosnier, L.O. Activated protein C: biased for translation. *Blood* (2015).
6. Chang, C.F., *et al.* Polar opposites: Erk direction of CD4 T cell subsets. *Journal of immunology* **189**, 721-731 (2012).
7. Huang, H., *et al.* MAP4K4 deletion inhibits proliferation and activation of CD4(+) T cell and promotes T regulatory cell generation in vitro. *Cell Immunol* **289**, 15-20 (2014).
8. Matta, B.M., *et al.* Peri-alloHCT IL-33 administration expands recipient T-regulatory cells that protect mice against acute GVHD. *Blood* **128**, 427-439 (2016).
9. Flecknell, P. Replacement, reduction and refinement. *ALTEX* **19**, 73-78 (2002).
10. Covassin, L., *et al.* Human peripheral blood CD4 T cell-engrafted non-obese diabetic-scid IL2rgamma(null) H2-Ab1 (tm1Gru) Tg (human leucocyte antigen D-related 4) mice: a mouse model of human allogeneic graft-versus-host disease. *Clinical and experimental immunology* **166**, 269-280 (2011).
11. Bock, F., *et al.* Activated protein C ameliorates diabetic nephropathy by epigenetically inhibiting the redox enzyme p66Shc. *Proceedings of the National Academy of Sciences of the United States of America* **110**, 648-653 (2013).
12. Isermann, B., *et al.* Activated protein C protects against diabetic nephropathy by inhibiting endothelial and podocyte apoptosis. *Nature medicine* **13**, 1349-1358 (2007).
13. Stavenuiter, F. & Mosnier, L.O. Noncanonical PAR3 activation by factor Xa identifies a novel pathway for Tie2 activation and stabilization of vascular integrity. *Blood* **124**, 3480-3489 (2014).
14. Kaneider, N.C., *et al.* 'Role reversal' for the receptor PAR1 in sepsis-induced vascular damage. *Nature immunology* **8**, 1303-1312 (2007).
15. Hollenberg, M.D., *et al.* Biased signalling and proteinase-activated receptors (PARs): targeting inflammatory disease. *British journal of pharmacology* **171**, 1180-1194 (2014).
16. Edinger, M., *et al.* CD4+CD25+ regulatory T cells preserve graft-versus-tumor activity while inhibiting graft-versus-host disease after bone marrow transplantation. *Nature medicine* **9**, 1144-1150 (2003).
17. Trenado, A., *et al.* Recipient-type specific CD4+CD25+ regulatory T cells favor immune reconstitution and control graft-versus-host disease while maintaining graft-versus-leukemia. *The Journal of clinical investigation* **112**, 1688-1696 (2003).
18. Bucher, C., *et al.* IL-21 blockade reduces graft-versus-host disease mortality by supporting inducible T regulatory cell generation. *Blood* **114**, 5375-5384 (2009).
19. Di Ianni, M., *et al.* Tregs prevent GVHD and promote immune reconstitution in HLA-haploidentical transplantation. *Blood* **117**, 3921-3928 (2011).
20. Martelli, M.F., *et al.* HLA-haploidentical transplantation with regulatory and conventional T-cell adoptive immunotherapy prevents acute leukemia relapse. *Blood* **124**, 638-644 (2014).
21. Brunstein, C.G., *et al.* Infusion of ex vivo expanded T regulatory cells in adults transplanted with umbilical cord blood: safety profile and detection kinetics. *Blood* **117**, 1061-1070 (2011).
22. Theil, A., *et al.* Adoptive transfer of allogeneic regulatory T cells into patients with chronic graft-versus-host disease. *Cytotherapy* **17**, 473-486 (2015).
23. Pierini, A., *et al.* Donor Requirements for Regulatory T Cell Suppression of Murine Graft-versus-Host Disease. *Journal of immunology* **195**, 347-355 (2015).
24. Michael, M., Shimoni, A. & Nagler, A. Regulatory T cells in allogeneic stem cell transplantation. *Clinical & developmental immunology* **2013**, 608951 (2013).
25. Boehme, M.W., Galle, P. & Stremmel, W. Kinetics of thrombomodulin release and endothelial cell injury by neutrophil-derived proteases and oxygen radicals. *Immunology* **107**, 340-349 (2002).
26. Laszik, Z.G., Zhou, X.J., Ferrell, G.L., Silva, F.G. & Esmon, C.T. Down-regulation of endothelial expression of endothelial cell protein C receptor and thrombomodulin in coronary atherosclerosis. *The American journal of pathology* **159**, 797-802 (2001).
27. Wang, J., Boerma, M., Fu, Q. & Hauer-Jensen, M. Significance of endothelial dysfunction in the pathogenesis of early and delayed radiation enteropathy. *World J Gastroenterol* **13**, 3047-3055 (2007).
28. Hauer-Jensen, M., Fink, L.M. & Wang, J. Radiation injury and the protein C pathway. *Crit Care Med* **32**, S325-330 (2004).

29. Ross, C.C., *et al.* Inactivation of thrombomodulin by ionizing radiation in a cell-free system: possible implications for radiation responses in vascular endothelium. *Radiat Res* **169**, 408-416 (2008).
30. Dietrich, S., *et al.* Endothelial vulnerability and endothelial damage are associated with risk of graft-versus-host disease and response to steroid treatment. *Biology of blood and marrow transplantation : journal of the American Society for Blood and Marrow Transplantation* **19**, 22-27 (2013).
31. Andrulis, M., *et al.* Loss of endothelial thrombomodulin predicts response to steroid therapy and survival in acute intestinal graft-versus-host disease. *Haematologica* **97**, 1674-1677 (2012).
32. Luft, T., *et al.* Steroid-refractory GVHD: T-cell attack within a vulnerable endothelial system. *Blood* **118**, 1685-1692 (2011).
33. Rachakonda, S.P., *et al.* Single-Nucleotide Polymorphisms Within the Thrombomodulin Gene (THBD) Predict Mortality in Patients With Graft-Versus-Host Disease. *Journal of clinical oncology : official journal of the American Society of Clinical Oncology* **32**, 3421-3427 (2014).
34. Kerschen, E.J., *et al.* Endotoxemia and sepsis mortality reduction by non-anticoagulant activated protein C. *The Journal of experimental medicine* **204**, 2439-2448 (2007).
35. Ranieri, V.M., *et al.* Drotrecogin alfa (activated) in adults with septic shock. *The New England journal of medicine* **366**, 2055-2064 (2012).
36. Marti-Carvajal, A.J., Sola, I., Lathyris, D. & Cardona, A.F. Human recombinant activated protein C for severe sepsis. *Cochrane Database Syst Rev*, CD004388 (2012).
37. Lyden, P., *et al.* Phase 1 safety, tolerability and pharmacokinetics of 3K3A-APC in healthy adult volunteers. *Current pharmaceutical design* **19**, 7479-7485 (2013).
38. Liaw, P.C., *et al.* Patients with severe sepsis vary markedly in their ability to generate activated protein C. *Blood* **104**, 3958-3964 (2004).
39. Bar-Shavit, R., *et al.* Signalling pathways induced by protease-activated receptors and integrins in T cells. *Immunology* **105**, 35-46 (2002).
40. Hansen, K.K., Saifeddine, M. & Hollenberg, M.D. Tethered ligand-derived peptides of proteinase-activated receptor 3 (PAR3) activate PAR1 and PAR2 in Jurkat T cells. *Immunology* **112**, 183-190 (2004).
41. Ramelli, G., *et al.* Protease-activated receptor 2 signalling promotes dendritic cell antigen transport and T-cell activation in vivo. *Immunology* **129**, 20-27 (2010).
42. Bennett, C.M., Guo, M. & Dharmage, S.C. HbA(1c) as a screening tool for detection of Type 2 diabetes: a systematic review. *Diabetic medicine : a journal of the British Diabetic Association* **24**, 333-343 (2007).
43. Kim, B.S., *et al.* Treatment with agonistic DR3 antibody results in expansion of donor Tregs and reduced graft-versus-host disease. *Blood* **126**, 546-557 (2015).
44. He, S., *et al.* Inhibition of histone methylation arrests ongoing graft-versus-host disease in mice by selectively inducing apoptosis of alloreactive effector T cells. *Blood* **119**, 1274-1282 (2012).
45. Rowe, V., *et al.* Host B cells produce IL-10 following TBI and attenuate acute GVHD after allogeneic bone marrow transplantation. *Blood* **108**, 2485-2492 (2006).
46. Simonetta, F., *et al.* Increased CD127 expression on activated FOXP3+CD4+ regulatory T cells. *European journal of immunology* **40**, 2528-2538 (2010).
47. Lahl, K., *et al.* Selective depletion of Foxp3+ regulatory T cells induces a scurfy-like disease. *The Journal of experimental medicine* **204**, 57-63 (2007).
48. Lahl, K. & Sparwasser, T. In vivo depletion of FoxP3+ Tregs using the DEREK mouse model. *Methods in molecular biology* **707**, 157-172 (2011).
49. Jones, S.C., Murphy, G.F. & Korngold, R. Post-hematopoietic cell transplantation control of graft-versus-host disease by donor CD425 T cells to allow an effective graft-versus-leukemia response. *Biology of blood and marrow transplantation : journal of the American Society for Blood and Marrow Transplantation* **9**, 243-256 (2003).
50. Isermann, B., *et al.* Endothelium-specific loss of murine thrombomodulin disrupts the protein C anticoagulant pathway and causes juvenile-onset thrombosis. *Journal of Clinical Investigation* **108**, 537-546 (2001).
51. Bezhly, M., *et al.* Role of activated protein C and its receptor in inhibition of tumor metastasis. *Blood* **113**, 3371-3374 (2009).
52. Han, M.H., *et al.* Proteomic analysis of active multiple sclerosis lesions reveals therapeutic targets. *Nature* **451**, 1076-1081 (2008).
53. Riewald, M. & Ruf, W. Protease-activated receptor-1 signaling by activated protein C in cytokine-perturbed endothelial cells is distinct from thrombin signaling. *The Journal of biological chemistry* **280**, 19808-19814 (2005).
54. Schuepbach, R.A., Madon, J., Ender, M., Galli, P. & Riewald, M. Protease-activated receptor-1 cleaved at R46 mediates cytoprotective effects. *Journal of thrombosis and haemostasis : JTH* **10**, 1675-1684 (2012).

55. Schuepbach, R.A., Feistritzer, C., Brass, L.F. & Riewald, M. Activated protein C-cleaved protease activated receptor-1 is retained on the endothelial cell surface even in the presence of thrombin. *Blood* **111**, 2667-2673 (2008).
56. Hancock, W.W. & Bach, F.H. Immunobiology and therapeutic applications of protein c/protein s/thrombomodulin in human and experimental allotransplantation and xenotransplantation. *Trends in cardiovascular medicine* **7**, 174-183 (1997).
57. Carrera Silva, E.A., *et al.* T cell-derived protein S engages TAM receptor signaling in dendritic cells to control the magnitude of the immune response. *Immunity* **39**, 160-170 (2013).

Reviewers' comments:

Reviewer #1 (Remarks to the Author):

As you noticed, the paper still has numerous careless mistakes.

1. comment1

The previous journal we kindly and clearly suggested the authors to reference is;

J Biol Chem. 2012 May 11;287(20):16356-64. doi: 10.1074/jbc.M111.325951. Epub 2012 Mar 23.

Activated protein C inhibits pancreatic islet inflammation, stimulates T regulatory cells, and prevents diabetes in non-obese diabetic (NOD) mice.

Xue M1, Dervish S, Harrison LC, Fulcher G, Jackson CJ.

not the journal the authors cited as reference list 1 (actually cited as ref. 25 in the main paper). The authors completely negotiate this important work.

2. comment 11

Even if aPC alters the epigenetic state of T cells, the authors should explain why the treatment schedule is completely differed between in vivo and in vitro study.

3. comment 12

The authors demonstrated that cytokine profiles contributed to the differentiation from Teff to Treg by the direct effect on T cells. However, do they insist that these Treg-inducible cytokines (TGF- β without IL-6) derive from T cells? The authors showed the in vivo data but the main source of TGFB was not analyzed.

4. comment 34

we are discussing the ethical issue, not the picture. I highly doubt that these severely affected mice were ethically approved.

5. FACS plots regarding T-bet, ROR γ t, Foxp3 are not convincing. (Sup. fig2b, 5b) as are FACS plots regarding IL-17A, IL-10, IFN γ , TNF α (sup fig3b, 6b).

6. Fig 4b. T-Tr (aPC) + AgPC is missing.

7. Fig. 5c Teff(aPC) + Treg(aPC) + AgPC is missing.

8. Fig5a; the authors should explain why (B6T + B6Treg + BM + DT) did not have any impact on the survival even if these mice received Tregs.

9. Abstract

The authors should not conclude "The protective effect of aPC on GVHD does not compromise the GVL effect" merely by the experiment with a single tumor cell line.

Overall, the authors have made a courteous effort on revising, however, because the new insight by this paper is only a combination of the previous paper by Xue (aPC and Treg) and the authors (aPC and PAR3), I must say this paper is not suitable for Nature Communication.

Reviewer #2 (Remarks to the Author):

The authors have answered all questions, provided significant additional data and the paper is well written, the findings are novel and the data now appear robust.

Reviewer #3 (Remarks to the Author):

1)The response to my comment 5 does not address my comment, but rather is a repeat (copy/paste) of a preceding section.

2) the response to my comment 6 (request to conduct dose response) is not addressed.

Reviewer #4 (Remarks to the Author):

This manuscript is not really a revised manuscript but rather is a new manuscript based on its huge expansion in both size and extensively new sets of data. The current paper shows that aPC pretreatment of donor T cells reduces graft versus host disease (GVHD) in a murine model. The current paper shows that this is associated with a notable expansion of Tregs. Data suggested the increase of Tregs was needed for reduction of GVHD. Furthermore, in mixed human lymphocyte cultures, aPC treatment of pan T-cells augments Tregs and limits Th1 and Th17 cells. In more than 30 pages single-spaced text of Responses to Reviewers, the authors have attempted to address all major points for the previous review of the previous interesting but poorly executed manuscript. Overall, the authors have addressed satisfactorily many previous weaknesses of their paper. However, notable major points remain that are presented below.

Major Points

1. The data in Fig. 7 used to implicate a requirement for murine PAR2 for aPC's effects involved "a PAR2 blocking peptide (FSLLRY-NH₂)" (cited as coming from refs #75,76). However, this peptide does not have a convincing track record that it is really a definitive PAR2 blocker in any sense of critical pharmacology. It was best described actually by Al-ani B et al, Hollenberg J Pharmacol Exp Therapeutics, 2002) and it was shown to inhibit trypsin-initiated signaling but not by the PAR2 agonist peptide, SLIGRL-amide. Hence, it does not block PAR2-specific signaling initiated by a PAR2 peptide agonist (SLIGRL-amide). Moreover, when this peptide was used in ref. # 75, a negative control peptide with the reverse sequence, YRLLSF, was used. The authors here fail to provide a negative control peptide and show it has no effect; this is a standard requirement when using peptides as inhibitors in any critically designed experiment. . The peptide, FSLLRY is a modified PAR1 sequence, so its effects might actually inhibit some PAR2-PAR1 crosstalk or PAR1/PAR3 crosstalk rather than simply inhibiting PAR2 antagonism by PAR3. Thus, the authors fail to provide strong and clear evidence that murine PAR2 is required for aPC's effects on murine T cells and GVHD. Better reagents, with positive and negative controls, are needed to make the claims of Fig. 7 relating to PAR2's role or requirement.

2. The mechanistic scheme in Suppl. Fig. 10 for the mechanism of aPC's protection against GVHD as due to PAR3 cleavage followed by binding of the newly generated PAR3 tail to PAR2 is thus not demonstrated at all for the in vivo mechanism of action (MOA) for reducing GVHD, invalidated any conclusions for in vivo MOA. It might more likely be the case that in vivo, the protection against GVHD is due to PAR3/PAR1 interactions and signaling, as in podocytes (ms.ref. #39/Responses.ref.#2, Madhusudhan et al, Isermann). Thus, considering the known differences between murine and human PAR3 signaling, the mechanistic scheme of Suppl Fig. 10 is not justified for in vivo GVHD prevention by the available data.

3. Based on the point of view that data for PAR2 role in aPC's protection in vivo vs. GVHD are not at all definitive, the Title is simply a misleading, very possibly erroneous, statement because clear data are lacking for aPC-induced signaling via PAR2/PAR3 in mice for protection against GVHD (the Title's message).

The problem is not an easy one to solve. The challenge of defining mechanisms for PAR3-dependent crosstalk involving PAR3-PAR2 or PAR3-PAR1 interactions was highlighted by these authors (ms.ref. #39/Responses.ref.#2, Madhusudhan et al, Isermann) where they reported that in the case of aPC's protective actions on podocytes, human cells used PAR3/PAR2 crosstalk (with data for their coprecipitation) whereas in murine cells, aPC used PAR3/PAR1 crosstalk (with data for their coprecipitation). In those previously published studies, the authors stated that there was "plasticity of aPC mediated cytoprotection." So what is going on for T cells and for aPC's effects on human cells vs. aPC's effects on murine cells and in vivo in mice? It is very possible that this GVHD model involves aPC's actions via PAR3/PAR2 for human T cells but PAR3/PAR1 for murine T cells. Further work is needed to clarify these possibilities.

4. The authors should address the challenging question of how does aPC cleave and activate murine PAR3, in discussion and in experiments? A key paper cited (ms.ref. #49/Responses.ref.#3, Burnier & Mosnier) for aPC's cleavage of PAR3 proves that aPC cleaves human PAR3 at Arg41, not at the Lys39 thrombin canonical cleavage site. However, murine PAR3 lacks this Arg residue although it has a similar Lys cleavage site for thrombin. So there is no evidence, as far as this reviewer knows in the literature, which show whether and where aPC cleaves murine PAR3. Clearly the authors have data for murine cells from PAR3 knockout mice that PAR3 is required for aPC's cytoprotection but no data for how aPC actually cleaves murine PAR3 to effect cytoprotection. So the scheme for mechanism of action in Suppl. Fig. 10 as that which explains GVHD effects seems premature without ore data for cleavage of murine PAR3 by aPC. It is demonstrated here that T cells from PAR3 knockout mice do not show aPC's beneficial effects, indicating PAR3 is required. But PAR3 mechanism schemes for in vivo mechanisms are not yet very well justified.

Reviewer 1:

We thank this reviewer for conducting a critical review of the revised manuscript and for the constructive comments and suggestions.

Comment 1 (follow up previous comment 1)

The previous journal we kindly and clearly suggested the authors to reference is;

J Biol Chem. 2012 May 11;287(20):16356-64. doi: 10.1074/jbc.M111.325951. Epub 2012 Mar 23.
Activated protein C inhibits pancreatic islet inflammation, stimulates T regulatory cells, and prevents diabetes in non-obese diabetic (NOD) mice.
Xue M1, Dervish S, Harrison LC, Fulcher G, Jackson CJ.

not the journal the authors cited as reference list 1 (actually cited as ref. 25 in the main paper). The authors completely negotiate this important work.

Response 1:

We thank the reviewer for bringing this important publication again to our attention and we apologize that we did not emphasize the work by Xue et al. in the response to comment 1 by this reviewer. Considering that several aspects were raised in comment 1 we discussed the work by Xue et al. in detail within the response to comment 12 of this reviewer in the previous response letter. We also cited the important work by Xue et al. within the manuscript (reference 25).

As we stated within our previous response we believe that our findings constitute a significant advance beyond the important observations made by Xue et al. Xue et al. demonstrate an increase of T_{regs} in NOD.SCID mice following injecting of aPC and following treatment of spleen cells from NOD.SCID mice *in vitro*. However, the mechanism through which aPC induces T_{regs} remained unclear and the authors conclude in their discussion that “the exact mechanisms require further investigation”. To address this unresolved question we focus on mechanistic studies of T-cell activation using *in vitro* studies with primary human and mouse cells and *in vivo* models. These studies are conducted in the context of allogeneic T-cell activation, an aspect that was not addressed by Xue et al. Within the current study we demonstrate for the first time that aPC conveys a direct effect on regulatory T-cells, increasing their number and hence promoting a tolerogenic response. By depletion of T_{regs} we demonstrate that T_{regs} are essential for the aPC induced tolerogenic response. In addition, we demonstrate that the receptors PAR2 and PAR3, expressed by T-cells, are required for aPC's effect on regulatory T-cells. We use knock out mice for both PAR2 and PAR3 (see Fig. 8 of the revised manuscript) and hence are confident that these receptors are required for the effect of aPC on T_{regs} and the induction of a tolerogenic response. The question which receptors on T-cells are required for the modulation of T-cell-function by aPC was not addressed by Xue et al. We also identify the cleavage site of aPC in the N-terminal end of PAR3. Furthermore, we demonstrate that *ex vivo* incubation of T_{regs} with aPC (followed by washing of T_{regs} to remove excess aPC prior to transferring the T_{regs} into recipient mice or prior to conducting MLR-experiments) is sufficient for the immune-suppressive effect, which is of high potential translational relevance. This possibility was not explored by Xue et al. Finally, we show the importance of these findings for allogeneic T-cell stimulation and in a corresponding disease model (GvHD), which is of high medical relevance and addresses an unsolved medical need. Thus, the current study provides new mechanistic insights of potential translational relevance and thus adds substantial novelty beyond what has been shown in the important study by Xue et al.

Following the comments made by this reviewer we state in the introduction of the revised version:

“Additionally, previous work showed that aPC dampens activation of effector T-cells and increases the frequency of T_{regs} in a model for type 1 diabetes mellitus (non-obese diabetic (NOD) mice), but the underlying mechanism, e.g. which immune cell type is targeted by aPC and the receptors involved, remained unknown²⁵” (page 4, line 31 to page 5 line 2).

We now added a new passage within the discussion to acknowledge the important work by Xue et al. and to put their findings into the context of our current findings:

“An effect of aPC on T_{reg} expansion was previously reported by Xue et al. in NOD mice in the context of pancreatic islet inflammation⁵⁰. However, the underlying mechanisms, e.g. the immune cells targeted by aPC and the receptors required for aPC's effect, remained unknown” (page 15, line 17 to 19).

As stated above we believe that the current work significantly advances the knowledge in the field and has implications for future translational research.

Comment 2 (follow up previous comment 11)

Even if aPC alters the epigenetic state of T cells, the authors should explain why the treatment schedule is completely differed between *in vivo* and *in vitro* study.

Response:

We are not quite sure why the reviewer believes that “the treatment schedule is completely different between *in vivo* and *in vitro*“ studies. We use two parallel approaches both *in vivo* and *in vitro*. *In vivo* we first used APC^{high} mice, which express a mutant PC that can be efficiently activated by thrombin even in the absence of thrombomodulin, resulting in elevated plasma levels of aPC¹. The corresponding data are shown in Fig. 1. This situation mimics persistent elevated aPC levels *in vivo*. In addition, we transplant T-cells, which have been pretreated with aPC prior to transplantation. The corresponding data are shown in Fig. 3. This situation reflects a one-time treatment of T-cells with aPC prior to transplantation and hence prior to allogeneic stimulation. In this case T-cells were pretreated with aPC, which was used at a concentration of 20 nM, for 1 h at 37°C in AIM V serum free medium (methods section, page 22, lines 8-10).

Similarly, we use two approaches *in vitro*. First, we conducted mixed lymphocyte reactions in the presence of aPC. In this case aPC was added at a final concentration of 20 nM ever 12 h. The corresponding data are shown in in Fig. 2a. Second, we preincubated T-cells with aPC, which – following the same approach as *in vivo* – was used at a concentration of 20 nM for 1 h at 37°C in AIM V serum free medium (method section, page 24, line 32 to page 25 line 7). This approach mimics the *ex vivo* pretreatment of T-cells with aPC prior to transplantation.

Hence, the preincubation protocol for T-cells was identical for the *in vivo* and *in vitro* situation. According to the experimental design proliferation of T-cells has to be analysed after 96 h *in vitro*, which is a well-established approach when conducting MLR²⁻⁵. To determine the impact of aPC *in vivo* later time-points were used. Either animals were followed up and monitored for GvHD, or animals were sacrificed 2 weeks post transplantation to determine the impact of T-cell pretreatment by aPC on T-cell sub-populations and cytokines. This is in agreement with established approaches⁶⁻⁹. These experimental details are given in the material and methods section.

In summary, we use two very similar approaches both *in vitro* and *in vivo*, which were only slightly modified to meet the specific experimental requirements in the *in vitro* and *in vivo* situation.

Comment 3 (follow up previous comment 12)

The authors demonstrated that cytokine profiles contributed to the differentiation from Teff to Treg by the direct effect on T cells. However, do they insist that these Treg-inducible cytokines (TGF-b without IL-6) derive from T cells ? The authors showed the *in vivo* data but the main source of TGFb was not analyzed.

Response:

We appreciate this comment. We did not claim that the cytokines promoting T_{reg} induction are only derived from T-cells. As stated by the reviewer, we observed altered expression of cytokines in donor CD4⁺ T-cells (Fig. 3f and corresponding text passage, page 7, lines 10 – 12). In addition, we observed congruent changes of cytokines in the plasma (Fig. 3g and corresponding text passage, page 7, lines 12 – 14). Similar observations were made in human

T-cells in the MLR (Fig. 4e,f). We concur with the reviewer that different cytokine sources may contribute to the plasma cytokines.

We agree with the reviewer, that it is an interesting question, which cells contribute to the cytokine profile promoting T_{reg} induction, and we will follow up on this in the future. To clarify that the cytokines promoting T_{reg} induction are not necessarily derived exclusively from T-cells we now added the following passage to the text:

“While the plasma cytokine profile in mice is in agreement with the cytokine expression pattern observed in murine donor CD4⁺ T-cells, it is likely that cells other than T-cells contributed to the plasma cytokine profile” (page 7, lines 14-16).

Comment 4 (follow up previous comment 34)

we are discussing the ethical issue, not the picture. I highly doubt that these severely affected mice were ethically approved.

Response:

As stated in our previous response we concur with the reviewer. Severely sick mice were immediately euthanized after noticing their status.

Comment 5

FACS plots regarding T-bet, ROR γ t, Foxp3 are not convincing. (Sup. fig2b, 5b) as are FACS plots regrading IL-17A, IL-10, IFN γ , TNFa (sup fig3b, 6b).

Response:

We appreciate this comment. Following the reviewer’s suggestion we internally reviewed the available data (S.R., J.H., M.B-W., B.I.). In addition, we conducted new experiments to validate our earlier observations. New FACS plots and analyses are now provided within the manuscript.

Comment 6

Fig 4b. T-Tr (aPC) + AgPC is missing.

Response:

We thank the reviewer for bringing this to our attention. Following the reviewers suggestion we now include T-cells depleted of T_{regs} (T-Tr), which were preincubated with aPC. Preincubation was conducted following the same protocol as outlined above (reviewer 1, comment 2) and in the methods section (page 25, lines 4-7). Preincubation of CD4⁺ T-cells depleted of T_{regs} with aPC has a suppressive effect compared to CD4⁺ T-cells depleted of T_{regs} (T-Tr) without aPC preincubation. However, the effect is significantly less than that observed following preincubation of T-cells with aPC (83% proliferation vs. 53% of proliferation as compared to the control: T-Tr without aPC pre-treatment). The partial effect is entirely congruent with the effect observed when preincubating human T_{eff} with aPC (Fig. 5c) and with the observed induction of T_{regs} (Fig. 5d-f). These new data are now added to Fig. 4b and within the text we added the following text passage:

“Of note, preincubation of human T_{reg}-depleted pan T-cells with aPC ((T-Tr)(aPC)+AgPC) partially reduced T-cell reactivity (83% vs. 100% ³H incorporation in (T-Tr)+AgPC, *P*=0.016, Fig. 4b). The latter indicates that aPC’s inhibitory effect in the MLR is only partially dependent on pre-existing T_{regs}“(page 7, line 29 – page 8, line 1).

Comment 7:

Fig. 5c T_{eff}(aPC) + T_{reg}(aPC) + AgPC is missing.

Response:

We thank the reviewer for this suggestion. We now conducted new experiments to evaluate the effect when independently preincubating T_{eff}s and T_{regs} with aPC prior conducting the MLR. The results for the (T_{eff}(aPC) + T_{reg}(aPC) + AgPC) group is now included (Fig. 5c). Congruent with the above data preincubation of T_{eff} and T_{regs} results in further reduction of T-

cell proliferation as compared to preincubation of T_{regs} only. This difference, however, does not reach statistical significance (42% vs. 27%; $P=0.11$). These new data are now included in Fig. 5c and within the results section we state:

“Preincubation of both T_{eff} and T_{reg} separately with aPC ($T_{\text{eff}}(\text{aPC})+T_{\text{reg}}(\text{aPC})+\text{AgPC}$) suppressed T-cell reactivity more than preincubation of either T_{eff} or T_{reg} only, but this effect was not significantly different from that observed following aPC preincubation of T_{regs} only (42% vs. 27%; $P=0.11$; **Fig. 5c**)” (page 10, lines 2-5).

Comment 8:

Fig5a; the authors should explain why (B6T + B6Treg + BM + DT) did not have any impact on the survival even if these mice received Tregs.

Response:

We appreciate this comment. Publications in which an effect of T_{regs} in GvHD is shown typically use a T_{reg} proportion of 50%, but not below 33%¹²⁻¹⁶. Taking this information into consideration we titrated the number of T_{regs} used in our experiments to levels where T_{regs} themselves fail to provide protection. Thus, the proportion of T_{regs} used is 20% of the total T-cell number in our setting. The number of T-cells and T_{regs} used is given in the corresponding Figure legend (Fig. 5a). This approach enables us to reliably detect the effect of aPC. When we use a higher proportion of T_{regs} (50%) we likewise see a survival benefit conveyed by T_{regs} in our experiments, which is in agreement with the work by others. These data are shown below (Fig. R1).

Figure R1: Survival of mice following induction of GvHD and transplantation of different proportions of T_{regs} (20% and 50%). T_{regs} at a proportion of 20% (in relation to the total number of T-cells) fail to protect mice from GvHD, while a higher proportion of T_{regs} (50%) is protective.

Recipient BALB/c mice were lethally irradiated (11Gy) and transplanted with 5×10^6 bone marrow and 0.4×10^6 T-cells and 0.4×10^6 T_{reg} ($B6T(0.4 \times 10^6)+T_{\text{reg}}(0.4 \times 10^6)$; 50% T_{regs}) or with 0.4×10^6 T-cells and 0.1×10^6 T_{reg} ($B6T(0.4 \times 10^6)+T_{\text{reg}}(0.1 \times 10^6)$; 20% T_{regs}) or with 0.4×10^6 T-cells and 0.1×10^6 T_{reg} with aPC-preincubation (20nM, 1h, 37°C) ($B6T(0.4 \times 10^6)+T_{\text{reg}}(0.1 \times 10^6)\text{aPC}$, 20% T_{regs} with aPC pretreatment) from donor C57BL/6 wt mice; * $P < 0.05$ vs. $B6T(0.4 \times 10^6)+T_{\text{reg}}(0.4 \times 10^6)$ (Kaplan Meyer log-rank analyses).

Comment 9:

Abstract: The authors should not conclude "The protective effect of aPC on GVHD does not compromise the GvL effect" merely by the experiment with a single tumor cell line.

Response:

We appreciate this comment. To address this concern we conducted new experiments using a different tumor cell line. To this end tumor cells on a C57BL/6 were generated following protocols previously established within the group¹⁷ (see method section, page 31, lines 1-16). Again, protection from GvHD and a sustained GvL effect were apparent (new results shown in **Supplementary Fig. 14**), corroborating our previous results. These new data strengthens the

conclusion that aPC provides a protective effect on GvHD without compromising the GVL effect. Yet, we concur with the reviewer that we should be more cautious in our conclusion. To this end we now state within the abstract:

“The protective effect of aPC on GVHD does not compromise the GVL effect using two independent tumor cell models” (page 3, line 12).

Furthermore, we now state within the discussion:

“While expansion of T_{regs} following *ex vivo* preincubation of pan T-cells with aPC protects from GvHD, it does not compromise the GVL effect when using two independent tumor cell models” (first paragraph, page 15, lines 7-9).

Comment 10:

Overall, the authors have made a courteous effort on revising, however, because the new insight by this paper is only a combination of the previous paper by Xue (aPC and Treg) and the authors (aPC and PAR3), I must say this paper is not suitable for Nature Communication.

Response:

As stated above in the response to comment 1 of this reviewer and in our previous response we believe that the data within the manuscript provide relevant new insights into mechanisms controlling allogenic T-cell activation. We identify a new function of PAR-receptors (specifically PAR2 and PAR3) on T-cells (which has not been shown before on primary T-cells), which is mechanistic relevant in an important disease model. We also identify the aPC cleavage site within mouse PAR3. In addition, by showing that the effect of aPC requires PAR3 and PAR2 on T_{regs} we provide new insights into the plasticity of protease dependent signalling. Furthermore, we identify a physiological pathway controlling T_{reg} expansion. Finally, the data are of potential translational relevance. Thus, the data are of interest for a broad readership.

Reviewer 2:

The authors have answered all questions, provided significant additional data and the paper is well written, the findings are novel and the data now appear robust.

Response:

We thank this reviewer for the supportive comments and the constructive suggestions made during the first revision.

Reviewer 3:

Comment 1:

The response to my comment 5 does not address my comment, but rather is a repeat (copy/paste) of a preceding section.

Response:

We apologize for not addressing comment 5 adequately. As we had the impression that comment 5 by this reviewer addressed the same concern as raised by reviewer 1 we used largely the same answer (“copy/paste”) for both reviewers.

Comment 5 by this reviewer had two parts. The first part correctly pointed out that the labelling of Fig. 3h was not correct. This had been corrected in the revised version (see also Fig. 6c of the revised version).

The second part addresses the observation that the blocking peptide for PAR2 had no effect on aPC’s ability to suppress proliferation, while the other data implied that PAR2 is necessary for the effect. The difference can be explained by the different mode of action of inhibitory antibodies and blocking peptides. Briefly, the inhibitory antibody prevents partial proteolysis of the N-terminal end of PAR2 and thus the generation of the PAR2 derived tethered ligand by a protease (1st activation step by a protease). The tethered ligand generated by partial proteolysis is required for the 2nd step in PAR2-dependent signalling, as the tethered ligand can bind to the 2nd extracellular loop of PAR2, which is required for signalling¹⁸. The 2nd step in the activation of PARs can be “uncoupled” by using a peptide, which either mimics or blocks the effect of the tethered ligand generated during the 1st step of PAR-activation.

In detail, a cleavage-inhibiting antibody blocks the generation of the tethered ligand (1st activation step by a protease). However, the receptor nevertheless can be activated if an appropriate ligand, which mimics the tethered ligand (e.g. small agonists peptides (AP), e.g. SLIGKV for human PAR2 and SLIGRL for mouse PAR2), is added and binds to the 2nd extracellular loop¹⁸. In addition to adding a small peptide mimicking the tethered ligand it is believed that a tethered ligand of a “neighbouring” PAR may “reach over”, bind to the 2nd extracellular loop of PAR2 and hence activate the receptor. Signalling through PAR2 by a small peptide or a tethered ligand of a different PAR does not require cleavage of the N-terminal end of PAR2¹⁹. In particular the PAR3-derived tethered ligand has been proposed to “reach over” to activate other PARs, including PAR2^{20,21}.

A blocking peptide, on the other hand, is thought to bind to the 2nd extracellular loop of PAR2 and prevents activation of the receptor by its own tethered ligand, but also activation by small agonist peptides or tethered ligands derived from another PAR¹⁸.

During the current revision we have – following a point raised by reviewer 4 – removed the PAR2 inhibitory peptide, as this lacks specificity. Instead, we use now PAR2 ko mice, T-cells derived from PAR2 ko mice, and human PAR2 knock down T-cells to strengthen the conclusion that PAR2 is required. More details regarding this aspect are provided in the answers to reviewer 4. As we no longer use the inhibitory peptide for PAR2 we removed the former Supplementary Fig. 10.

Comment 2:

the response to my comment 6 (request to conduct dose response) is not addressed.

Response:

We appreciate this comment and concur that this point was not adequately addressed. During the last revision we conducted dose-dependent experiments but due to technical problems we did not manage to generate sufficient repeat experiments within the revision time. This open issue has now been addressed (**Supplementary Fig. 4 and Fig. R2 below**). At concentrations as low as 5 nM we still observe an inhibitory effect of aPC when preincubating T-cells with aPC followed by the MLR. Activated PC at 2 nM had no significant effect anymore ($P=0,058$). These new data are now included within the manuscript:

“An aPC concentration of 5 nM was sufficient for aPC’s inhibitory effect (**Supplementary Fig. 4**)” (page 7, lines 27-28).

Figure R2: Dose-response of aPC in the MLR

Following preincubation of human pan T-cells with various concentrations of aPC (concentration as indicated, preincubation for 1 h in AIMV serum free medium) T-cells were co-cultured with irradiated allogenic antigen-presenting cells for 96h. Allogenic T-cell reactivity was measured by thymidine incorporation during the final 16h. Mean value \pm SEM of at least 3 independent experiments, each containing cells from three different donors; * $P < 0.05$, ** $P < 0.01$, NS: not significant (ANOVA).

Reviewer 4:

This manuscript is not really a revised manuscript but rather is a new manuscript based on its huge expansion in both size and extensively new sets of data. The current paper shows that aPC pretreatment of donor T cells reduces graft versus host disease (GVHD) in a murine model. The current paper shows that this is associated with a notable expansion of Tregs. Data suggested the increase of Tregs was needed for reduction of GVHD. Furthermore, in mixed human lymphocyte cultures, aPC treatment of pan T-cells augments Tregs and limits Th1 and Th17 cells. In more than 30 pages single-spaced text of Responses to Reviewers, the authors have attempted to address all major points for the previous review of the previous interesting but poorly executed manuscript. Overall, the authors have addressed satisfactorily many previous weaknesses of their paper. However, notable major points remain that are presented below.

Major Points

Comment 1:

The data in Fig. 7 used to implicate a requirement for murine PAR2 for aPC's effects involved "a PAR2 blocking peptide (FSLRY-NH₂)" (cited as coming from refs #75,76). However, this peptide does not have a convincing track record that it is really a definitive PAR2 blocker in any sense of critical pharmacology. It was best described actually by Al-ani B et al, Hollenberg J Pharmacol Exp Therapeutics, 2002) and it was shown to inhibit trypsin-initiated signaling but not by the PAR2 agonist peptide, SLIGRL-amide. Hence, it does not block PAR2-specific signaling initiated by a PAR2 peptide agonist (SLIGRL-amide). Moreover, when this peptide was used in ref. # 75, a negative control peptide with the reverse sequence, YRLLSF, was used. The authors here fail to provide a negative control peptide and show it has no effect; this is a standard requirement when using peptides as inhibitors in any critically designed experiment. . The peptide, FSLRY is a modified PAR1 sequence, so its effects might actually inhibit some PAR2-PAR1 crosstalk or PAR1/PAR3 crosstalk rather than simply inhibiting PAR2 antagonism by PAR3. Thus, the authors fail to provide strong and clear evidence that murine PAR2 is required for aPC's effects on murine T cells and GVHD. Better reagents, with positive and negative controls, are needed to make the claims of Fig. 7 relating to PAR2's role or requirement.

Response:

We thank for the positive appraisal of our revision work and the critical revision and detailed, but very constructive comments by this reviewer. We concur with the reviewer that the peptide used has limitations.

To address these points, but also those points raised by this reviewer in the following comments, we remove the data obtained with the PAR2 inhibitory peptide and instead used PAR2 ko mice, PAR2 ko mice derived T-cells, and human PAR2 knock down primary T-cells.

When using PAR2 deficient T-cells (obtained from PAR2 ko mice) for the MLR aPC failed to inhibit T-cell activation. These *in vitro* data are shown in Fig. 7b and are addressed in the results section:

"Furthermore, the requirement of PAR for aPC's inhibitory effect on T-cell activation was confirmed using T-cells isolated from PAR2-deficient mice (Fig. 7b)" (page 11, lines 24-26).

Likewise, when using PAR2 deficient T-cells in the GvHD experiments the protective effect of aPC was lost. The protective effect of aPC was lost following pre-treatment of PAR2-deficient T-cells or specifically of PAR2-deficient T_{regs} with aPC. These new data are included in Figure 8 and the results section of the revised manuscript:

"To assess whether PAR2 and PAR3 are required for aPC's ameliorating effect on GvHD we transplanted lethally irradiated BALB/c mice with allogenic (C57BL/6) 5×10^6 BM and 0.5×10^6 C57BL/6 wild-type (B6T), C57BL/6 PAR2^{-/-} (PAR2^{-/-}T), and C57BL/6 PAR3^{-/-} (PAR3^{-/-}T) T-cells. Wild-type (B6T) or receptor deficient T-cells without (B6T+BM; PAR2^{-/-}T+BM; PAR3^{-/-}T+BM) or with (B6T(aPC)+BM; PAR2^{-/-}T(aPC)+BM; PAR3^{-/-}T(aPC)+BM) aPC pretreatment were used. When using PAR2^{-/-} or PAR3^{-/-} pan T-cells the

protective effect of aPC was lost (**Fig. 8a,b**), corroborating the above *in vitro* results (Fig. 6 and 7)” (page 12, lines 5-11),
and

“...this protective effect was lost when by using PAR2-deficient or PAR3-deficient T_{reg} s preincubated with aPC (PAR2^{-/-}T+PAR2^{-/-}T_{reg}(aPC)+BM; PAR3^{-/-}T+PAR3^{-/-}T_{reg}(aPC)+BM, **Fig. 8c,d**)” (page 12, lines 16-18).

Finally, we reduced expression of PAR2 in primary human T-cells (Supplementary Fig. 12) by lentiviral knock down. Following knock down of PAR2 the inhibitory effect of aPC in the MLR using human T-cells was lost, while a scrambled control shRNA had no effect. These new data are shown in Fig. 7a and within the results section we state:

“Considering the lack of specific PAR2-inhibitory peptides¹⁸ we knocked down PAR2 expression in primary human T-cells. Efficient shRNA-mediated knock down of PAR2 was achieved in pan T-cells using transient transfection with lentiviral particles, while a scrambled control shRNA had no effect (**Supplementary Fig. 12**). Knock down of PAR2 abolished the inhibitory effect of aPC in the MLR (**Fig. 7a**)” (page 11, line 20-24).

Taken together, we now provide strong evidence that PAR2 is involved in aPC’s inhibitory effect in regard to allogenic T-cell activation.

Comment 2:

The mechanistic scheme in Suppl. Fig. 10 for the mechanism of aPC’s protection against GVHD as due to PAR3 cleavage followed by binding of the newly generated PAR3 tail to PAR2 is thus not demonstrated at all for the *in vivo* **mechanism of action (MOA)** for reducing GVHD, invalidated any conclusions for *in vivo* MOA. It might more likely be the case that *in vivo*, the protection against GVHD is due to PAR3/PAR1 interactions and signaling, as in podocytes (ms.ref. #39/Responses.ref.#2, Madhusudhan et al, Isermann). Thus, considering the known differences between murine and human PAR3 signaling, the mechanistic scheme of Suppl Fig. 10 is not justified for *in vivo* GVHD prevention by the available data.

Response:

We thank the reviewer for the critical appraisal of the proposed scheme. We would like to emphasize that we referred to the PAR2/PAR3 interaction as a “proposed model”. As we have now used PAR2 ko mice, T-cells derived from PAR2 ko mice, and human PAR2 knock-down T-cells we are confident that PAR2 is involved in the mechanism of action. Regardless, we concur that the exact nature of the interaction, e.g. whether the tethered PAR3-derived ligand indeed “reaches over” to activate PAR2, has not been conclusively shown and needs to be addressed in future follow up studies. Given these limitations, and the fact that we removed the data regarding the PAR2 inhibitory peptide, we now remove Supplementary Fig. 10, which we had added during the previous revision.

We would like to emphasize, however, that PAR-signalling, including involved co-receptors, is highly cell-specific. While PAR3 is a co-receptor for PAR4 on rodent platelets, the mechanism of action appears to be different on human platelets and involves PAR3 and PAR1^{19,21,22}. Furthermore, PAR3 interacts with PAR2 on human podocytes, while on mouse podocytes PAR3 interacts with PAR1²⁰. These data show that PAR cofactoring¹⁹ is cell- and species specific. Accordingly, and in agreement with the current findings, we concluded in our earlier publication that the “identification of this novel signaling mechanism supports a concept of cell-specific signaling complexes, through which coagulation proteases regulate cellular function”²⁰. We believe that the current findings support the concept of cell-specific signalling complexes or “receptorsomes”. To address this issue and to acknowledge the open issues we rephrased our conclusion in the results section and within the discussion:

“We acknowledge that the exact mechanism of action through which PAR2 and PAR3 interact remains currently unresolved” (page 16, lines 3-4).

Comment 3:

Based on the point of view that data for PAR2 role in aPC’s protection *in vivo* vs. GVHD are not at all definitive, the Title is simply a misleading, very possibly erroneous, statement because clear data are lacking for aPC-induced signaling via PAR2/PAR3 in mice for protection against GVHD (the Title’s

message).

The problem is not an easy one to solve. The challenge of defining mechanisms for PAR3-dependent crosstalk involving PAR3-PAR2 or PAR3-PAR1 interactions was highlighted by these authors (ms.ref. #39/Responses.ref.#2, Madhusudhan et al, Isermann) where they reported that in the case of aPC's protective actions on podocytes, human cells used PAR3/PAR2 crosstalk (with data for their coprecipitation) whereas in murine cells, aPC used PAR3/PAR1 crosstalk (with data for their coprecipitation). In those previously published studies, the authors stated that there was "plasticity of aPC mediated cytoprotection." So what is going on for T cells and for aPC's effects on human cells vs. aPC's effects on murine cells and *in vivo* in mice? It is very possible that this GVHD model involves aPC's actions via PAR3/PAR2 for human T cells but PAR3/PAR1 for murine T cells. Further work is needed to clarify these possibilities.

Response 3:

Again, we appreciate this comment. To address this question we have used now PAR2 ko mice, T-cells derived from PAR2 ko mice, and human PAR2 knock down T-cells. Using PAR2 ko mice and T-cells obtained from PAR2 ko mice we now convincingly show that PAR2 is required on murine T-cells, including murine T_{regs}, for aPC's inhibitory effect in the GvHD model and the MLR. Furthermore, using human PAR2 knock down T-cells we demonstrate that PAR2 is required for aPC's inhibitory effect in the MLR using human cells. These results support our previous conclusion drawn based on the employment of the inhibitory peptide (which, as discussed, has limitations). Given the new insights provided during this revision, which demonstrate that PAR2 and PAR3 are required for aPC's effect on T-cells in *in vitro* and *in vivo* models of allogeneic T-cell stimulation, we would like to keep the title of the manuscript. However, if the editor or reviewer feels that the title should be changed we will do so.

We believe the fact that "PAR-cofactoring" on T-cells differs from that on podocytes is entirely congruent with the proposed plasticity of aPC mediated cytoprotection or signalling. This issue is already addressed in comment 2 / response 2 of this reviewer. Deciphering the plasticity of PAR-cofactoring will not only provide new mechanistic insights, but may also provide rationale for cell or tissue specific therapeutic approaches. We believe that this is one of the exciting challenges in the field.

To address this important issue we added the following section to the discussion:

"However, cofactoring of PARs is established and activation of other PARs, including PAR2, by the PAR3 derived tethered ligand has already been proposed^{33,40,41}. Intriguingly, while aPC signals *via* species-specific PAR-heterodimers in human and mouse podocytes (human: PAR3/PAR2; mouse: PAR3/PAR1), the same PAR-heterodimer (PAR3/PAR2) is required for aPC's effect on mouse and human T_{regs}. This observation emphasizes the plasticity of PAR-signalling, both in regard to different cell-types and species" (page 16, lines 4-9).

Comment 4:

The authors should address the challenging question of how does aPC cleave and activate murine PAR3, in discussion and in experiments? A key paper cited (ms.ref. #49/Responses.ref.#3, Burnier & Mosnier) for aPC's cleavage of PAR3 proves that aPC cleaves human PAR3 at Arg41, not at the Lys39 thrombin canonical cleavage site. However, murine PAR3 lacks this Arg residue although it has a similar Lys cleavage site for thrombin. So there is no evidence, as far as this reviewer knows in the literature, which show whether and where aPC cleaves murine PAR3. Clearly the authors have data for murine cells from PAR3 knockout mice that PAR3 is required for aPC's cytoprotection but no data for how aPC actually cleaves murine PAR3 to effect cytoprotection. So the scheme for mechanism of action in Suppl. Fig. 10 as that which explains GVHD effects seems premature without ore data for cleavage of murine PAR3 by aPC. It is demonstrated here that T cells from PAR3 knockout mice do not show aPC's beneficial effects, indicating PAR3 is required. But PAR3 mechanism schemes for *in vivo* mechanisms are not yet very well justified.

Response 4:

We thank the reviewer for the critical appraisal of this issue. We have previously demonstrated in the Blood paper published in 2013 that aPC cleaves human and mouse PAR3 (Fig. 4 and Supplementary Figure S4 within this publication, see **Fig. R3**)²⁰. Within the 2013 study we transfected renal cells (mesangial cells) with N-terminal V5-tagged human or mouse PAR3 constructs. We were able to demonstrate that following exposure of cells to aPC, but not to the solvent PBS, the V5 epitope could be detected in the supernatant, reflecting proteolytic cleavage of both the human and mouse PAR3 by aPC (**Fig. R3**). Furthermore, when we induced a mutation into PAR3 aPC failed to cleave PAR3, corroborating that aPC is able to cleave PAR3 (**Fig. R3**). These data strongly suggest that aPC is able to cleave the N-terminal end of human and mouse PAR3. This aspect is now addressed within the discussion:

“Proteolytic cleavage of human and mouse PAR3 by aPC has been previously reported by us and others in endothelial and renal cells^{35,36}” (page 15, lines 25-27).

We concur that mouse PAR3 lacks – at face value – a specific cleavage site for aPC, as a residue corresponding to Arg41 in human PAR3 is lacking. To address this issue we conducted experiments similar to those conducted by Burnier and Mosnier²³. To validate our approach we used in parallel the human PAR3 peptide and followed the same protocol as described by Burnier and Mosnier. We observed the same cleavage sites for thrombin (Lys38) and aPC (Arg41) in the human PAR3 derived peptide (**Fig. R4** and **Supplementary Fig. 11**). This validates our experimental approach.

Next, we incubated the mouse PAR3-derived peptide with human thrombin and human aPC following the same protocol. We used human aPC as this was also used in our studies and as mouse aPC was not available to us in sufficient quantity. Both, human thrombin and human aPC cleaved the mouse PAR3-derived peptide at Arg37 (**Fig. 6e**). This result is entirely congruent with the results of our mutagenesis studies mentioned above and published in 2012²⁰.

Taken together, our previously published mutagenesis studies and the current proteolysis studies establish that aPC cleaves mouse PAR3. These new results are now included within the manuscript:

“Burnier and Mosnier³⁵ previously established generation of a human PAR3-dependent tethered ligand by aPC. We previously demonstrated that aPC cleaves the N-terminal end of mouse PAR3, but the exact cleavage site remained unknown³⁶. To analyse aPC cleavage of mouse PAR3 we followed established protocols³⁵. Using the human PAR3 derived N-terminal peptide we first validated the approach by replicating previous results³⁵ (**Supplementary Fig. 11**). Cleavage analyses of the mouse PAR3-derived N-terminal peptide revealed that both human thrombin and human aPC efficiently cleave at Arg37 within 2h (**Fig. 6e**). This result is congruent with our previous mutagenesis studies³⁶ and establishes for the first time an aPC cleavage site within the murine PAR3 derived N-terminus” (page 11, lines 3-11).

In addition, we discuss the limitations and implications of these findings:

“The current results together with our previous studies³⁶ establish that aPC cleaves the N-terminal end of PAR3, thus generating a tethered ligand, which may interact with other PARs. As we used human aPC throughout our study the finding that human aPC cleaves mouse PAR3 is relevant in the context of the current study. Future studies need to address whether likewise mouse aPC cleaves mouse PAR3. Intriguingly, thrombin and aPC cleaved mouse PAR3 at the same residue site (Arg37), contrasting the observations made with the human PAR3 derived peptide³⁵ (and current study). Signalling specificity of thrombin and aPC *via* mouse PAR3 can hence not be explained by different tethered ligands generated by either protease. Biased signalling independent of the cleavage site, but rather secondary to involvement of specific receptor and signalling complexes, has been recently proposed for aPC and thrombin dependent β -arrestin-2 signalling *via* PAR1⁵¹. Accordingly, we propose that specificity of PAR3 signalling is not only dependent on specific PAR3 cleavage sites, but in addition on a specific PAR3-containing protein complex” (page 16, lines 10-22).

Fig. R3: Data supporting that aPC cleaves the N-terminal end of both human and mouse PAR3. **A:** Representative immunoblot showing V5 levels in the culture supernatant after treatment with PBS or aPC (20nM) for 1 hour of cells transfected with V5-tagged human PAR3 wild-type and V5-tagged mutant (T39P) human PAR3 expression constructs. The detection of the V5 epitope in the supernatant reflects proteolytic activation of PAR3. PAR3-TEM1-V5: Expression of a chimeric protein consisting of the N-terminal end of PAR3 and the transmembrane and cytoplasmic domain of endosialin (TEM-1); for further details see original publication (Madhusudhan, Blood, 2013)¹⁸. **B:** Representative immunoblot showing V5 levels in the culture supernatant after treatment with PBS or aPC (20 nM) for 1 h of mouse mesangial cells transfected with V5 tagged wild type mouse PAR3 and V5 tagged mutant mouse PAR3 (S38P) expression constructs.

Fig R4: Summary of PAR3-N-terminal peptide cleavage analyses. Peptides derived from the N-terminal ends of human (A) or mouse (B) PAR3 were incubated with human thrombin (10 nM) or human aPC (500 nM). Proteolysis was determined at indicated time-points (left, t(h)). Cleavage sites are shown in red lines and estimated cleavage efficiency is shown in percentage (right).

References:

1. Isermann, B., *et al.* Activated protein C protects against diabetic nephropathy by inhibiting endothelial and podocyte apoptosis. *Nature medicine* **13**, 1349-1358 (2007).
2. Deng, R., *et al.* B7H1/CD80 interaction augments PD-1-dependent T cell apoptosis and ameliorates graft-versus-host disease. *Journal of immunology* **194**, 560-574 (2015).
3. Locke, F.L., Zha, Y.Y., Zheng, Y., Driessens, G. & Gajewski, T.F. Conditional deletion of PTEN in peripheral T cells augments TCR-mediated activation but does not abrogate CD28 dependency or prevent anergy induction. *Journal of immunology* **191**, 1677-1685 (2013).
4. Peng, W., *et al.* Dendritic cells transfected with PD-L1 recombinant adenovirus induces T cell suppression and long-term acceptance of allograft transplantation. *Cellular immunology* **271**, 73-77 (2011).
5. Kim, B.S., *et al.* Treatment with agonistic DR3 antibody results in expansion of donor Tregs and reduced graft-versus-host disease. *Blood* **126**, 546-557 (2015).
6. Yi, T., *et al.* Reciprocal differentiation and tissue-specific pathogenesis of Th1, Th2, and Th17 cells in graft-versus-host disease. *Blood* **114**, 3101-3112 (2009).
7. Mochizuki, K., *et al.* Programming of donor T cells using allogeneic delta-like ligand 4-positive dendritic cells to reduce GVHD in mice. *Blood* **127**, 3270-3280 (2016).
8. Belle, L., *et al.* Blockade of interleukin 27 signaling reduces GVHD in mice by augmenting Treg reconstitution and stabilizing FOXP3 expression. *Blood* (2016).
9. Leclerc, M., *et al.* Control of GVHD by regulatory T cells depends on TNF produced by T cells and TNFR2 expressed by regulatory T cells. *Blood* **128**, 1651-1659 (2016).
10. Zheng, H., Matte-Martone, C., Jain, D., McNiff, J. & Shlomchik, W.D. Central memory CD8+ T cells induce graft-versus-host disease and mediate graft-versus-leukemia. *Journal of immunology* **182**, 5938-5948 (2009).
11. Pierini, A., *et al.* Donor Requirements for Regulatory T Cell Suppression of Murine Graft-versus-Host Disease. *Journal of immunology* **195**, 347-355 (2015).
12. Heinrichs, J., *et al.* CD8(+) Tregs promote GVHD prevention and overcome the impaired GVL effect mediated by CD4(+) Tregs in mice. *Oncoimmunology* **5**, e1146842 (2016).
13. Cohen, J.L., Trenado, A., Vasey, D., Klatzmann, D. & Salomon, B.L. CD4(+)CD25(+) immunoregulatory T Cells: new therapeutics for graft-versus-host disease. *The Journal of experimental medicine* **196**, 401-406 (2002).
14. Hoffmann, P., Ermann, J., Edinger, M., Fathman, C.G. & Strober, S. Donor-type CD4(+)CD25(+) regulatory T cells suppress lethal acute graft-versus-host disease after allogeneic bone marrow transplantation. *The Journal of experimental medicine* **196**, 389-399 (2002).
15. Ermann, J., *et al.* Only the CD62L+ subpopulation of CD4+CD25+ regulatory T cells protects from lethal acute GVHD. *Blood* **105**, 2220-2226 (2005).
16. Bolton, H.A., *et al.* Selective Treg reconstitution during lymphopenia normalizes DC costimulation and prevents graft-versus-host disease. *The Journal of clinical investigation* **125**, 3627-3641 (2015).
17. Heidel, F.H., *et al.* Genetic and pharmacologic inhibition of beta-catenin targets imatinib-resistant leukemia stem cells in CML. *Cell Stem Cell* **10**, 412-424 (2012).
18. Adams, M.N., *et al.* Structure, function and pathophysiology of protease activated receptors. *Pharmacology & therapeutics* **130**, 248-282 (2011).
19. Lin, H., Liu, A.P., Smith, T.H. & Trejo, J. Cofactoring and dimerization of proteinase-activated receptors. *Pharmacological reviews* **65**, 1198-1213 (2013).
20. Madhusudhan, T., *et al.* Cytoprotective signaling by activated protein C requires protease-activated receptor-3 in podocytes. *Blood* **119**, 874-883 (2012).
21. Coughlin, S.R. Thrombin signalling and protease-activated receptors. *Nature* **407**, 258-264 (2000).
22. Gieseler, F., Ungefroren, H., Settmacher, U., Hollenberg, M.D. & Kaufmann, R. Proteinase-activated receptors (PARs) - focus on receptor-receptor-interactions and their physiological and pathophysiological impact. *Cell communication and signaling : CCS* **11**, 86 (2013).

23. Burnier, L. & Mosnier, L.O. Novel mechanisms for activated protein C cytoprotective activities involving noncanonical activation of protease-activated receptor 3. *Blood* **122**, 807-816 (2013).

Reviewer #1 comments

The authors have substantially worked on the issues brought up, however the manuscript has still major points that need clarification.

Reply:

We thank the reviewer for carefully reviewing our manuscript again. The points raised are addressed as follows:

Major:

1. (Follow up from the authors response to the first review comment #26). Regarding the mouse experiments showing sustained GvL effect after aPC treatment (Fig. 10c, Supple Fig. 13b,c), the tumor burden on day 28 was rather diminished by aPC treatment in two tumor cell lines.

The authors still do not provide an adequate response i.e. a mechanistic hypothesis on how aPC treatment may have enhanced the GvL effect despite increased numbers of Tregs. Their previous response explained virtually nothing about the underlying mechanism.

Moreover, since the authors claim that aPC treatment also decreased Th1 and Th17 cells, then which compartment played the critical role in their model?

In the same context, why didn't the authors in their experiments analyze CD8 T cells at all? In Supplementary Fig. 1 they should at least show the data for CD8 and Treg as well.

Answer:

We appreciate these comments. Regarding the sustained GvL effect following aPC preincubation of T-cells, we did not provide more details concerning potential mechanisms due to space restriction. A sustained or enhanced GvL effect following treatment of mice with T_{regs} has been repeatedly demonstrated [1-4]. Thus, we hypothesize that the increased frequency of T_{regs} following *ex vivo* preincubation with aPC does not only ameliorate GvHD, but also allows an efficient GvL effect. We concur with the reviewer that the dual effect of T_{regs} in regard to GvHD and GvL, as shown by us and others [1-4], is of high interest. Several potential mechanisms have been discussed, and we propose to cite relevant publications (please see new text passage below). Furthermore, it will be interesting to determine whether the preincubation of T-cells with aPC is superior to “simply” using a higher number of T_{regs} (e.g. whether there is simply a quantitative effect of aPC on T_{regs}, or whether there is in addition an qualitative effect of aPC on T_{reg} function). We believe, however, that such studies are beyond the scope of the current manuscript.

Regarding the potential role of the decreased Th1 and Th17 cell frequency: we demonstrate that the expansion of T_{regs} following preincubation with aPC is critical for aPC's effect in respect to GvHD. This was demonstrated by depleting T_{regs} using the DEREK mice. Secondary effects depending on a decreased frequency of Th1 or Th17 cells cannot be excluded, but a pivotal role of the increased T_{reg} number is demonstrated by the depletion of T_{regs}.

Similar approaches could be used to evaluate the role of T_{regs}, Th1, or Th17 cells for aPC's effect on GvL. The feasibility of such approaches has been shown [5]. Furthermore, we concur with the reviewer that CD8⁺ T cells are an interesting target to investigate, as previously shown by others [5, 6]. We detected expression of PAR3 on CD8⁺ T-cells, albeit at lower levels than on T_{regs} (Fig. 6b of the main manuscript), making CD8⁺ cells a less attractive target in the context of the current study. Considering the amount of data shown and the length of the manuscript we don't think that it is possible to appropriately address aPC's effect on other T-cell population within the current manuscript.

We thank the reviewer for bringing these points to our attention and we will follow up on these points in future studies. If agreed by the reviewer and the editor we will add the following passage to the discussion to address these points (new text passage shown in red):

“While expansion of T_{regs} following *ex vivo* preincubation of pan T-cells with aPC protects from GvHD, it does not compromise the GvL effect when using two independent tumor cell models. **The sustained GvL effect following preincubation of T-cells with aPC and subsequent expansion of T_{regs} is congruent with previous studies demonstrating that T_{regs} do not only prevent GvHD, but simultaneously convey an efficient GvL effect [1-4]. Several mechanisms have been proposed for the dual effect of T_{regs} in regard to GvHD and the GvL effect, e.g. inhibition of JAK1/JAK2 signalling, expression of NKG2D by CD8⁺ cells, IL-21 signalling, or the differential expression of granzyme B by CD4⁺CD25⁺ and CD8⁺ cells [7-10]. Whether these mechanisms are regulated by aPC remains unknown. Likewise, the relevance of other T-cell populations, such as Th1, Th17, or CD8⁺ T-cells for aPC's effects as observed in the current study remains to be evaluated” (new text passage to be placed on page 15, between line 9 and 10).**

2. Follow up from the previous authors response to my comment #3. They should address more specifically which cells other than T cells produced Treg-inducible cytokines, and provide some data or references. Their comment to the reviewer is contradictory in itself.

We apologize that we did not address this question appropriately. In addition to lymphocytes macrophages, platelets, liver endothelial cells, and parenchymal cells can produce TGF β and potentially promote T_{reg} induction [11-13]. Similar, IL-10 is known to be produced by various cell types, including macrophages, neutrophils, dendritic cells, B cells, and different subsets of CD4⁺ and CD8⁺ T cells [14, 15]. As coagulation proteases regulate in particular macrophages, neutrophils, and platelets we envision that coagulation factors, including aPC, may regulate TGF β and / or IL-10 secretion from these cells [16-21]. Which cell types are targeted by aPC to regulate cytokines promoting T_{regs} differentiation remains to be evaluated in independent studies.

If agreed by the editor we will add the following sentence to the discussion to address this point (new text passage shown in red):

“While the plasma cytokine profile in mice is in agreement with the cytokine expression pattern observed in murine donor CD4⁺ T-cells, it is likely that cells other than T-cells contributed to the plasma cytokine profile. **Of note, IL-10 and TGF β are produced by various cell types, including macrophages, neutrophils, or platelets, which are subject to regulation by coagulation proteases [16-21]**” (new text passage to be placed on page 7, between line 16 and 17).

3. (Follow up from the authors response to previous comment #5). Their response was not what I meant. What is the strategy for defining positivity on FACS plots of Supplementary Fig. 1, Supplementary Fig2, Supplementary 3ab, Supplementary 6a, Supplementary 7a and Supplementary Fig. 10.

First, unstained controls did not work so that the MFI values of negative subpopulations differed between unstained controls and stained samples.

Second, the differences in the frequency of T-bet⁺ and ROR-gt⁺ cells in control plots between B6T+BM and B6T(aPC)+BM (Supplementary Fig. 2) are not acceptable.

Third, controls stained only for CD4 are not appropriate for the true control when they stain multiple cytokines or intranuclear molecules at the same time.

Along with this context, the authors should show the FACS data for isotype controls at least for cytokines and transcription factors staining in Supplementary Fig2, Supplementary 3ab, Supplementary 6a, Supplementary 7a and Supplementary Fig. 10.

They should also pay attention to the fact that almost all of the MFI data DO NOT provide statistical significance.

In addition, why were CD3⁺ and CD3⁻ subpopulations not clearly discriminated in Supplementary Fig. 1?

Why do the MFI values of CD25 obviously differ inappropriately and tremendously between the four cohorts in Supplementary Fig. 10? If this is based on differences in the isotype stainings they should provide the isotype control plots in the figure.

Answer:

We appreciate the detailed comments in regard to the FACS analyses. Contributing authors met to discuss these comments. Briefly, we followed standard procedures for staining of samples and gating [22-24].

1. Staining of samples

All antibodies were carefully titrated to reduce antibody specific background. In addition, Fc-binding reagent was used to block non-specific staining of samples in all cases. After intracellular staining of transcription factors or cytokines samples were washed 3 times with staining buffer, followed by a final washing step with PBS and re-suspension of the cells in PBS-BSA (0.2%). Cells were then used for data acquisition.

2. Gating control

As we carefully titrated antibodies to reduce their background staining and as we typically used 2 or 3 colours for each sample we used FMO controls (fluorescent minus one) to set up the gating strategy [22-24]. FMO controls have the advantage that they account for spillover effects on the channel of interest [22].

In a pilot experiment we initially attempted to use isotype controls for intracellular staining of transcription factors and cytokines, but we observed inappropriate variations and differences in the background staining when comparing test antibodies with corresponding isotype control antibodies. Such limitations of isotype control antibodies are known [22].

Furthermore, the conclusions drawn within the manuscript are based on comparison with biological controls. Biological comparison controls are frequently considered to be the most relevant control to determine an effect [22].

We address the specific comments as follows:

1. Representative gating strategies are now shown for supplementary Figures 1, 2, 3, 6, 7, and 10 (see “attachment-01.pdf”). These new information can be included within the manuscript as supplementary information. Furthermore, an improved version of supplementary Figure 1 is included as “attachment-02.pdf”.
2. As we are not showing MFIs for unstained controls we assume that the reviewer is referring to the different frequencies shown in the negative control samples of some FACS plots (e.g. Supplementary Fig. 2, T-bet and ROR- γ t). As outline above, we carefully titrated antibodies to avoid non-specific staining as much as possible and we used FMO as controls for our FACS analyses. Conclusions are drawn based on comparison of different biological samples. However, we concur with the reviewer that in particular the controls shown in Supplementary Fig. 2 for T-bet and ROR- γ t were not appropriate (see next answer).
3. Regarding the frequency of T-bet⁺ and ROR- γ t⁺ cells in control plots between B6T+BM and B6T(aPC)+BM (Supplementary Fig. 2) we re-analysed the data. When control plots are used which show an equal frequency of T-bet⁺ and ROR- γ t⁺ cells the absolute numbers for “stained” samples are overall slightly lower, but the differences remain significant. The corrected FACS-plots and the corresponding corrected Fig. 3e are included in “attachment-03.pdf”. We will incorporate these corrected figures into the manuscript if the manuscript is accepted.
4. We would like to point out that we did not stain for multiple intracellular cytokines or transcription factors. We conducted double staining for CD4 and the transcription factor or cytokine of interest. In these cases CD4 served as FMO control as outlined above.
5. As outline above, we used FMO as control [22-24]. To support our conclusion we determined cytokine levels in the plasma (Fig. 3g) and in the supernatant of stimulated cells (Fig. 4f) or expression of FOXP3 by immunoblotting (Fig. 5e). The results from these analyses corroborate the conclusions drawn from the FACS analyses.
6. We appreciate the comment regarding the MFI data. In the first draft of the manuscript we provided iMFI (integrated mean fluorescent intensity) data, and reviewers requested us to show the FACS plots (frequency) and the MFI separately. We concur with this approach, as the frequency (in %, as shown in the FACS plots and corresponding bar-graphs) provides information regarding the relative cell-population size, while the MFI provides (semi-)quantitative information regarding the fluorescent signal strength (which depends on various factors, including the number of antibodies bound to a cell and the voltage setting). Accordingly, the information provided by these measures is not the same and hence differences – as in our case – can be observed. We checked our primary data and would like to emphasize that some of the MFI differences are of borderline significance (see “attachment-04.pdf”). If accepted, we can include this information in the manuscript.
7. CD3⁺ cells in Supplementary Figure 1: We thank the reviewer for bringing this issue to our attention. We checked available FACS plot images and provide a revised version of supplementary Figure 1 showing a FACS plot which clearly discriminates CD3⁻ and CD3⁺ populations (see “attachment-02.pdf”).
8. Supplementary Figure 10 does not show MFI, but the frequency. Here we stimulated human T_{eff} cells with antigen presenting cells (AgPC) and compared them to non-stimulated controls (T_{eff}), both in the absence (top) or with aPC preincubation. In humans, CD4⁺CD25^{high} cells convey the function of T_{regs}, while CD4⁺CD25^{intermediate} cells have no immunosuppressive function [25-29]. Following activation we observe, as expected, a shift of CD25⁺ cells, which reflects both T_{eff} activation (CD25^{intermediate}) and T_{reg} induction (CD25^{high}). We gated on the CD25^{high} cells, which are a distinct population in the plots. To corroborate induction of T_{regs} cells we determine in parallel FOXP3 expression by immunoblotting (Fig. 5f).

minor:

Supple Fig.8. C57BL76?

Supple Fig. 10. FACS?

Answer:

We thank this reviewer for bringing these typing errors to our attention. These errors will be corrected in a finalized version of the manuscript, if the manuscript is accepted.

References:

- 1 Jones SC, Murphy GF, Korngold R. Post-hematopoietic cell transplantation control of graft-versus-host disease by donor CD425 T cells to allow an effective graft-versus-leukemia response. *Biology of blood and marrow transplantation : journal of the American Society for Blood and Marrow Transplantation*. 2003; **9**: 243-56. 10.1053/bbmt.2003.50027.
- 2 Edinger M, Hoffmann P, Ermann J, Drago K, Fathman CG, Strober S, Negrin RS. CD4+CD25+ regulatory T cells preserve graft-versus-tumor activity while inhibiting graft-versus-host disease after bone marrow transplantation. *Nature medicine*. 2003; **9**: 1144-50. 10.1038/nm915.
- 3 Trenado A, Charlotte F, Fisson S, Yagello M, Klatzmann D, Salomon BL, Cohen JL. Recipient-type specific CD4+CD25+ regulatory T cells favor immune reconstitution and control graft-versus-host disease while maintaining graft-versus-leukemia. *The Journal of clinical investigation*. 2003; **112**: 1688-96. 10.1172/JCI17702.
- 4 Yao Y, Wang L, Zhou J, Zhang X. HIF-1 α inhibitor echinomycin reduces acute graft-versus-host disease and preserves graft-versus-leukemia effect. *Journal of translational medicine*. 2017; **15**: 28. 10.1186/s12967-017-1132-9.
- 5 Gartlan KH, Markey KA, Varelias A, Bunting MD, Koyama M, Kuns RD, Raffelt NC, Olver SD, Lineburg KE, Cheong M, Teal BE, Lor M, Comerford I, Teng MW, Smyth MJ, McCluskey J, Rossjohn J, Stockinger B, Boyle GM, Lane SW, Clouston AD, McColl SR, MacDonald KP, Hill GR. Tc17 cells are a proinflammatory, plastic lineage of pathogenic CD8+ T cells that induce GVHD without antileukemic effects. *Blood*. 2015; **126**: 1609-20. 10.1182/blood-2015-01-622662.
- 6 Zheng J, Liu Y, Liu Y, Liu M, Xiang Z, Lam KT, Lewis DB, Lau YL, Tu W. Human CD8+ regulatory T cells inhibit GVHD and preserve general immunity in humanized mice. *Science translational medicine*. 2013; **5**: 168ra9. 10.1126/scitranslmed.3004943.
- 7 Carniti C, Gimondi S, Vendramin A, Recordati C, Confalonieri D, Bermema A, Corradini P, Mariotti J. Pharmacologic Inhibition of JAK1/JAK2 Signaling Reduces Experimental Murine Acute GVHD While Preserving GVT Effects. *Clinical cancer research : an official journal of the American Association for Cancer Research*. 2015; **21**: 3740-9. 10.1158/1078-0432.CCR-14-2758.
- 8 Karimi MA, Bryson JL, Richman LP, Fesnak AD, Leichner TM, Satake A, Vonderheide RH, Raulet DH, Reshef R, Kambayashi T. NKG2D expression by CD8+ T cells contributes to GVHD and GVT effects in a murine model of allogeneic HSCT. *Blood*. 2015; **125**: 3655-63. 10.1182/blood-2015-02-629006.
- 9 Hanash AM, Kappel LW, Yim NL, Nejat RA, Goldberg GL, Smith OM, Rao UK, Dykstra L, Na IK, Holland AM, Dudakov JA, Liu C, Murphy GF, Leonard WJ, Heller G, van den Brink MR. Abrogation of donor T-cell IL-21 signaling leads to tissue-specific modulation of immunity and separation of GVHD from GVL. *Blood*. 2011; **118**: 446-55. 10.1182/blood-2010-07-294785.
- 10 Du W, Leigh ND, Bian G, Alqassim E, O'Neill RE, Mei L, Qiu J, Liu H, McCarthy PL, Cao X. Granzyme B Contributes to the Optimal Graft-Versus-Tumor Effect Mediated by Conventional CD4+ T Cells. *J Immunol Res Ther*. 2016; **1**: 22-8.
- 11 Branton MH, Kopp JB. TGF-beta and fibrosis. *Microbes Infect*. 1999; **1**: 1349-65.
- 12 Carambia A, Freund B, Schwinge D, Heine M, Laschtowitz A, Huber S, Wraith DC, Korn T, Schramm C, Lohse AW, Heeren J, Herkel J. TGF-beta-dependent induction of CD4(+)CD25(+)Foxp3(+) Tregs by liver sinusoidal endothelial cells. *Journal of hepatology*. 2014; **61**: 594-9. 10.1016/j.jhep.2014.04.027.
- 13 Yang H, Sun J, Li Y, Duan WM, Bi J, Qu T. Human umbilical cord-derived mesenchymal stem cells suppress proliferation of PHA-activated lymphocytes in vitro by inducing CD4(+)CD25(high)CD45RA(+) regulatory T cell production and modulating cytokine secretion. *Cell Immunol*. 2016; **302**: 26-31. 10.1016/j.cellimm.2016.01.002.
- 14 Couper KN, Blount DG, Riley EM. IL-10: the master regulator of immunity to infection. *Journal of immunology*. 2008; **180**: 5771-7.
- 15 Saraiva M, O'Garra A. The regulation of IL-10 production by immune cells. *Nature reviews Immunology*. 2010; **10**: 170-81. 10.1038/nri2711.
- 16 Liang HP, Kerschen EJ, Hernandez I, Basu S, Zogg M, Botros F, Jia S, Hessner MJ, Griffin JH, Ruf W, Weiler H. EPCR-dependent PAR2 activation by the blood coagulation initiation complex regulates LPS-triggered interferon responses in mice. *Blood*. 2015; **125**: 2845-54. 10.1182/blood-2014-11-610717.
- 17 Braach N, Frommhold D, Buschmann K, Pflaum J, Koch L, Hudalla H, Staudacher K, Wang H, Isermann B, Nawroth P, Poeschl J. RAGE controls activation and anti-inflammatory signalling of protein C. *PloS one*. 2014; **9**: e89422. 10.1371/journal.pone.0089422.
- 18 Yang XV, Banerjee Y, Fernandez JA, Deguchi H, Xu X, Mosnier LO, Urbanus RT, de Groot PG, White-Adams TC, McCarty OJ, Griffin JH. Activated protein C ligation of ApoER2 (LRP8) causes Dab1-

- dependent signaling in U937 cells. *Proceedings of the National Academy of Sciences of the United States of America*. 2009; **106**: 274-9. 10.1073/pnas.0807594106.
- 19 Shua F, Kobayashia H, Fukudomeb K, Tsuneyoshib N, Kimotob M, Teraoa T. Activated protein C suppresses tissue factor expression on U937 cells in the endothelial protein C receptor-dependent manner. *FEBS Lett*. 2000; **477**: 208-12.
 - 20 Cao C, Gao Y, Li Y, Antalis TM, Castellino FJ, Zhang L. The efficacy of activated protein C in murine endotoxemia is dependent on integrin CD11b. *The Journal of clinical investigation*. 2010; **120**: 1971-80. 10.1172/JCI40380.
 - 21 Elphick GF, Sarangi PP, Hyun YM, Hollenbaugh JA, Ayala A, Biffl WL, Chung HL, Rezaie AR, McGrath JL, Topham DJ, Reichner JS, Kim M. Recombinant human activated protein C inhibits integrin-mediated neutrophil migration. *Blood*. 2009; **113**: 4078-85. 10.1182/blood-2008-09-180968.
 - 22 Maecker HT, Trotter J. Flow cytometry controls, instrument setup, and the determination of positivity. *Cytometry A*. 2006; **69**: 1037-42. 10.1002/cyto.a.20333.
 - 23 Roederer M. Spectral compensation for flow cytometry: visualization artifacts, limitations, and caveats. *Cytometry*. 2001; **45**: 194-205.
 - 24 Alvarez DF, Helm K, Degregori J, Roederer M, Majka S. Publishing flow cytometry data. *American journal of physiology Lung cellular and molecular physiology*. 2010; **298**: L127-30. 10.1152/ajplung.00313.2009.
 - 25 Chen W, Jin W, Hardegen N, Lei KJ, Li L, Marinos N, McGrady G, Wahl SM. Conversion of peripheral CD4⁺CD25⁻ naive T cells to CD4⁺CD25⁺ regulatory T cells by TGF- β induction of transcription factor Foxp3. *The Journal of experimental medicine*. 2003; **198**: 1875-86. 10.1084/jem.20030152.
 - 26 Jacob C, Yang PC, Darmoul D, Amadesi S, Saito T, Cottrell GS, Coelho AM, Singh P, Grady EF, Perdue M, Bunnett NW. Mast cell tryptase controls paracellular permeability of the intestine. Role of protease-activated receptor 2 and beta-arrestins. *The Journal of biological chemistry*. 2005; **280**: 31936-48. 10.1074/jbc.M506338200.
 - 27 Lundgren A, Stromberg E, Sjoling A, Lindholm C, Enarsson K, Edebo A, Johnsson E, Suri-Payer E, Larsson P, Rudin A, Svennerholm AM, Lundin BS. Mucosal FOXP3-expressing CD4⁺ CD25^{high} regulatory T cells in Helicobacter pylori-infected patients. *Infection and immunity*. 2005; **73**: 523-31. 10.1128/IAI.73.1.523-531.2005.
 - 28 Duhon T, Duhon R, Lanzavecchia A, Sallusto F, Campbell DJ. Functionally distinct subsets of human FOXP3⁺ Treg cells that phenotypically mirror effector Th cells. *Blood*. 2012; **119**: 4430-40. 10.1182/blood-2011-11-392324.
 - 29 Nyirenda TS, Molyneux ME, Kenefeck R, Walker LS, MacLennan CA, Heyderman RS, Mandala WL. T-Regulatory Cells and Inflammatory and Inhibitory Cytokines in Malawian Children Residing in an Area of High and an Area of Low Malaria Transmission During Acute Uncomplicated Malaria and in Convalescence. *J Pediatric Infect Dis Soc*. 2015; **4**: 232-41. 10.1093/jpids/piu140.

REVIEWERS' COMMENTS:

Reviewer #1 (Remarks to the Author):

The authors have substantially worked on the issues brought up, however the manuscript has still major points that need clarification.

major:

1. (Follow up from the authors response to the first review comment #26). Regarding the mouse experiments showing sustained GVL effect after aPC treatment (Fig. 10c, Supple Fig. 13b,c), the tumor burden on day 28 was rather diminished by aPC treatment in two tumor cell lines.

The authors still do not provide an adequate response i.e. a mechanistic hypothesis on how aPC treatment may have enhanced the GVL effect despite increased numbers of Tregs. Their previous response explained virtually nothing about the underlying mechanism.

Moreover, since the authors claim that aPC treatment also decreased Th1 and Th17 cells, then which compartment played the critical role in their model?

In the same context, why didn't the authors in their experiments analyze CD8 T cells at all? In Supplementary Fig. 1 they should at least show the data for CD8 and Treg as well.

2. Follow up from the previous authors response to my comment #3. They should address more specifically which cells other than T cells produced Treg-inducible cytokines, and provide some data or references. Their comment to the reviewer is contradictory in itself.

3. (Follow up from the authors response to previous comment #5). Their response was not what I meant. What is the strategy for defining positivity on FACS plots of Supplementary Fig. 1, Supplementary Fig2, Supplementary 3ab, Supplementary 6a, Supplementary 7a and Supplementary Fig. 10.

First, unstained controls did not work so that the MFI values of negative subpopulations differed between unstained controls and stained samples.

Second, the differences in the frequency of T-bet+ and ROR-gt+ cells in control plots between B6T+BM and B6T(aPC)+BM (Supplementary Fig. 2) are not acceptable.

Third, controls stained only for CD4 are not appropriate for the true control when they stain multiple cytokines or intranuclear molecules at the same time.

Along with this context, the authors should show the FACS data for isotype controls at least for cytokines and transcription factors staining in Supplementary Fig2, Supplementary 3ab, Supplementary 6a, Supplementary 7a and Supplementary Fig. 10.

They should also pay attention to the fact that almost all of the MFI data DO NOT provide statistical significance.

In addition, why were CD3+ and CD3- subpopulations not clearly discriminated in Supplementary Fig. 1?

Why do the MFI values of CD25 obviously differ inappropriately and tremendously between the four cohorts in Supplementary Fig. 10? If this is based on differences in the isotype stainings they should provide the isotype control plots in the figure.

minor:

Supple Fig.8. C57BL76?

Supple Fig. 10. FACs?

Reviewer #4 (Remarks to the Author):

The authors have done an excellent job in considering and responding satisfactorily to all of this reviewer's comments.

Only one very minor item needs correction. Fig. 6e indicates that the cleavage site in mouse PAR3

for aPC is Lys37. But the revised paper refers to this cleavage site as Arg37 on p.11, line 9 and p.16 on l5. So the text should be corrected to indicate "Lys37" not Arg37.

Reviewer #1 comments

The authors have substantially worked on the issues brought up, however the manuscript has still major points that need clarification.

Reply:

We thank the reviewer for carefully reviewing our manuscript again. The points raised are addressed as follows:

Major:

1. (Follow up from the authors response to the first review comment #26). Regarding the mouse experiments showing sustained GvL effect after aPC treatment (Fig. 10c, Supple Fig. 13b,c), the tumor burden on day 28 was rather diminished by aPC treatment in two tumor cell lines.

The authors still do not provide an adequate response i.e. a mechanistic hypothesis on how aPC treatment may have enhanced the GvL effect despite increased numbers of Tregs. Their previous response explained virtually nothing about the underlying mechanism.

Moreover, since the authors claim that aPC treatment also decreased Th1 and Th17 cells, then which compartment played the critical role in their model?

In the same context, why didn't the authors in their experiments analyze CD8 T cells at all? In Supplementary Fig. 1 they should at least show the data for CD8 and Treg as well.

Answer:

We appreciate these comments. Regarding the sustained GvL effect following aPC preincubation of T-cells, we did not provide more details concerning potential mechanisms due to space restriction. A sustained or enhanced GvL effect following treatment of mice with T_{regs} has been repeatedly demonstrated [1-4]. Thus, we hypothesize that the increased frequency of T_{regs} following *ex vivo* preincubation with aPC does not only ameliorate GvHD, but also allows an efficient GvL effect. We concur with the reviewer that the dual effect of T_{regs} in regard to GvHD and GvL, as shown by us and others [1-4], is of high interest. Several potential mechanisms have been discussed, and we propose to cite relevant publications (please see new text passage below). Furthermore, it will be interesting to determine whether the preincubation of T-cells with aPC is superior to "simply" using a higher number of T_{regs} (e.g. whether there is simply a quantitative effect of aPC on T_{regs} , or whether there is in addition an qualitative effect of aPC on T_{reg} function). We believe, however, that such studies are beyond the scope of the current manuscript.

Regarding the potential role of the decreased Th1 and Th17 cell frequency: we demonstrate that the expansion of T_{regs} following preincubation with aPC is critical for aPC's effect in respect to GvHD. This was demonstrated by depleting T_{regs} using the DEREK mice. Secondary effects depending on a decreased frequency of Th1 or Th17 cells cannot be excluded, but a pivotal role of the increased T_{reg} number is demonstrated by the depletion of T_{regs} .

Similar approaches could be used to evaluate the role of T_{regs} , Th1, or Th17 cells for aPC's effect on GvL. The feasibility of such approaches has been shown [5]. Furthermore, we concur with the reviewer that $CD8^+$ T cells are an interesting target to investigate, as previously shown by others [5, 6]. We detected expression of PAR3 on $CD8^+$ T-cells, albeit at lower levels than on T_{regs} (Fig. 6b of the main manuscript), making $CD8^+$ cells a less attractive target in the context of the current study. Considering the amount of data shown and the length of the manuscript we don't think that it is possible to appropriately address aPC's effect on other T-cell population within the current manuscript.

We thank the reviewer for bringing these points to our attention and we will follow up on these points in future studies. If agreed by the reviewer and the editor we will add the following passage to the discussion to address these points (new text passage shown in red):

"While expansion of T_{regs} following *ex vivo* preincubation of pan T-cells with aPC protects from GvHD, it does not compromise the GvL effect when using two independent tumor cell models. **The sustained GvL effect following preincubation of T-cells with aPC and subsequent expansion of T_{regs} is congruent with previous studies demonstrating that T_{regs} do not only prevent GvHD, but simultaneously convey an efficient GvL effect [1-4]. Several mechanisms have been proposed for the dual effect of T_{regs} in regard to GvHD and the GvL effect, e.g. inhibition of JAK1/JAK2 signalling, expression of NKG2D by $CD8^+$ cells, IL-21 signalling, or the differential expression of granzyme B by $CD4^+CD25^+$ and $CD8^+$ cells [7-10]. Whether these mechanisms are regulated by aPC remains unknown. Likewise, the relevance of other T-cell populations, such as Th1, Th17, or $CD8^+$ T-cells for aPC's effects as observed in the current study remains to be evaluated"** (new text passage to be placed on page 15, between line 14 and 15).

2. Follow up from the previous authors response to my comment #3. They should address more specifically which cells other than T cells produced Treg-inducible cytokines, and provide some data or references. Their comment to the reviewer is contradictory in itself.

We apologize that we did not address this question appropriately. In addition to lymphocytes macrophages, platelets, liver endothelial cells, and parenchymal cells can produce TGF β and potentially promote T_{reg} induction [11-13]. Similar, IL-10 is known to be produced by various cell types, including macrophages, neutrophils, dendritic cells, B cells, and different subsets of CD4⁺ and CD8⁺ T cells [14, 15]. As coagulation proteases regulate in particular macrophages, neutrophils, and platelets we envision that coagulation factors, including aPC, may regulate TGF β and / or IL-10 secretion from these cells [16-21]. Which cell types are targeted by aPC to regulate cytokines promoting T_{regs} differentiation remains to be evaluated in independent studies.

If agreed by the editor we will add the following sentence to the discussion to address this point (new text passage shown in red):

“While the plasma cytokine profile in mice is in agreement with the cytokine expression pattern observed in murine donor CD4⁺ T-cells, it is likely that cells other than T-cells contributed to the plasma cytokine profile. **Of note, IL-10 and TGF β are produced by various cell types, including macrophages, neutrophils, or platelets, which are subject to regulation by coagulation proteases [11, 14-21]**” (new text passage to be placed on page 7, between line 16 and 17).

3. (Follow up from the authors response to previous comment #5). Their response was not what I meant. What is the strategy for defining positivity on FACS plots of Supplementary Fig. 1, Supplementary Fig2, Supplementary 3ab, Supplementary 6a, Supplementary 7a and Supplementary Fig. 10.

First, unstained controls did not work so that the MFI values of negative subpopulations differed between unstained controls and stained samples.

Second, the differences in the frequency of T-bet⁺ and ROR-gt⁺ cells in control plots between B6T+BM and B6T(aPC)+BM (Supplementary Fig. 2) are not acceptable.

Third, controls stained only for CD4 are not appropriate for the true control when they stain multiple cytokines or intranuclear molecules at the same time.

Along with this context, the authors should show the FACS data for isotype controls at least for cytokines and transcription factors staining in Supplementary Fig2, Supplementary 3ab, Supplementary 6a, Supplementary 7a and Supplementary Fig. 10.

They should also pay attention to the fact that almost all of the MFI data DO NOT provide statistical significance.

In addition, why were CD3⁺ and CD3⁻ subpopulations not clearly discriminated in Supplementary Fig. 1?

Why do the MFI values of CD25 obviously differ inappropriately and tremendously between the four cohorts in Supplementary Fig. 10? If this is based on differences in the isotype stainings they should provide the isotype control plots in the figure.

Answer:

We appreciate the detailed comments in regard to the FACS analyses. Contributing authors met to discuss these comments. Briefly, we followed standard procedures for staining of samples and gating [22-24].

1. Staining of samples

All antibodies were carefully titrated to reduce antibody specific background. In addition, Fc-binding reagent was used to block non-specific staining of samples in all cases. After intracellular staining of transcription factors or cytokines samples were washed 3 times with staining buffer, followed by a final washing step with PBS and re-suspension of the cells in PBS-BSA (0.2%). Cells were then used for data acquisition.

2. Gating control

As we carefully titrated antibodies to reduce their background staining and as we typically used 2 or 3 colours for each sample we used FMO controls (fluorescent minus one) to set up the gating strategy [22-24]. FMO controls have the advantage that they account for spillover effects on the channel of interest [22].

In a pilot experiment we initially attempted to use isotype controls for intracellular staining of transcription factors and cytokines, but we observed inappropriate variations and differences in the background staining when comparing test antibodies with corresponding isotype control antibodies. Such limitations of isotype control antibodies are known [22].

Furthermore, the conclusions drawn within the manuscript are based on comparison with biological controls. Biological comparison controls are frequently considered to be the most relevant control to determine an effect [22].

We address the specific comments as follows:

1. Representative gating strategies are now shown for supplementary Figures 1, 2, 3, 6, 7, and 10 (see “attachment-01”). This new information can be included within the manuscript as supplementary information. Furthermore, an improved version of supplementary Figure 1 is included as “attachment-02”.
2. As we are not showing MFIs for unstained controls we assume that the reviewer is referring to the different frequencies shown in the negative control samples of some FACS plots (e.g. Supplementary Fig. 2, T-bet and ROR- γ t). As outlined above, we carefully titrated antibodies to avoid non-specific staining as much as possible and we used FMO as controls for our FACS analyses. Conclusions are drawn based on comparison of different biological samples. However, we concur with the reviewer that in particular the controls shown in Supplementary Fig. 2 for T-bet and ROR- γ t were not appropriate (see next answer).
3. Regarding the frequency of T-bet⁺ and ROR- γ t⁺ cells in control plots between B6T+BM and B6T(aPC)+BM (Supplementary Fig. 2) we re-analysed the data. When control plots are used which show an equal frequency of T-bet⁺ and ROR- γ t⁺ cells the absolute numbers for “stained” samples are overall slightly lower, but the differences remain significant. The corrected FACS-plots and the corresponding corrected Fig. 3e are included in “attachment-03”. We will incorporate these corrected figures into the manuscript if the manuscript is accepted.
4. We would like to point out that we did not stain for multiple intracellular cytokines or transcription factors. We conducted double staining for CD4 and the transcription factor or cytokine of interest. In these cases CD4 served as FMO control as outlined above.
5. As outlined above, we used FMO as control [22-24]. To support our conclusion we determined cytokine levels in the plasma (Fig. 3g) and in the supernatant of stimulated cells (Fig. 4f) or expression of FOXP3 by immunoblotting (Fig. 5e). The results from these analyses corroborate the conclusions drawn from the FACS analyses.
6. We appreciate the comment regarding the MFI data. In the first draft of the manuscript we provided iMFI (integrated mean fluorescent intensity) data, and reviewers requested us to show the FACS plots (frequency) and the MFI separately. We concur with this approach, as the frequency (in %, as shown in the FACS plots and corresponding bar-graphs) provides information regarding the relative cell-population size, while the MFI provides (semi-)quantitative information regarding the fluorescent signal strength (which depends on various factors, including the number of antibodies bound to a cell and the voltage setting). Accordingly, the information provided by these measures is not the same and hence differences – as in our case – can be observed. We checked our primary data and would like to emphasize that some of the MFI differences are of borderline significance (see “attachment-04”). If accepted, we can include this information in the manuscript.
7. CD3⁺ cells in Supplementary Figure 1: We thank the reviewer for bringing this issue to our attention. We checked available FACS plot images and provide a revised version of supplementary Figure 1 showing a FACS plot which clearly discriminates CD3⁻ and CD3⁺ populations (see “attachment-02”).
8. Supplementary Figure 10 does not show MFI, but the frequency. Here we stimulated human T_{eff} cells with antigen presenting cells (AgPC) and compared them to non-stimulated controls (T_{eff}), both in the absence (top) or with aPC preincubation. In humans, CD4⁺CD25^{high} cells convey the function of T_{regs}, while CD4⁺CD25^{intermediate} cells have no immunosuppressive function [25-29]. Following activation we observe, as expected, a shift of CD25⁺ cells, which reflects both T_{eff} activation (CD25^{intermediate}) and T_{reg} induction (CD25^{high}). We gated on the CD25^{high} cells, which are a distinct population in the plots. To corroborate induction of T_{regs} cells we determine in parallel FOXP3 expression by immunoblotting (Fig. 5f).

minor:

Supple Fig.8. C57BL/6?

Supple Fig. 10. FACS?

Answer:

We thank this reviewer for bringing these typing errors to our attention. These errors will be corrected in a finalized version of the manuscript, if the manuscript is accepted.

References:

- 1 Jones SC, Murphy GF, Korngold R. Post-hematopoietic cell transplantation control of graft-versus-host disease by donor CD425 T cells to allow an effective graft-versus-leukemia response. *Biology of blood and marrow transplantation : journal of the American Society for Blood and Marrow Transplantation*. 2003; **9**: 243-56. 10.1053/bbmt.2003.50027.
- 2 Edinger M, Hoffmann P, Ermann J, Drago K, Fathman CG, Strober S, Negrin RS. CD4+CD25+ regulatory T cells preserve graft-versus-tumor activity while inhibiting graft-versus-host disease after bone marrow transplantation. *Nature medicine*. 2003; **9**: 1144-50. 10.1038/nm915.
- 3 Trenado A, Charlotte F, Fisson S, Yagello M, Klatzmann D, Salomon BL, Cohen JL. Recipient-type specific CD4+CD25+ regulatory T cells favor immune reconstitution and control graft-versus-host disease while maintaining graft-versus-leukemia. *The Journal of clinical investigation*. 2003; **112**: 1688-96. 10.1172/JCI17702.
- 4 Yao Y, Wang L, Zhou J, Zhang X. HIF-1alpha inhibitor echinomycin reduces acute graft-versus-host disease and preserves graft-versus-leukemia effect. *Journal of translational medicine*. 2017; **15**: 28. 10.1186/s12967-017-1132-9.
- 5 Gartlan KH, Markey KA, Varelias A, Bunting MD, Koyama M, Kuns RD, Raffelt NC, Olver SD, Lineburg KE, Cheong M, Teal BE, Lor M, Comerford I, Teng MW, Smyth MJ, McCluskey J, Rossjohn J, Stockinger B, Boyle GM, Lane SW, Clouston AD, McColl SR, MacDonald KP, Hill GR. Tc17 cells are a proinflammatory, plastic lineage of pathogenic CD8+ T cells that induce GVHD without antileukemic effects. *Blood*. 2015; **126**: 1609-20. 10.1182/blood-2015-01-622662.
- 6 Zheng J, Liu Y, Liu Y, Liu M, Xiang Z, Lam KT, Lewis DB, Lau YL, Tu W. Human CD8+ regulatory T cells inhibit GVHD and preserve general immunity in humanized mice. *Science translational medicine*. 2013; **5**: 168ra9. 10.1126/scitranslmed.3004943.
- 7 Carniti C, Gimondi S, Vendramin A, Recordati C, Confalonieri D, Bermema A, Corradini P, Mariotti J. Pharmacologic Inhibition of JAK1/JAK2 Signaling Reduces Experimental Murine Acute GVHD While Preserving GVT Effects. *Clinical cancer research : an official journal of the American Association for Cancer Research*. 2015; **21**: 3740-9. 10.1158/1078-0432.CCR-14-2758.
- 8 Karimi MA, Bryson JL, Richman LP, Fesnak AD, Leichner TM, Satake A, Vonderheide RH, Raulet DH, Reshef R, Kambayashi T. NKG2D expression by CD8+ T cells contributes to GVHD and GVT effects in a murine model of allogeneic HSCT. *Blood*. 2015; **125**: 3655-63. 10.1182/blood-2015-02-629006.
- 9 Hanash AM, Kappel LW, Yim NL, Nejat RA, Goldberg GL, Smith OM, Rao UK, Dykstra L, Na IK, Holland AM, Dudakov JA, Liu C, Murphy GF, Leonard WJ, Heller G, van den Brink MR. Abrogation of donor T-cell IL-21 signaling leads to tissue-specific modulation of immunity and separation of GVHD from GVL. *Blood*. 2011; **118**: 446-55. 10.1182/blood-2010-07-294785.
- 10 Du W, Leigh ND, Bian G, Alqassim E, O'Neill RE, Mei L, Qiu J, Liu H, McCarthy PL, Cao X. Granzyme B Contributes to the Optimal Graft-Versus-Tumor Effect Mediated by Conventional CD4+ T Cells. *J Immunol Res Ther*. 2016; **1**: 22-8.
- 11 Branton MH, Kopp JB. TGF-beta and fibrosis. *Microbes Infect*. 1999; **1**: 1349-65.
- 12 Carambia A, Freund B, Schwinge D, Heine M, Laschtowitz A, Huber S, Wraith DC, Korn T, Schramm C, Lohse AW, Heeren J, Herkel J. TGF-beta-dependent induction of CD4(+)CD25(+)Foxp3(+) Tregs by liver sinusoidal endothelial cells. *Journal of hepatology*. 2014; **61**: 594-9. 10.1016/j.jhep.2014.04.027.
- 13 Yang H, Sun J, Li Y, Duan WM, Bi J, Qu T. Human umbilical cord-derived mesenchymal stem cells suppress proliferation of PHA-activated lymphocytes in vitro by inducing CD4(+)CD25(high)CD45RA(+) regulatory T cell production and modulating cytokine secretion. *Cell Immunol*. 2016; **302**: 26-31. 10.1016/j.cellimm.2016.01.002.
- 14 Couper KN, Blount DG, Riley EM. IL-10: the master regulator of immunity to infection. *Journal of immunology*. 2008; **180**: 5771-7.
- 15 Saraiva M, O'Garra A. The regulation of IL-10 production by immune cells. *Nature reviews Immunology*. 2010; **10**: 170-81. 10.1038/nri2711.
- 16 Liang HP, Kerschen EJ, Hernandez I, Basu S, Zogg M, Botros F, Jia S, Hessner MJ, Griffin JH, Ruf W, Weiler H. EPCR-dependent PAR2 activation by the blood coagulation initiation complex regulates LPS-triggered interferon responses in mice. *Blood*. 2015; **125**: 2845-54. 10.1182/blood-2014-11-610717.
- 17 Braach N, Frommhold D, Buschmann K, Pflaum J, Koch L, Hudalla H, Staudacher K, Wang H, Isermann B, Nawroth P, Poeschl J. RAGE controls activation and anti-inflammatory signalling of protein C. *PLoS one*. 2014; **9**: e89422. 10.1371/journal.pone.0089422.
- 18 Yang XV, Banerjee Y, Fernandez JA, Deguchi H, Xu X, Mosnier LO, Urbanus RT, de Groot PG, White-Adams TC, McCarty OJ, Griffin JH. Activated protein C ligation of ApoER2 (LRP8) causes Dab1-dependent signaling in U937 cells. *Proceedings of the National Academy of Sciences of the United States of America*. 2009; **106**: 274-9. 10.1073/pnas.0807594106.

- 19 Shua F, Kobayashia H, Fukudomeb K, Tsuneyoshib N, Kimotob M, Teraoa T. Activated protein C suppresses tissue factor expression on U937 cells in the endothelial protein C receptor-dependent manner. *FEBS Lett.* 2000; **477**: 208-12.
- 20 Cao C, Gao Y, Li Y, Antalis TM, Castellino FJ, Zhang L. The efficacy of activated protein C in murine endotoxemia is dependent on integrin CD11b. *The Journal of clinical investigation.* 2010; **120**: 1971-80. 10.1172/JCI40380.
- 21 Elphick GF, Sarangi PP, Hyun YM, Hollenbaugh JA, Ayala A, Biffi WL, Chung HL, Rezaie AR, McGrath JL, Topham DJ, Reichner JS, Kim M. Recombinant human activated protein C inhibits integrin-mediated neutrophil migration. *Blood.* 2009; **113**: 4078-85. 10.1182/blood-2008-09-180968.
- 22 Maecker HT, Trotter J. Flow cytometry controls, instrument setup, and the determination of positivity. *Cytometry A.* 2006; **69**: 1037-42. 10.1002/cyto.a.20333.
- 23 Roederer M. Spectral compensation for flow cytometry: visualization artifacts, limitations, and caveats. *Cytometry.* 2001; **45**: 194-205.
- 24 Alvarez DF, Helm K, Degregori J, Roederer M, Majka S. Publishing flow cytometry data. *American journal of physiology Lung cellular and molecular physiology.* 2010; **298**: L127-30. 10.1152/ajplung.00313.2009.
- 25 Chen W, Jin W, Hardegen N, Lei KJ, Li L, Marinos N, McGrady G, Wahl SM. Conversion of peripheral CD4+CD25- naive T cells to CD4+CD25+ regulatory T cells by TGF-beta induction of transcription factor Foxp3. *The Journal of experimental medicine.* 2003; **198**: 1875-86. 10.1084/jem.20030152.
- 26 Jacob C, Yang PC, Darmoul D, Amadesi S, Saito T, Cottrell GS, Coelho AM, Singh P, Grady EF, Perdue M, Bunnett NW. Mast cell tryptase controls paracellular permeability of the intestine. Role of protease-activated receptor 2 and beta-arrestins. *The Journal of biological chemistry.* 2005; **280**: 31936-48. 10.1074/jbc.M506338200.
- 27 Lundgren A, Stromberg E, Sjoling A, Lindholm C, Enarsson K, Edebo A, Johnsson E, Suri-Payer E, Larsson P, Rudin A, Svennerholm AM, Lundin BS. Mucosal FOXP3-expressing CD4+ CD25high regulatory T cells in Helicobacter pylori-infected patients. *Infection and immunity.* 2005; **73**: 523-31. 10.1128/IAI.73.1.523-531.2005.
- 28 Duhon T, Duhon R, Lanzavecchia A, Sallusto F, Campbell DJ. Functionally distinct subsets of human FOXP3+ Treg cells that phenotypically mirror effector Th cells. *Blood.* 2012; **119**: 4430-40. 10.1182/blood-2011-11-392324.
- 29 Nyirenda TS, Molyneux ME, Kenefeck R, Walker LS, MacLennan CA, Heyderman RS, Mandala WL. T-Regulatory Cells and Inflammatory and Inhibitory Cytokines in Malawian Children Residing in an Area of High and an Area of Low Malaria Transmission During Acute Uncomplicated Malaria and in Convalescence. *J Pediatric Infect Dis Soc.* 2015; **4**: 232-41. 10.1093/jpids/piu140.

Attachment-01: Representative gating strategy for unstained control and stained samples in Supplementary Figure 1, part 1

Unstained Control

Stained

Attachment-01: Representative gating strategy for FMO control and stained samples in Supplementary Figure 1, part 2

Attachment-01: Representative gating strategy for FMO control and stained samples in Supplementary Figure 2

FMO Control (B6T+BM)

Stained (B6T+BM)

Attachment-01: Representative gating strategy for FMO control and stained samples in Supplementary Figure 3

FMO Control (B6T(aPC)+BM)

Stained (B6T(aPC)+BM)

**Attachment-01: Representative gating strategy for control and stained samples in
Supplementary Figure 6**

Control (only CD4 stained)

Stained (T+AgPC)

Stained (T(aPC)+AgPC)

Attachment-01: Representative gating strategy for control and stained samples in Supplementary Figure 7

Control (only CD4 stained)

Stained (T+AgPC)

Stained (T(aPC)+AgPC)

Attachment-01: Representative gating strategy for unstained control and stained samples in Supplementary Figure 10

Supplementary Fig. 1: Engraftment of donor cells ($H2^{b+}$) and donor $CD3^+$ and $CD4^+$ T-cells is not altered by aPC-preincubation of T-cells

Recipient BALB/c mice were lethally irradiated (11Gy) and transplanted with 5×10^6 bone marrow and 0.5×10^6 T-cells without (B6T+BM) or with (B6T(aPC)+BM) aPC-preincubation (20nM, 1h, 37°C) from donor C57BL/6 wt mice. Recipient mice were sacrificed on day 3 post transplantation, splenic cells were harvested and stained for $H2^b$, CD3, or CD4 and analysed by flow cytometry. Control: unstained (top) or stained for $H2^b$ only (middle, bottom). Representative FACS images are shown.

3e

Supplementary Fig. 2 : Exemplary FACS-scans and MFI corresponding to Fig. 3e

Recipient BALB/c mice were lethally irradiated (11Gy) and transplanted with 5×10^6 bone marrow and 0.5×10^6 T-cells without (B6T+BM) or with (B6T(aPC)+BM) aPC-preincubation (20nM, 1h, 37°C) from donor C57BL/6 wt mice. Recipient mice were sacrificed 2 weeks post transplantation, splenic T-cells were harvested and stained for H2^b, CD4, T-bet, ROR- γ t, or FOXP3 and analysed by flow cytometry. For T-bet, ROR- γ t, and FOXP3 cells were gated on H2^b+CD4⁺ cells. Control: stained for H2^b and CD4. Representative FACS images (a) and Mean Fluorescence Intensity (b, MFI, Mean value \pm SEM; N=5 each group) are shown.

Supplementary Fig. 2b

Supplementary Fig. 3b